# Dual functionality of the amyloid protein TasA in *Bacillus* physiology and fitness on the phylloplane

Jesús Cámara-Almirón [1,3], Yurena Navarro[1,3], Luis Díaz-Martínez[1], María Concepción Magno-Pérez-Bryan[1], Carlos Molina-Santiago [1], John R. Pearson [2], Antonio de Vicente [1], Alejandro Pérez-García [1] & Diego Romero [1✉]

Bacteria can form biofilms that consist of multicellular communities embedded in an extracellular matrix (ECM). In *Bacillus subtilis*, the main protein component of the ECM is the functional amyloid TasA. Here, we study further the roles played by TasA in *B. subtilis* physiology and biofilm formation on plant leaves and in vitro. We show that Δ*tasA* cells exhibit a range of cytological symptoms indicative of excessive cellular stress leading to increased cell death. TasA associates to the detergent-resistant fraction of the cell membrane, and the distribution of the flotillin-like protein FloT is altered in Δ*tasA* cells. We propose that, in addition to a structural function during ECM assembly and interactions with plants, TasA contributes to the stabilization of membrane dynamics as cells enter stationary phase.

---

[1] Instituto de Hortofruticultura Subtropical y Mediterránea "La Mayora" – Departamento de Microbiología, Universidad de Málaga, Bulevar Louis Pasteur 31 (Campus Universitario de Teatinos), 29071 Málaga, Spain. [2] Nano-imaging Unit, Andalusian Centre for Nanomedicine and Biotechnology, BIONAND, Málaga, Spain. [3] These authors contributed equally: Jesús Cámara-Almirón, Yurena Navarro. ✉email: diego_romero@uma.es

In response to a wide range of environmental factors[1,2], some bacterial species establish complex communities called biofilms[3]. To do so, planktonic cells initiate a transition into a sedentary lifestyle and trigger a cell differentiation program that leads to: (1) a division of labor, in which different subpopulations of cells are dedicated to covering different processes needed to maintain the viability of the community[4,5], and (2) the secretion of a battery of molecules that assemble the extracellular matrix (ECM)[3,6].

Studies of *Bacillus subtilis* biofilms have contributed to our understanding of the intricate developmental program that underlies biofilm formation[7–10] that ends with the secretion of ECM components. It is known that the genetic pathways involved in biofilm formation are active during the interaction of several microbial species with plants[11,12]. In *B. subtilis*, the lipopeptide surfactin acts as a self-trigger of biofilm formation on the melon phylloplane, which is connected with the suppressive activity of this bacterial species against phytopathogenic fungi[13]. Currently, the *B. subtilis* ECM is known to consist mainly of exopolysaccharide (EPS) and the TasA and BslA proteins[7]. The EPS acts as the adhesive element of the biofilm cells at the cell-to-surface interface, which is important for biofilm attachment[14], and BslA is a hydrophobin that forms a thin external hydrophobic layer and is the main factor that confers hydrophobic properties to biofilms[15]. Both structural factors contribute to maintain the defense function performed by the ECM[11,15]. TasA is a functional amyloid protein that forms fibers resistant to adverse physicochemical conditions that confer biofilms with structural stability[16,17]. Additional proteins are needed for the polymerization of these fibers: TapA appears to favor the transition of TasA into the fiber state, and the signal peptidase SipW processes both proteins into their mature forms[18,19]. The ability of amyloids to transition from monomers into fibers represents a structural, biochemical, and functional versatility that microbes exploit in different contexts and for different purposes[20].

Like in eukaryotic tissues, the bacterial ECM is a dynamic structure that supports cellular adhesion, regulates the flux of signals to ensure cell differentiation[21,22], provides stability and serves as an interface with the external environment, working as a formidable physicochemical barrier against external assaults[23–25]. In eukaryotic cells, the ECM plays an important role in signaling[26,27] and has been described as a reservoir for the localization and concentration of growth factors, which in turn form gradients that are critical for the establishment of developmental patterning during morphogenesis[28–30]. Interestingly, in senescent cells, partial loss of the ECM can influence cell fate, e.g., by activating the apoptotic program[31,32]. In both eukaryotes and prokaryotes, senescence involves global changes in cellular physiology, and in some microbes, this process begins with the entry of the cells into stationary phase[33–35]. This process triggers a response typified by molecular mechanisms evolved to overcome environmental adversities and to ensure survival, including the activation of general stress response genes[36,37], a shift to anaerobic respiration[38], enhanced DNA repair[39], and induction of pathways for the metabolism of alternative nutrient sources or sub-products of primary metabolism[40].

Based on previous works[13], we hypothesize that the ECM makes a major contribution to the ecology of *B. subtilis* in the poorly explored phyllosphere. Our study of the ecology of *B. subtilis* NCIB3610-derived strains carrying single mutations in different ECM components in the phyllosphere highlights the role of TasA in bacteria-plant interactions. Moreover, we demonstrate a complementary role for TasA in the stabilization of the bacteria's physiology. In Δ*tasA* cells, gene expression changes and dynamic cytological alterations eventually lead to a premature increase in cell death within the colony. Complementary evidences prove that these alterations are independent of the structural role of TasA in ECM assembly. All these results indicate that these two complementary roles of TasA, both as part of the ECM and in contributing to the regulation of cell membrane dynamics, are important to preserve cell viability within the colony and for the ecological fitness of *B. subtilis* in the phylloplane.

## Results

**TasA contributes to the fitness of *Bacillus* on the phylloplane.** Surfactin, a member of a subfamily of lipopeptides produced by *B. subtilis* and related species, contributes to multicellularity in *B. subtilis* biofilms[41]. We previously reported how a mutant strain defective for lipopeptide production showed impaired biofilm assembly on the phylloplane[13]. These observations led us to evaluate the specific contributions made by the ECM structural components TasA and the EPS to *B. subtilis* fitness on melon leaves. Although not directly linked to the surfactin-activated regulatory pathway, we also studied the gene encoding the hydrophobin protein BslA (another important ECM component). A *tasA* mutant strain (Δ*tasA*) is defective in the initial cell attachment to plant surfaces (4 h and 2 days post-inoculation) (Fig. 1A). As expected, based on their structural functions, all of the matrix mutants showed reduced adhesion and survival (Supplementary Figs. 1A and 1B); however, the population of Δ*tasA* cells continuously and steadily decreased over time compared to the populations of *eps* or *bslA* mutant cells (Fig. 1B and Supplementary Fig. 1B). Examination of plants inoculated with the wild-type strain (WT) or with the Δ*tasA* strain via scanning electron microscopy (SEM) revealed variability in the colonization patterns of the strains. WT cells assembled in ordered and compact colonies, with the cells embedded in a network of extracellular material (Fig. 1C, top). In contrast, the Δ*tasA* cells were prone to irregular distribution as large masses of cells on the leaves, which also showed collapsed surfaces or lack of surface integrity, suggesting alterations in cellular structures (Fig. 1C, center). Finally, *eps* and *bslA* mutant cells formed flat colonies (Supplementary Fig. 2A) with the same colonization defects observed in the *tasA* mutant cells (Supplementary Fig. 1C).

Based on the reduced fitness exhibited by the single ECM component mutant strains and their deficiencies in biofilm formation, we hypothesized that these strains may also be defective in their antagonistic interactions with *Podosphaera xanthi* (an important fungal biotrophic phytopathogen of crops[42]) on plant leaves. Strains with mutations in *eps* and *bslA* partially ameliorated the disease symptoms, although their phenotypes were not significantly different from those of the WT strain (Supplementary Fig. 1D). However, contrary to our expectations, the Δ*tasA* strain retained similar antagonistic activity to that of the WT strain (Fig. 1D). The simplest explanation for this finding is that the antifungal activity exhibited by the Δ*tasA* cells is due to higher production of antifungal compounds. In situ mass spectrometry analysis revealed a consistently higher relative amount of the antifungal compound plipastatin (also known as fengycin, the primary antifungal compound produced by *B. subtilis*) on leaves treated with Δ*tasA* cells compared to those treated with WT cells (Fig. 1E). These observations argue in favor of the relevance of the ECM and specifically TasA in the colonization, survival, and antagonistic activity of *B. subtilis* on the phylloplane.

**Loss of TasA causes a global change in bacterial cell physiology.** The increased fengycin production and the previously reported deregulation of the expression pattern of the *tapA* operon in a Δ*tasA* mutant strain[23] led us to explore whether loss of *tasA*

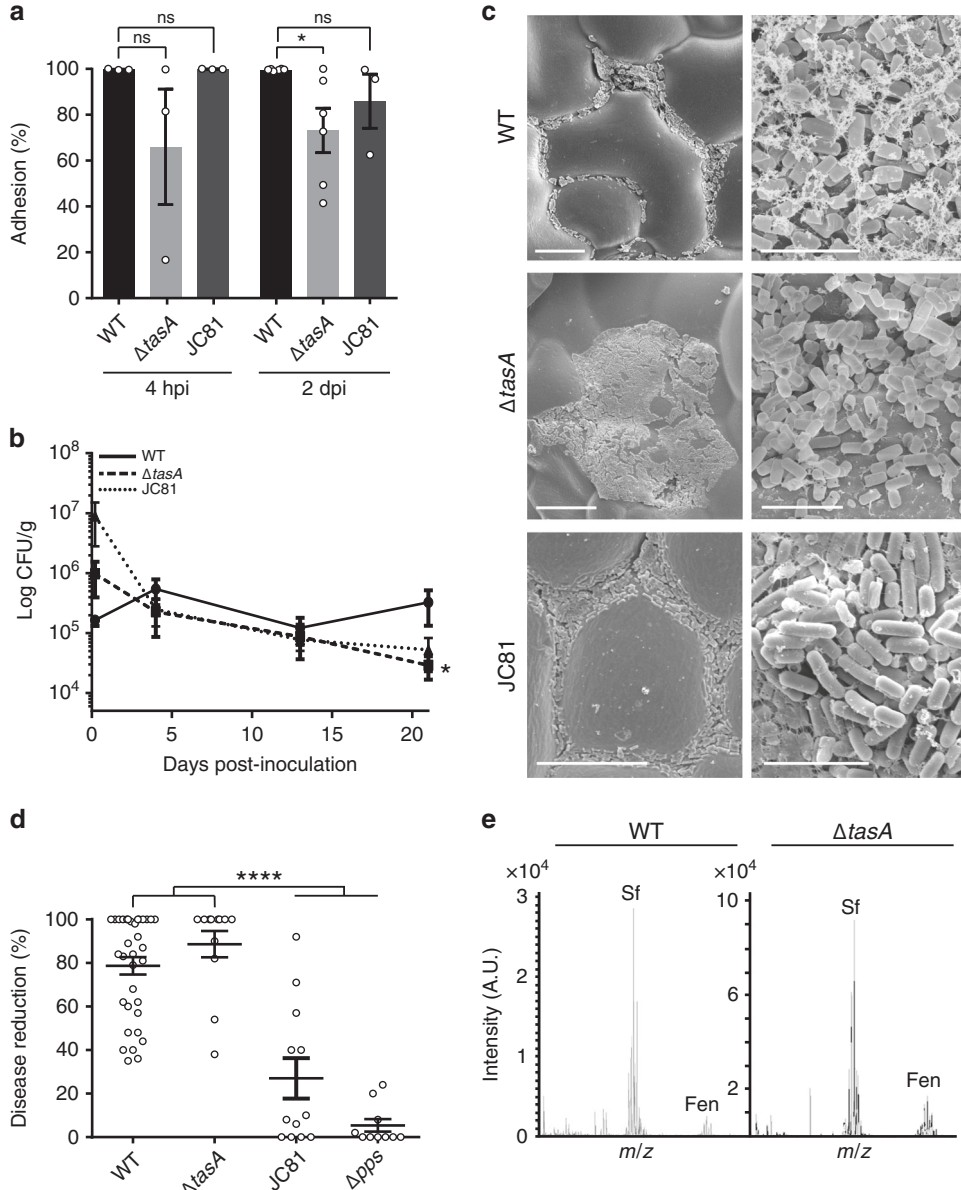

**Fig. 1 TasA is essential for the fitness of *Bacillus* on the melon phylloplane. a** Adhesion of the WT, Δ*tasA* and JC81 (TasA Lys68Ala, Asp69Ala) strains to melon leaves at 4 h (hpi) and 2 days post-inoculation (dpi). Statistically significant differences between WT and Δ*tasA* were found at 2 dpi. At 4 hpi, $N = 3$ for all the strains. At 2 dpi $N = 6$ for the WT strain, $N = 6$ for the Δ*tasA* strain and $N = 3$ for the JC81 strain. $N$ refers to the number independent experiments. In each experiment, 10 leaves were analyzed. Average values are shown. Error bars represent the SEM. Statistical significance was assessed via two-tailed independent *t*-tests at each time-point (*$p$ value = 0.0262). **b** The persistence of the Δ*tasA* cells at 21 days was significantly reduced compared with that of the WT cells. The persistence of JC81 cells on melon leaves was reduced compared to that of the WT cells. The first point is taken at 4 hpi. Average values of five biological replicates are shown with error bars representing the SEM. Statistical significance was assessed by two-tailed independent *t*-test at each time-point (*$p$ value = 0.0329). **c** Representative scanning electron microscopy micrographs of inoculated plants taken 20 days post-inoculation show the WT cells (top) distributed in small groups covered by extracellular material and the Δ*tasA* cells (bottom) in randomly distributed plasters of cells with no visible extracellular matrix. JC81 (TasA Lys68Ala, Asp69Ala) strain shows an intermediate colonization pattern between those of the WT and Δ*tasA* null mutant strains. Scale bars = 25 μm (left panels) and 5 μm (right panels). Experiments have been repeated at least three times with similar results. **d** The WT and Δ*tasA* strains showed comparable biocontrol activity against the fungal phytopathogen *Podosphaera xanthii*. However, JC81 (TasA Lys68Ala, Asp69Ala) failed to control the disease. Biocontrol activity was measured after 15 days post-inoculation of the pathogen. For the WT strain, $N = 22$. For the Δ*tasA* strain and JC81 strain, $N = 12$. For the Δ*pps* strain, $N = 9$. N refers to number of plants analyzed over three independent assays. Three leaves per plant were infected and inoculated. Average values are shown with error bars indicating the SEM. The Δ*pps* is a mutant strain in fengycin production and it is used as a negative control. Statistical significance was assessed by two-tailed independent Mann–Whitney tests between each strain and the Δ*pps* mutant (****$p < 0.0001$). **e** MALDI-TOF/TOF MS analysis revealed higher fengycin levels on melon leaves treated with Δ*tasA* (right) cells compared with that on leaves treated with WT cells (left) after 20 days post-inoculation. Source data are provided as a Source Data file.

disrupts the genetic circuitry of the cells. We sequenced and analyzed the whole transcriptomes of ΔtasA and WT cells grown in vitro on MSgg agar plates, a chemically defined medium specifically formulated to support biofilm. We observed that deletion of tasA resulted in pleiotropic effects on the overall gene expression profile of this mutant (Fig. 2A and Supplementary Fig. 3A), with 601, 688, and 333 induced genes and 755, 1077, and 499 repressed genes at 24, 48, and 72 h, respectively (Supplementary Fig. 3A). A closer look at the data allowed us to cluster the expression of different genes into groups with similar expression profiles over the time course (Fig. 2A). In general, four different expression profiles were found in which genes show positive (profiles 1 and 2) or negative (profiles 3 and 4) variations from 24 h to 48 h, and genes with expression levels that remain either stable (profiles 1 and 3) or altered (profiles 2 and 4) from 48 h to 72 h (Fig. 2A). Profiles 1 and 2 included genes related to the SOS response (profile 1), transcription and replication (profile 1 and 2), purine biosynthetic process (profile 2), and toxin-antitoxin systems (profile 2). Profiles 3 and 4 included genes related to sporulation (profiles 3 and 4), cellular metabolism in general (profile 3) and lipids (profile 3), carbohydrates (profile 3 and 4), monosaccharides (profile 3), polysaccharides (profile 4), or peptidoglycans (profile 3) in particular. These gene expression profiles reflect a general picture that suggests: (i) the existence of cellular stress and DNA damage, in which the cells needs to fully activate different sets of genes to cope with and compensate for the damage and maintain viability, (ii) a decrease in the overall cellular energy metabolism, and (iii) strong repression of the sporulation pathway. To study the observed alterations in gene expression in ΔtasA cells, the differentially expressed genes at all the time points were classified into their different regulons. Indeed, the sigK, sigG, gerR, and gerE regulators (Supplementary Data 2 and 3), which control the expression of many of the genes related to sporulation, were repressed in the ΔtasA cells from 48 h (Fig. 2B and Supplementary Fig. 3B), consistent with the delayed sporulation defect previously reported in ECM mutants[10,23] (Supplementary Fig. 4). In contrast, the expression levels of biofilm-related genes, including the epsA-O, and tapA operons, were higher in the ΔtasA cells at all times compared to their expression levels in WT cells (Fig. 2B and Supplementary Fig. 3B) (Supplementary Data 1–3). We found repression of sinR at 24 h (Supplementary Data 1), induction of the slrR transcriptional regulator at all times (Supplementary Data 1–3), and repression of the transition state genes transcriptional regulator abrB at 24 h and 48 h (Supplementary Data 1 and Supplementary Data 2), which could explain the induction of the ECM-related genes[7].

The analysis of the transcriptional changes in the tasA mutant cells highlighted the broad metabolic rearrangements that take place in ΔtasA colonies from 24 h to 72 h, including the expression alteration of genes implicated in energy metabolism, secondary metabolism and general stress, among other categories (Supplementary Data 1–3, Fig. 2B and Supplementary Fig. 3B). First, the alsS and alsD genes, which encode acetolactate synthase and acetolactate decarboxylase, respectively, were clearly induced at all times (Supplementary Data 1–3). This pathway feeds pyruvate into acetoin synthesis, a small four-carbon molecule that is produced in B. subtilis during fermentative and overflow metabolism[43]. Additionally, we found induction of several regulators and genes that are involved in anaerobic respiration and fermentative metabolism. The two-component regulatory system resD and resE, which senses oxygen limitation, and their target genes[44], were induced in ΔtasA cells at 24 h and 48 h (Supplementary Data 1 and 2 and Supplementary Fig. 3B). Consistently, induction of the transcriptional regulator fnr and the anaerobic respiration related genes narGHIJK, which encode the nitrate reductase complex, as well as all of the proteins

required for nitrate respiration were induced at 24 h and 48 h (Supplementary Data 1 and 2 and Supplementary Fig. 3B). Second, we observed induction at all times of the genes involved in fengycin biosynthesis (Supplementary Data 1–3), consistent with the overproduction of this antifungal lipopeptide in planta (Fig. 1E), genes involved in the biosynthesis of surfactin, subtilosin, bacilysin, and bacillaene (all secondary metabolites with antimicrobial activities[45–48]) (Fig. 2B, Supplementary Fig. 3B and Supplementary Data 1–3), as well as the operon encoding the iron-chelating protein bacillibactin (dhbACEBF) (Supplementary Data 1–3). The induction of all of these genes is possibly due to the repression of transcriptional repressors of transition state genes that occurs at 24 h and 48 h, e.g., abrB (which controls activation of the genes involved in the synthesis of fengycin, bacilysin, subtilosin, and bacillaene) and abh (which contributes to the transcriptional control of the genes involved in surfactin production) (Supplementary Data 1 and 2). The transcriptional changes of other regulators, such as resD (for subtilosin) or comA (for surfactin), both upregulated at 24 h and 48 h (Supplementary Data 1 and 2), also contribute to the induction of the genes that participate in the synthesis of all of these secondary metabolites and might explain their overall activation at 72 h (Fig. 2B). Finally, the gene encoding the regulator AscR was induced at 48 h and 72 h. AscR controls transcription of the snaA (snaAtcyJKLMNcmoOcmoJIrbfKsndAytnM) and yxe (yxeKsnaByxeMyxeNyxeOsndByxeQ) operons which are induced at all times (Supplementary Data 1–3). The products of these operons are members of alternative metabolic pathways that process modified versions of the amino acid cysteine. More specifically, the products of the snaA operon degrade alkylated forms of cysteine that are produced during normal metabolic reactions due to aging of the molecular machinery[40]. The yxe operon is implicated in the detoxification of S-(2-succino)cysteine, a toxic form of cysteine that is produced via spontaneous reactions between fumarate and cellular thiol groups in the presence of excess nutrients, which subsequently leads to increased bacterial stress[49,50].

Additional signs of excess cellular stress in the ΔtasA cells were: (i) the strong overexpression of the sigma factor SigB ($\sigma^B$) at 24 h and 72 h (Fig. 2B, Supplementary Fig. 3B and Supplementary Data 1 and 3), which controls the transcription of genes related to the general stress response[36], and (ii) the repression at 24 h of lexA (Supplementary Data 1), a transcriptional repressor of the SOS response regulon, as well as the induction of other genes that confer resistance to different types of stress, i.e., ahpC and ahpF (induced at all times, Supplementary Data 1–3) against peroxide stress or liaH and liaI (induced at 48 h, Supplementary Data 2), which confer resistance to cell wall antibiotics. Indeed, ~41% of the SigB-regulated genes are induced at 24 h (Supplementary Fig. 3B and Supplementary Data 1), and these genes are involved in multiple and different functions, including protease and chaperone activity, DNA repair or resistance against oxidative stress. At 72 h, ~10% of the genes of the SigB regulon were still upregulated, suggesting the existence of cellular stress during colony development (Fig. 2B and Supplementary Data 3). Furthermore, the activation of the SOS response points toward the existence of DNA damage in ΔtasA cells, another sign of stress, with induction of uvrA (at 24 and 72 h, Supplementary Data 1 and 3) and uvrB (at 24 h, Supplementary Data 1), both of which are involved in DNA repair. The presence of DNA damage in ΔtasA cells is further indicated by the induction of almost all of the genes belonging to the lysogenic bacteriophage PBSX at 72 h, a feature that has been reported to occur in response to mutations as well as to DNA or peptidoglycan damage[51,52] (Fig. 2B and Supplementary Data 3).

In general, the transcriptional changes observed in the ΔtasA cells illustrate an intrinsic major physiological change that

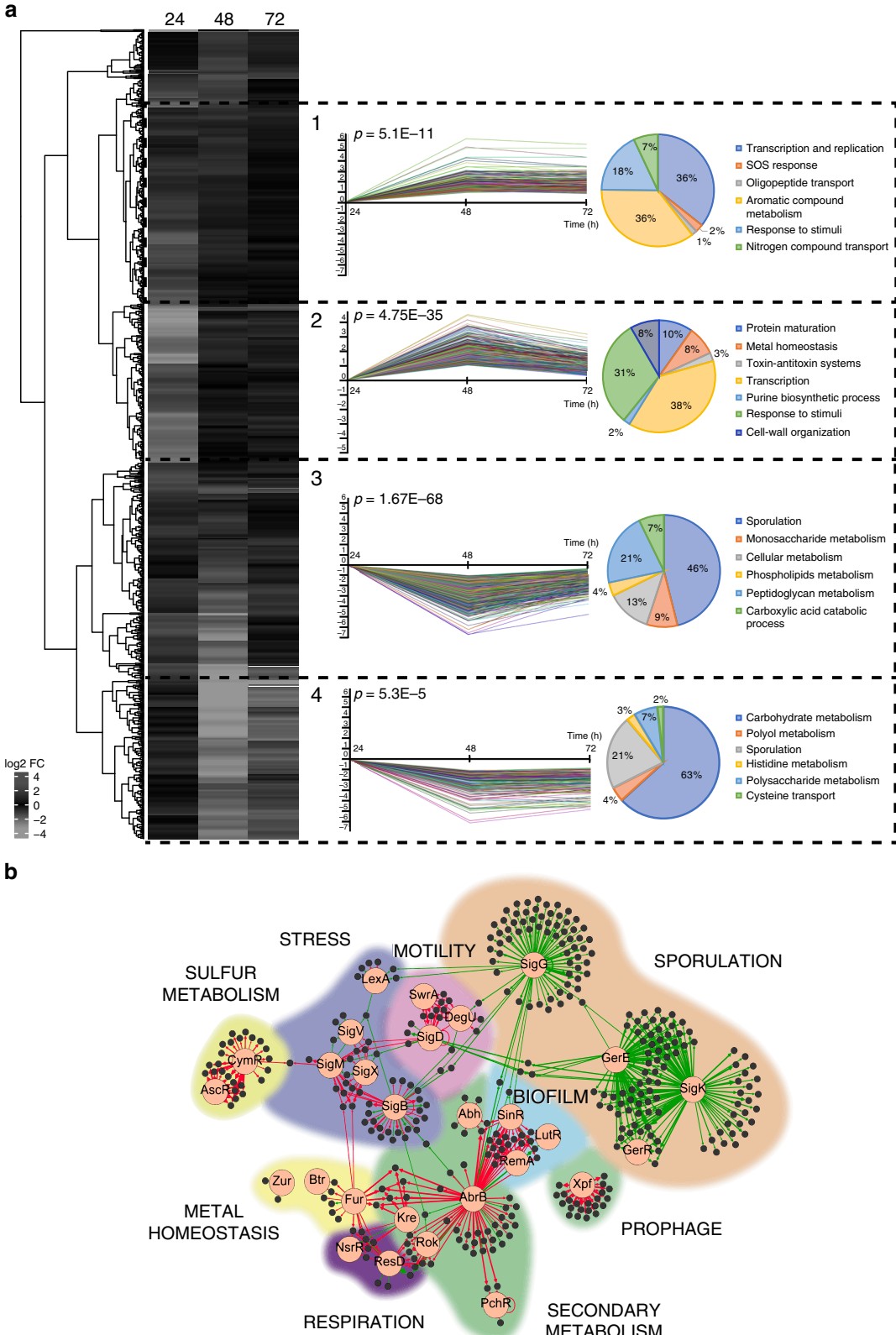

**Fig. 2 The *tasA* mutant displays major gene expression changes. a** Heatmap and gene profiles (1–4) of genes with similar expression patterns. Genes in the same profile are categorized (right) according to their gene ontology (GO) terms. Each gene profile represents statistically significant genes. Statistical significance was assessed by a $\chi^2$ test comparing the number of clustered genes with the expected theoretical number of genes if gene distribution were random. In the heatmap, induced genes are colored in dark gray and repressed genes in light gray. **b** Differentially expressed genes at 72 h clustered into different regulons. The bigger circles indicate de main regulator of that regulon, which is surrounded by arrows pointing to smaller circles that are the differentially expressed genes. The thickness of the arrows indicates expression levels. The color in the arrows indicates induction (red) or repression (green).

progresses over time and suggest the accumulation of excessive cellular stress. These changes result in the early entry of the cells into stationary phase, indicated by the state of the ΔtasA colony at 72 h compared to the WT (Fig. 2B) and supported by increased expression levels of genes related to: (i) biofilm formation (ii) synthesis of secondary metabolites (siderophores, antimicrobials, etc.); (iii) anaerobic respiration, fermentative metabolic pathways, and overflow metabolism; (iv) paralogous metabolism and assimilation of modified or toxic metabolic intermediates; (v) general stress and DNA damage; and (vi) induction of the lysogenic bacteriophage PBSX.

**ΔtasA cells exhibit impaired respiration and metabolic activity.** Our transcriptomic analysis suggested that ΔtasA cells exhibit a shift from aerobic respiration to fermentation and anaerobic respiration as well as activation of secondary metabolism, physiological features typical of stationary phase cells[38,53]. Based on the higher abundance of fengycin on leaves treated with ΔtasA cells and its key role in the antagonistic interaction between *B. subtilis* and fungal pathogens, we further investigated the kinetics of fengycin production in vitro. Flow cytometry analysis of cells expressing YFP under the control of the fengycin operon promoter demonstrated the induction of fengycin production in a subpopulation of cells (26.5%) at 48 h in the WT strain. However, more than half of the ΔtasA population (67.3%) actively expressed YFP from the fengycin operon promoter at this time point (Fig. 3A, top). At later stages of growth (72 h), the promoter was still active in the ΔtasA cells, and the population of positive cells was consistently higher than that in the WT strain (Fig. 3A, bottom). Mass spectrometry analysis of cell-free supernatants from WT or ΔtasA MOLP (a medium optimized for lipopeptide production) liquid cultures demonstrated that this expression level was sufficient for the *tasA* mutant cells to produce nearly an order of magnitude more fengycin (Fig. 3B, bottom spectrum) consistent with our findings in plants (Fig. 1E). Additionally, relatively higher levels of fengycin were detected in cells or agar fractions of ΔtasA colonies compared to WT colonies grown on solid MSgg, the medium used in all of our experimental settings (Supplementary Fig. 5, top and bottom spectra respectively). Similar results were obtained for the lipopeptide surfactin in these fractions (Fig. 3B, top spectrum, and Supplementary Fig. 5), consistent with our RNA-seq analysis (Supplementary Data 1–3). In agreement to these observations, in vitro experiments showed that the cell-free supernatants from ΔtasA cells exhibited antifungal activity against *P. xanthii* conidia equivalent to that of WT cells, even in highly diluted spent medium (Fig. 3C). These results confirm the robust antimicrobial potency of ΔtasA cells and imply that primary metabolic intermediates are diverted to different pathways to support the higher secondary metabolite production in the ΔtasA mutant cells.

Consistent with these findings, we observed two complementary results that indicate less efficient metabolic activity in ΔtasA cells compared to that in WT cells: first, the induction at 24 h and 48 h of the genes responsible for the synthesis of the anaerobic respiration machinery (Supplementary Data 1 and 2, and Supplementary Fig. 3B) mentioned above, and second, the differential expression at 72 h of the *nasD* and *nasF* genes (parts of the anaerobic respiration machinery) and the differential expression of genes at all times encoding several terminal oxidases found in the electron transport chain (Supplementary Data 1–3). The analysis of the respiration rates of these strains using the tetrazolium-derived dye 5-cyano-2,3-ditolyl tetrazolium chloride (CTC) and flow cytometry revealed a higher proportion of ΔtasA cells with lower respiration rates at 24 h and 72 h compared to the WT proportions (69.10% vs. 43.07% at 24 h and

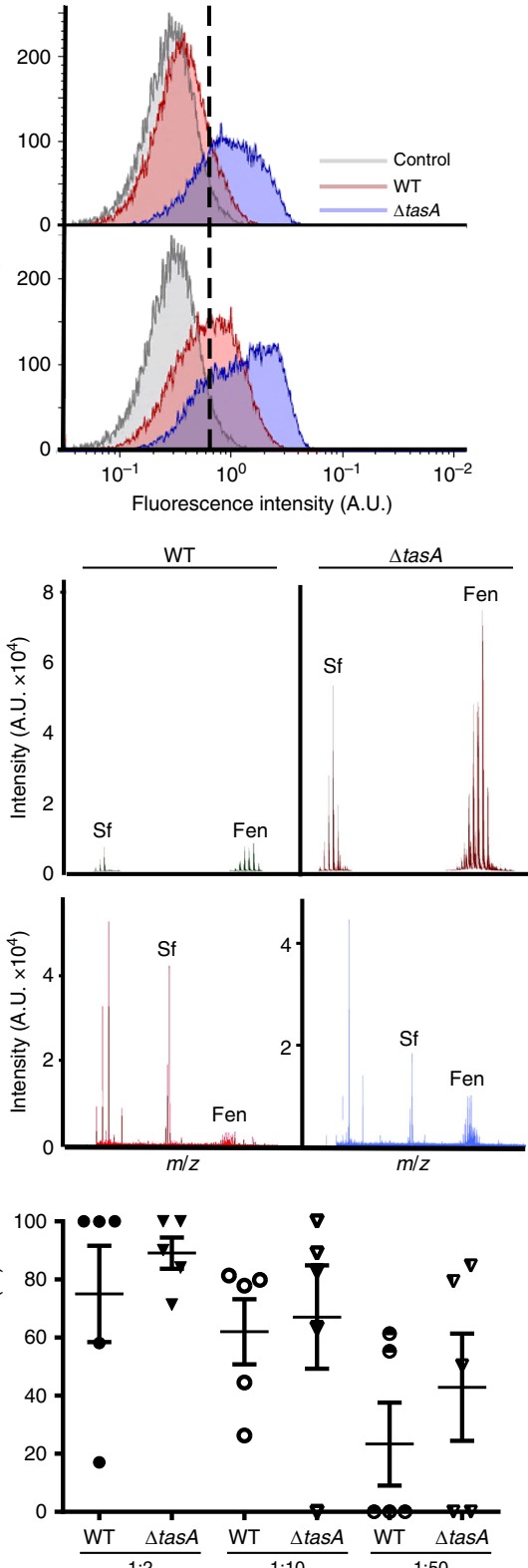

74.56 vs. 65.11% at 72 h, respectively) (Supplementary Table 1 and Fig. 4A). Second, the expression levels of the *alsSD* genes, which are responsible for the synthesis of acetoin (a metabolite produced by fermentative pathways) were higher in the ΔtasA strain than in the WT strain at all times (Supplementary Data 1–3). Indeed, all of the factors required for acetoin synthesis from

**Fig. 3 ΔtasA cells produce larger amounts of fengycin. a** Flow cytometry results of cells encoding the promoter of the fengycin production operon fused to YFP show that a higher percentage of ΔtasA cells (blue) expressed YFP compared with the percentage of YFP-expressing WT cells (red) at 48 h (top) and 72 h (bottom). A non-fluorescent negative control corresponding to the unlabeled WT strain at 72 h is shown (gray) in both experiments. **b** MALDI-TOF/TOF MS analysis of solid medium or spent MOLP medium after 72 h of growth showed higher fengycin levels in ΔtasA cultures (right) compared to that in WT cultures (left). **c** Serial dilutions of spent medium after 72 h of incubation deposited over infected leaf disks showed that the liquid medium from ΔtasA cultures retained as much antifungal activity as the medium from WT cultures. N = 5 independent experiments. In each experiment, three leaf disks were examined. Average values are shown. Error bars represent the SEM. Source data are provided as a Source Data file.

pyruvate were overexpressed at 72 h, whereas some key factors involved in the divergent or gluconeogenetic pathways were repressed (Supplementary Data 1–3 and Supplementary Fig. 6). Expression of *alsS* and *alsD* is induced by acetate, low pH and anaerobiosis[43,54,55]. Acetoin, in contrast to acetate, is a neutral metabolite produced to manage the intracellular pH and to ameliorate over-acidification caused by the accumulation of toxic concentrations of acetate or lactate, and its production is favored during bacterial growth under aerobic conditions[56]. Reduced respiration rates typically result in the accumulation of higher cellular proton concentrations, which leads to cytoplasmic acidification. These observations led us to postulate that the activation of the *alsSD* genes and the lower respiration rates observed in ΔtasA colonies might also reflect acidification of the intracellular environment, a potential cause of stationary phase-related stress. Measurements of the intracellular pH levels using the fluorescent probe 5-(6)carboxyfluorescein diacetate succini-midyl ester confirmed a significant decrease in the intracellular pH of nearly one unit ($-0.92 \pm 0.33$,) in ΔtasA cells at 72 h (Fig. 4B) compared to that in WT cells.

**Loss of TasA increases membrane fluidity and cell death.** The reduction in metabolic activity of ΔtasA cells, along with their acidification of the intracellular environment, might be expected to result in reduced bacterial viability. Measurements of the dynamics of viable bacterial cell density, expressed as CFU counts, showed that after 48 h, ΔtasA colonies possessed nearly an order of magnitude fewer CFUs than did WT colonies (Fig. 4C). These results suggest the hypothesis that ΔtasA colonies might exhibit higher rates of cell death than WT colonies. To test this possibility, we analyzed the live and dead sub-populations using the BacLight LIVE/DEAD viability stain and confocal microscopy (Fig. 4D right). The proportion of dead cells in ΔtasA colonies ranged from between 16.80% (16.80 ± 1.17) and 20.06% (20.06 ± 0.79) compared to 4.45% (4.45 ± 0.67) and 3.24% (3.24 ± 0.51) found in WT colonies at 48 h and 72 h, respectively (Fig. 4D, left). The significantly higher rate of cell death in ΔtasA compared to WT is consistent with the drastically lower bacterial counts found in the ΔtasA mutant colonies after 48 h. To rule out the influence of media composition on the observed phenotype, we performed the same experiments on solid LB medium, on which *B. subtilis* can still form a biofilm, as reflected by the wrinkly phenotype of the colonies (Supplementary Fig. 7A). We found that ΔtasA colonies exhibited a significantly higher proportion of dead cells at 48 h (17.86 ± 0.92) than did WT colonies (3.88 ± 0.33) (Supplementary Fig. 7B). Interestingly, the higher rate of cell death exhibited by the *tasA* mutant was not reproducible when both

strains were grown in liquid MSgg with shaking, conditions that promote planktonic growth. WT and ΔtasA cultures showed similar growth rates under these conditions (Supplementary Fig. 7C), and the proportion of cell death was measured in exponential (ΔtasA 0.32 ± 0.03 vs WT 0.78 ± 0.30) or stationary phase cultures (ΔtasA 2.31 ± 0.44 vs WT 0.56 ± 0.08) (Supplementary Fig. 7D), indicating that the lower viability of ΔtasA cells is observable when biofilms form on solid media.

The impaired respiration rates and the acidification of the cellular environment found in the ΔtasA cells are causes of cellular stress that can lead to ROS generation[57,58], a well-known trigger of stress-induced cell death[59]. To determine if ΔtasA cells possess abnormal ROS levels, we monitored ROS generation using hydroxyphenyl fluorescein (HPF), a fluorescent indicator of the presence of highly reactive hydroxyl radicals. Flow cytometry analysis revealed a larger proportion of HPF-positive cells (which have increased ROS levels) in the ΔtasA strain at 24 h compared to the WT proportion (42.38% vs. 28.61%, respectively) (Fig. 4A and Supplementary Table 1). To test whether this higher ROS production has negative effects on cellular components and functions, we first performed TUNEL assays to fluorescently stain bacterial cells containing DNA strand breaks, a known hallmark of the cell death induced by cellular damage and a frequent outcome of ROS production. At 24 h and 48 h, we found a significantly higher number of fluorescently stained ΔtasA cells compared with the number of fluorescently stained WT cells (Fig. 5A, left and 5B, left). These results indicated that DNA damage appears to occur not only earlier, but also with a higher frequency, in ΔtasA cells than in WT cells. A sizeable number of stained cells was also found at 72 h in the ΔtasA colonies, the same time-point at which the TUNEL signal started to increase in the WT colonies (Fig. 5A left). The TUNEL signal in the ΔtasA cells at this time-point was not significantly different from that of the WT cells (Fig. 5B left), probably due to the increased cell death in the ΔtasA cells.

Next, we examined the cellular membrane potential, another phenotype related to cell death, using the fluorescent indicator tetramethylrhodamine, methyl ester (TMRM). Consistent with all previous analysis, the alterations in the membrane potential of the ΔtasA cells were significantly different at all time points compared with the corresponding values for the WT cells (Fig. 6A left panel and 6B left). As a control, ΔtasA cells after 72 h of growth and treated with carbonyl cyanide m-chlorophenyl hydrazine (CCCP), a chemical ionophore that uncouples the proton gradient and can depolarize the membrane, showed a strong decrease in the fluorescence signal (Supplementary Figs. 8A, B). These results indicate that after 48 h (the same time point at which the cell death rate increases and the cell population plateaus in ΔtasA colonies) ΔtasA cells also exhibit increased membrane hyperpolarization compared with that in the WT cells, a feature that has been linked to mitochondrial-triggered cell death in eukaryotic cells[60–62].

The differences in ROS production, DNA damage level and membrane hyperpolarization between the WT and ΔtasA cell populations are consistent with increased cellular stress being the cause of the higher cell death rate observed in ΔtasA colonies after 24 h. To test the idea that loss of *tasA* results in increased cellular stress that leads to abnormal cellular physiology and increased cell death, we investigated the level of membrane lipid peroxidation, a chemical modification derived from oxidative stress that subsequently affects cell viability by inducing toxicity and apoptosis in eukaryotic cells[63,64]. Staining with BODIPY 581/591 C11, a fluorescent compound that is sensitive to the lipid oxidation state and localizes to the cell membrane, showed no significant detectable differences in the levels of lipid peroxidation at any time point (Supplementary Fig. 16). However, treatment

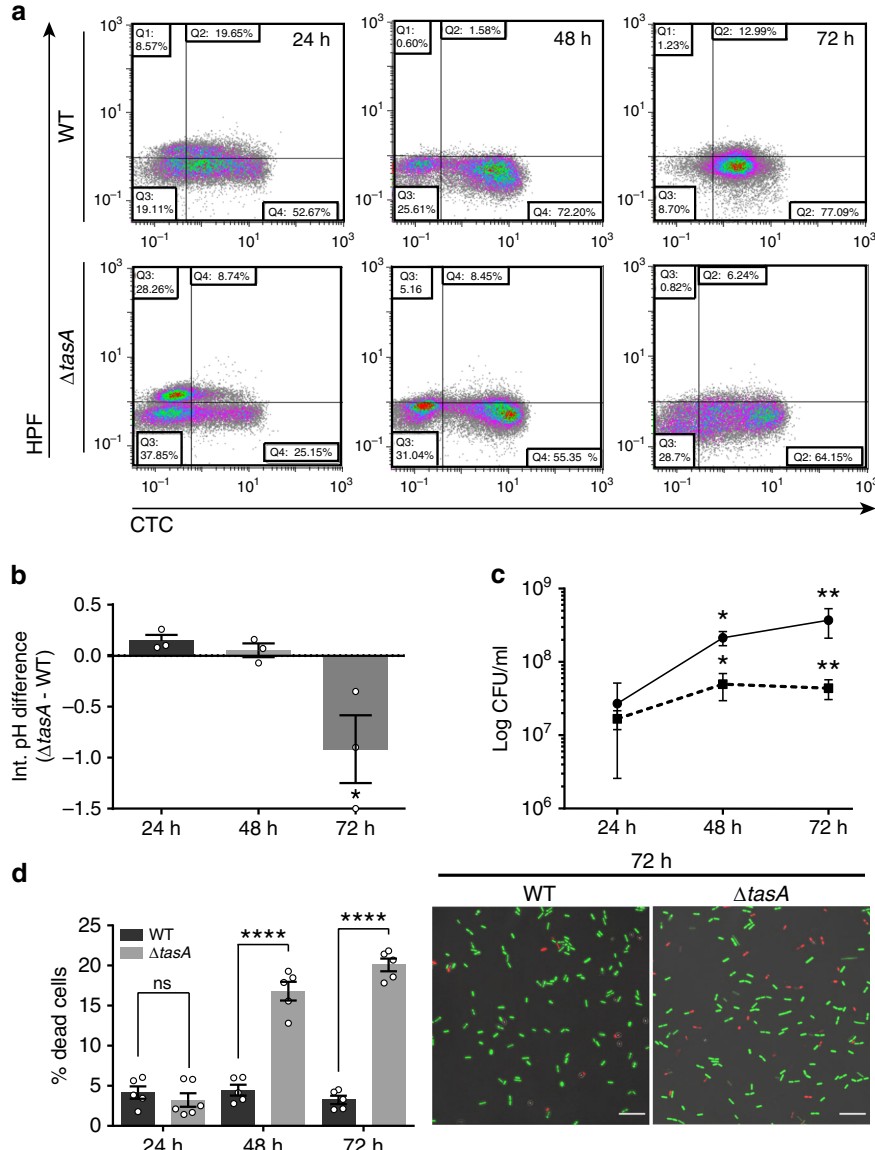

**Fig. 4 Respiration rates and cell viability are compromised in Δ*tasA* cells. a** Flow cytometry density plots of cells double stained with the HPF (Y axis) and CTC (X axis) dyes show that Δ*tasA* cells were metabolically less active (lower proportion of cells reducing CTC) and were under oxidative stress as early as 24 h (higher proportion of HPF-stained cells). **b** Measurements of intracellular pH show significant cytoplasmic acidification in the Δ*tasA* cells at 72 h. Average values of four biological replicates are shown. Error bars indicate the SEM. Statistical significance was assessed by one-way ANOVA with Tukey multiple comparison test (*$p < 0.05$). **c** The population dynamics in Δ*tasA* (dashed line) and WT colonies (solid line) grown on MSgg agar at 30 °C showed a difference of nearly one order of magnitude in the Δ*tasA* colony from 48 h. Average values of three biological replicates are shown. Error bars represent the SEM. Statistical significance was assessed by two-sided independent *t*-tests at each time point (**$p$ value = 0.0084 *$p$ value = 0.0410). **d** Left. Quantification of the proportion of dead cells treated with the BacLight LIVE/DEAD viability stain in WT and Δ*tasA* colonies at different time-points reveled a significantly higher population of dead cells at 48 h and 72 h in Δ*tasA* colonies compared to that found in the WT colonies. $N = 5$ colonies of the corresponding strains examined over three independent experiments. Average values are shown. Error bars represent the SEM. For each experiment and sample, at least three fields-of-view were measured. Statistical significance was assessed via two-tailed independent *t*-tests at each time-point (****$p < 0.0001$). Right. Representative confocal microscopy images of fields corresponding to LIVE/DEAD-stained WT or Δ*tasA* samples at 72 h. Scale bars = 10 μm. Source data are provided as a Source Data file.

with cumene hydroperoxide (CuHpx), a known inducer of lipid peroxide formation[65], resulted in different responses in the two strains. WT cells showed high reduced/oxidized ratios at 48 and 72 h and, thus, a low level of lipid peroxidation (Fig. 6A, center panel and Fig. 6B, center). In contrast, the comparatively lower reduced/oxidized ratios in Δ*tasA* cells at 48 and 72 h indicated increased lipid peroxidation. These results demonstrate that the Δ*tasA* strain is less tolerant to oxidative stress than is the WT strain, and, therefore, is more susceptible to ROS-induced

damage. This finding along with the increased ROS production in Δ*tasA* cells, led us to study the integrity and functionality of the plasma membrane. First, no clear differences in the integrity, shape, or thickness of the cell membrane or cell wall were observed via transmission electron microscopy (TEM) of negatively stained thin sections of embedded Δ*tasA* or WT cells at 24 h and 72 h under our experimental conditions (Supplementary Fig. 9). Next, we examined membrane fluidity, an important functional feature of biological membranes that affects their

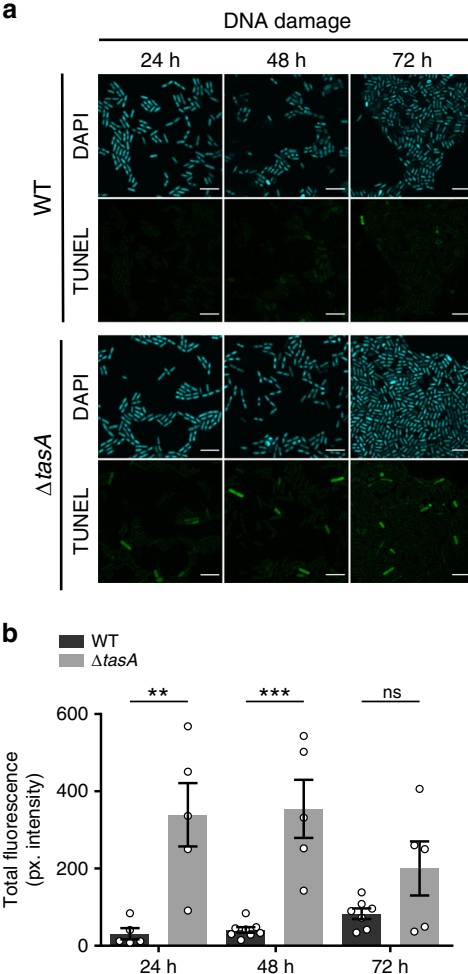

**Fig. 5 The ΔtasA cells exhibit higher levels of DNA damage. a** CLSM analysis of TUNEL assays revealed significant DNA damage in the ΔtasA cells (bottom panels) compared to that in the WT cells (top panels). Cells were counterstained with DAPI DNA stain (top images). Scale bars = 5 μm. Experiments have been repeated at least three times with similar results. **b** Quantification of the TUNEL signals in WT and ΔtasA colonies. The results showed significant differences in the DNA damage levels between ΔtasA and WT cells after 24 and 48 h of growth. For the WT strain, N = 5 at 24 h, N = 8 at 48 h and N = 7 at 72 h. For the ΔtasA strain N = 5 at all times. N refers to the number of colonies examined over three independent experiments. For each experiment and sample, at least three fields-of-view were measured. Error bars indicate the SEM. Statistical significance was assessed via two-tailed independent t-tests at each time-point (**p value = 0.061 ***p value = 0.002). Source data are provided as a Source Data file.

permeability and binding of membrane-associated proteins, by measuring the Laurdan generalized polarization (Laurdan GP) [63,66]. Our results show that the Laurdan GP values were significantly lower at 48 h and 72 h in ΔtasA cells compared with the values in WT cells (0.65 ± 0.03 or 0.82 ± 0.03 respectively, at 48 h, and 0.87 ± 0.006 or 0.73 ± 0.007 respectively, at 72 h) (Fig. 6A, right panel and Fig. 6B, right). These results indicate incremental changes in membrane fluidity, comparable to that resulting from treatment of cells with benzyl alcohol, a known membrane fluidifier (Supplementary Fig. 10A, top and center panels, B). Membrane fluidity has been associated with higher ion, small molecule, and proton permeability [67,68], which might contribute to the higher concentration of fengycin found in the in cell-free supernatants of ΔtasA cultures (Fig. 3B). These

effects could also explain why ΔtasA cells are impaired in energy homeostasis as well as the subsequent effects on the intracellular pH and membrane potential that eventually contribute to cell death.

**TasA associates to detergent-resistant fractions of the membrane**. The negative effects on membrane potential and fluidity observed in the ΔtasA cells suggest alterations in membrane dynamics, which in bacterial cells are directly related to functional membrane microdomains (FMM); FMMs are specialized membrane domains that also regulate multiple important cellular functions [69–72]. The bacterial flotillins FloT and FloA are localized in FMMs and are directly involved in the regulation of membrane fluidity [69]. This line of evidence led us to propose a connection between the membrane fluidity and permeability of ΔtasA cells and changes in the FMMs. We initially studied the membrane distribution of FloT as a marker for FMMs in WT and ΔtasA cells using a FloT-YFP translational fusion construct and confocal microscopy (Fig. 7A). The WT strain showed the typical FloT distribution pattern, in which the protein is located within the bacterial membrane in the form of discrete foci[73] (Fig. 7A, top). However, in the ΔtasA cells, the fluorescent signal was visible only in a subset of the population, and the normal distribution pattern was completely lost (Fig. 7A bottom). In agreement with these findings, quantification of the fluorescent signal in WT and ΔtasA samples showed significant decreases in the signal in the ΔtasA mutant cells at 48 and 72 h (Fig. 7B). Consistently, our RNA-seq data showed fluctuations in the floT expression levels at all times (Supplementary Data 1–3).

The alterations in floT expression and the loss of the normal FloT distribution pattern in the cell membrane that occurs in the ΔtasA mutant cells led us to consider the presence of TasA in FMMs. Membranes from both prokaryotes and eukaryotes can be separated into detergent-resistant (DRM) and detergent-sensitive fractions (DSM) based on their solubility in detergent solutions[73]. Although it is important to point out that the DRM and FMMs (or lipid rafts in eukaryotes) are not equivalent, the DRM fraction has a differential lipid composition and is enriched with proteins, rendering it more resistant to detergents; furthermore, many of the proteins present in FMMs are also present in the DRM[74]. Immunodetection assays of the DRM, DSM, and cytosolic fractions of each strain using an anti-TasA antibody showed the presence of anti-TasA reactive bands of the expected size primarily in the DRM fraction and in the cytosol (Fig. 7C, top). As controls, the fractions from the tasA mutant showed no signal (Fig. 7C, top). Western blots of the same fractions isolated from WT and ΔtasA strains carrying a FloT-YFP translational fusion with an anti-YFP antibody (Fig. 7C, bottom) confirmed that FloT was mainly present in the DRM of WT cells (Fig. 7C, bottom, lane 1). The signal was barely noticeable in the same fraction from ΔtasA cells (Fig. 7C, bottom, lane 4), mirroring the reduced fluorescence levels observed via microscopy (Fig. 7A), and consistent with the RNA-seq data. These results confirm that TasA is indeed associated with the DRM fraction of the cell membrane. Furthermore, we asked whether the loss of FloT foci is somehow related to the increased cell death observed in the absence of TasA. We used a LIVE/DEAD viability stain in a ΔfloT colony and in a ΔfloTfloA colony, a double mutant for the two flotillin-like proteins in the B. subtilis genome. The results show no significant differences in the proportion of cell death compared to the WT strain at 48 h and 72 h (Supplementary Fig. 11). These experiments demonstrate that the increased cell death is not caused by loss of the FloT distribution pattern that occurs in the tasA mutant.

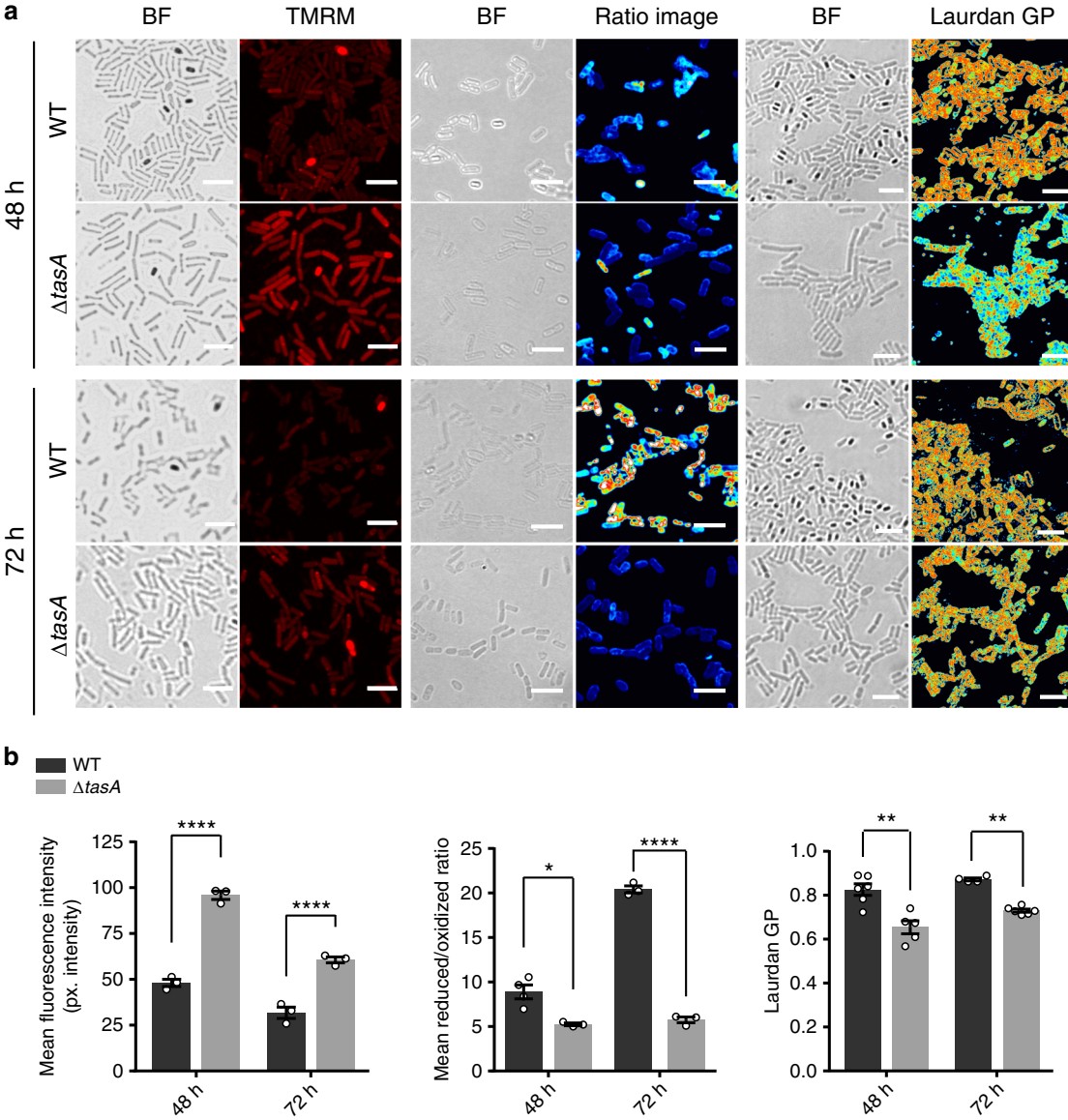

**Fig. 6 Δ*tasA* cell membrane exhibit cytological anomalies. a** Left panel. A TMRM assay of WT and Δ*tasA* cells, located at the top or bottom respectively in each set, showed a decrease in membrane potential in the WT cells, whereas the Δ*tasA* cells exhibited hyperpolarization at 48 and 72 h. Center panel. Assessment of the lipid peroxidation levels using BODIPY 581/591 C11 reagent in WT and Δ*tasA* cells after treatment with 5 mM CuHpx and analysis by CLSM. The ratio images represent the ratio between the two states of the lipid peroxidation sensor: reduced channel$_{(590-613 \, nm \, emission)}$/oxidized channel$_{(509-561 \, nm \, emission)}$. The ratio images were pseudo-colored depending on the reduced/oxidized ratio values. A calibration bar (from 0 to 50) is located at the bottom of the panel. Confocal microscopy images show that CuHpx treatment was ineffective in the WT strain at 72 h, whereas the mutant strain showed symptoms of lipid peroxidation. Right panel. Laurdan GP analyzed via fluorescence microscopy. The images were taken at two different emission wavelengths (gel phase, 432–482 nm and liquid phase, 509–547 nm) that correspond to the two possible states of the Laurdan reagent depending on the lipid environment. The Laurdan GP images represent the Laurdan GP value of each pixel (see Methods section). The Laurdan GP images were pseudo-colored depending on the laurdan GP values. A calibration bar (from 0 to 1) is located at the bottom of the set. The Laurdan GP images show an increase in membrane fluidity (lower Laurdan GP values) in the *tasA* mutant cells at 48 and 72 h. All scale bars are equal to 5 µm. **b** Left. Quantification of the TMRM signal. $N = 3$. Center. Quantification of lipid peroxidation. $N = 3$. For the WT at 48 h, $N = 4$. Right. Quantification of laurdan GP values. For the WT strain, $N = 6$ at 48 h and $N = 4$ at 72 h. For the Δ*tasA* strain, $N = 5$ at all times. $N$ refers to the number of colonies examined over three independent experiments. Average values are shown. Error bars represent the SEM. For each experiment and sample, at least three fields-of-view were measured. Statistical significance in the TMRM experiments was assessed via two-tailed independent *t*-tests at each time-point (****$p < 0.0001$). Statistical significance in the lipid peroxidation experiments was assessed via two-tailed independent *t*-tests at each time-point (****$p < 0.0001$, *$p$ value $= 0.0115$). Statistical significance in the laurdan GP experiments was assessed via two-tailed independent Mann–Whitney tests at each time-point (**$p$ value $= 0.0087$ at 48 h and **$p$ value $= 0.0095$ at 72 h respectively). Source data are provided as a Source Data file.

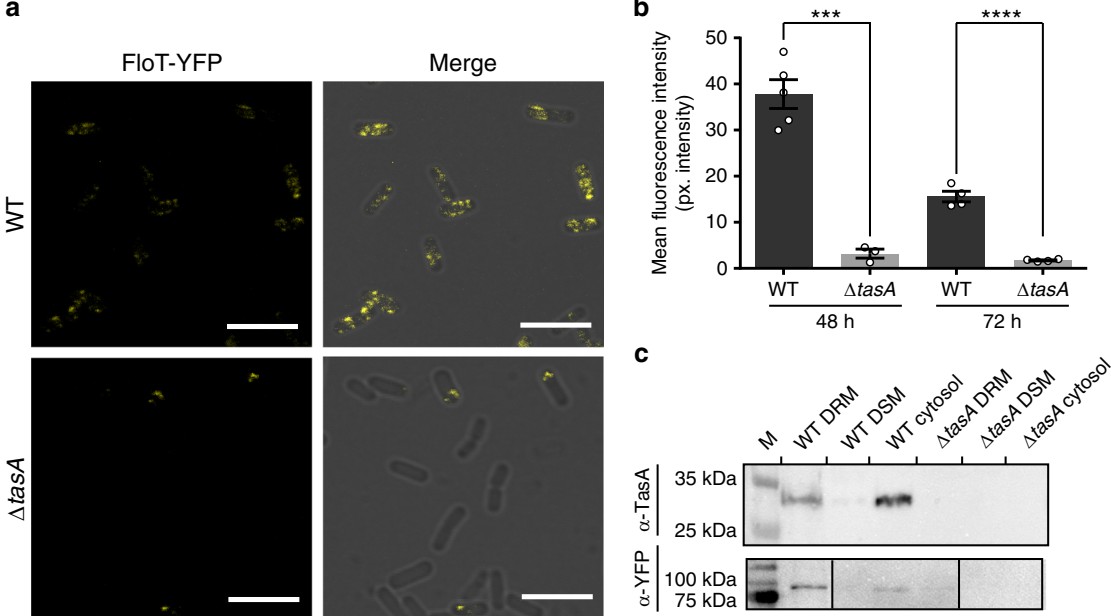

**Fig. 7 TasA is located in the DRM fraction of the cell membrane. a** Representative confocal microscopy images showing WT or Δ*tasA* cells expressing the *floT-yfp* construct at 72 hours. WT images show the typical punctate pattern associated to FloT. That pattern is lost in Δ*tasA* cells. **b** Quantification of fluorescence signal in WT (*N* = 5 at 48 h and *N* = 4 at 72 h) and Δ*tasA* (*N* = 3 at 48 h and *N* = 4 at 72 h) samples. *N* indicates the number of colonies examined over three independent experiments. Average values are shown. Error bars represent the SEM. Statistical significance was assessed via two-tailed independent *t*-tests at each time-point (***p value = 0.0002 ****p < 0.0001). For each experiment and sample, at least three fields-of-view were measured. **c** Western blot of different membrane fractions exposed to an anti-TasA or anti-YFP antibodies. Both antibodies have been used with the same set of samples in two independent immunoblots. Experiments have been repeated at least three times with similar results. Immunoblot images have been cropped (top image) or cropped and spliced (bottom image) for illustrative purposes. Black lines over the blot images delineate boundaries of immunoblot splicing. The three slices shown slices are derived from a single blot. Raw images of the blots presented in this figure can be found in the source data. Source data are provided as a Source Data file.

All together, these results allow us to conclude that TasA is located in the DRM fraction of the cell membrane where it contributes to membrane stability and fluidity, and that its absence leads to alterations in membrane dynamics and functionality, eventually leading to cell death.

**Mature TasA is required to maintain viable bacterial physiology.** TasA is a secreted protein located in the ECM and additionally found associated to the DRM fraction of the cell membrane (Fig. 7C). Reaching these sites requires the aid of secretion-dedicated chaperones, the translocase machinery and the membrane-bound signal peptidase SipW[75]. It is known that TasA processing is required for assembly of the amyloid fibrils and biofilm formation[18,76]. However, formation of the mature amyloid fibril requires the accessory protein TapA, which is also secreted via the same pathway[19], is present in the mature amyloid fibers and is found on the cell surface[76]. Considering these points, we first wondered whether TapA is involved in the increased cell death observed in the Δ*tasA* mutant. By applying the BacLight LIVE/DEAD viability stain to a Δ*tapA* colony, we found a similar proportion of live to dead cells as that found in the WT colony at 72 h (Fig. 8A), suggesting that the *tapA* mutant does not exhibit the cytological alterations and cellular damage that occurs in Δ*tasA* cells. Δ*tapA* cells produce a much lower number of TasA fibers but still expose TasA in their surfaces[76]; thus we reasoned that mature TasA is necessary for preserving the cell viability levels observed in the WT strain. To test this possibility, we constructed a strain bearing a mutation in the part of the *tasA* gene that encodes the TasA signal peptide[77]. To avoid confounding effects due to expression of the mutated *tasA* gene in the presence of the endogenous operon, we performed this analysis in

a strain in which the entire *tapA* operon was deleted and in which the modified operon encoding the mutated *tasA* allele was inserted into the neutral *lacA* locus. The strain carrying this construct was designated as "TasA SiPmutant" (for Signal Peptide mutant) and included three amino acid substitutions from the initial lysines of the signal peptide. Specifically, the introduced mutations were Lys4Ala, Lys5Ala, and Lys6Ala. The endogenous version of TasA successfully restored biofilm formation (Supplementary Fig. 12A), while the phenotype of SiP mutant on MSgg medium at 72 h was different from those of both the WT and *tasA* mutant strains (Fig. 8B and Supplementary Fig. 2B). Immunodetection analysis of TasA in fractionated biofilms confirmed the presence of TasA in the cells and ECM fractions from the WT strain and the strain expressing the endogenous version of *tasA* (Fig. 8C). However, a faint anti-TasA reactive signal was observed in both fractions of the SiP mutant (Fig. 8C). This result indicates that TasA is not efficiently processed in the SiP mutant and, thus, the protein levels in the ECM were drastically lower. The faint signal detected in the cell fraction might be due to the fact that the pre-processed protein is unstable in the cytoplasm and is eventually degraded over time[77]. Consistent with our hypothesis, the levels of cell death in the SiP mutant were significantly different from those of the WT strain (Fig. 8A, D). Taken together, these results rest relevance to TapA to the increase cell death observed in the absence of TasA and indicate that TasA must be processed to preserve the level of cell viability found in WT colonies.

**A TasA variant restores cellular physiological status but not biofilm.** The fact that the Δ*tapA* strain forms altered and fewer TasA fibers but does have normal cell death rates, as well as the

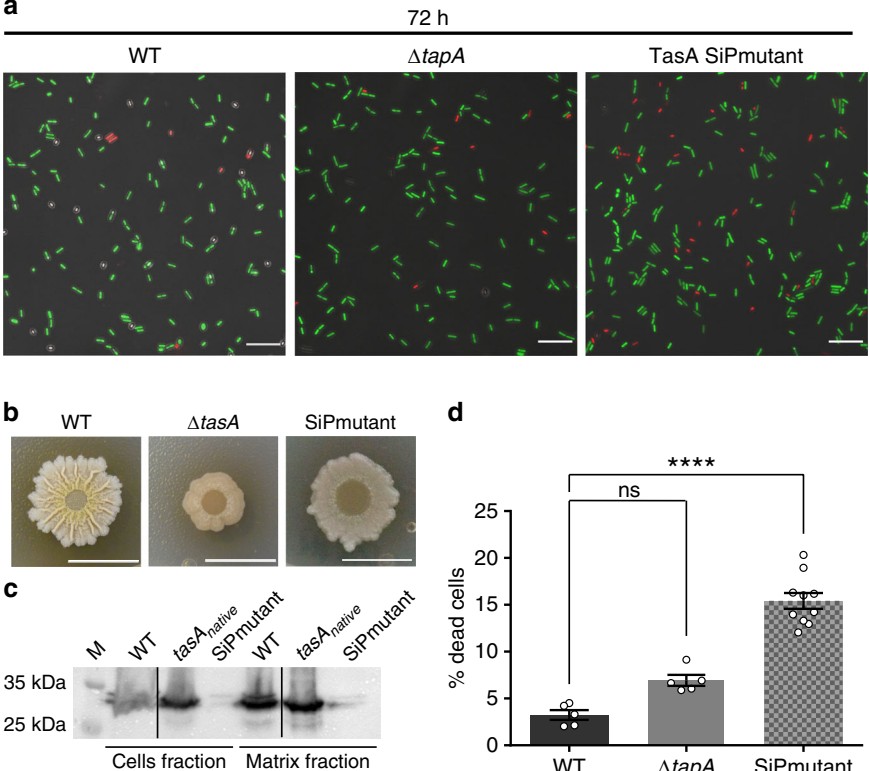

**Fig. 8 Mature TasA is required to stabilize cell viability within the colony. a** Representative confocal microscopy images of fields corresponding to LIVE/DEAD-stained WT, ΔtapA or signal peptide mutant (SiPmutant, Lys4Ala, Lys5Ala, Lys6Ala) cells at 72 h. Scale bars = 10 μm. **b** Colony phenotypes of WT, ΔtasA and the SiPmutant strains on MSgg agar at 72 h. Scale bars = 1 cm. **c** Western blot of the cell and matrix fractions of the three strains at 72 h exposed to an anti-TasA antibody. Experiments have been repeated at least three times with similar results. Immunoblot images have been cropped and spliced for illustrative purposes. Black lines over the blot images delineate boundaries of immunoblot splicing. The three slices shown are derived from a single blot. Raw images of the blots presented in this figure can be found in the source data file available with this manuscript. **d** Quantification of the proportion of dead cells in WT (N = 5), ΔtapA (N = 5), or SiPmutant (N = 10) colonies at 72 h. N refers to the number of colonies examined over three independent experiments. The WT data in Fig. 4 is from the same experiment as the data displayed in this figure and has been used as a control for the comparison between the WT, ΔtapA and SiPmutant colonies. Error bars represent the SEM. For each experiment and sample, at least three fields-of-view were measured. Statistical significance was assessed via two-tailed independent t-tests at each time-point (****p < 0.0001). Source data are provided as a Source Data file.

increased membrane fluidity and the changes in expression and loss of the normal distribution pattern of the flotillin-like protein FloT in the ΔtasA strain led us to hypothesize that the TasA in the DRM, and not that in the ECM, is responsible for maintaining the normal viability levels within the WT colonies. To explore this hypothesis, we performed an alanine scanning experiment with TasA to obtain an allele encoding a stable version of the protein that could support biofilm formation. To produce these constructs, we used the same genetic background described in the above section. The strain JC81, which expresses the TasA (Lys68Ala, Asp69Ala) variant protein, failed to fully restore the WT biofilm formation phenotype (Fig. 9A, Supplementary Figs. 2B and 12A). Immunodetection analysis of TasA in fractionated biofilms confirmed the presence of the mutated protein in the cells and in the ECM (Fig. 9B, left and Supplementary Fig. 12B). Tandem mass spectrometry analysis revealed that the mutated protein found in the ECM corresponded to the mature form of TasA (Supplementary Fig. 13A, left and Supplementary Fig. 13A, right), indicating exclusively a malfunction in the protein's structural role in proper ECM assembly. Electron microscopy coupled to immunodetection with anti-TasA and immunogold-conjugated secondary antibodies showed the presence of a dense mass of extracellular material in JC81 cells with

an absence of well-defined TasA fibers, as opposed to WT cells, in which we also observed a higher number of gold particles, indicative of the higher reactivity of the sample (Supplementary Fig. 13B, left and center panels). The cell membrane fractionation analysis revealed, however, the presence of mutated TasA in the DRM, DSM and cytosolic fractions (Fig. 9B right). Accordingly, JC81 was reverted to a physiological status comparable to that of the WT strain. This feature was demonstrated by similar expression levels of genes encoding factors involved in the production of secondary metabolites (i.e., ppsD, albE, bacB, and srfAA) or acetoin (alsS), indicating comparable metabolic activities between the two strains (Fig. 9C). Further evidence confirmed the restoration of the metabolic status in JC81. First, similar proportions of WT and JC81 cells expressing YFP from the fengycin operon promoter were detected after 72 h of growth via flow cytometry analysis (Fig. 9D, green curve). In agreement with these findings, there were no differences in the proportions of cells respiring or accumulating ROS or in the intracellular pH values between the JC81 and WT strains (Figs. 9E and 10A). Consistently, the population dynamics of JC81 resembled that of the WT strain (Fig. 10B), and, as expected, its level of cell death was comparable to that of the WT strain (Fig. 10C). Finally, there were no differences in any of the examined parameters related to

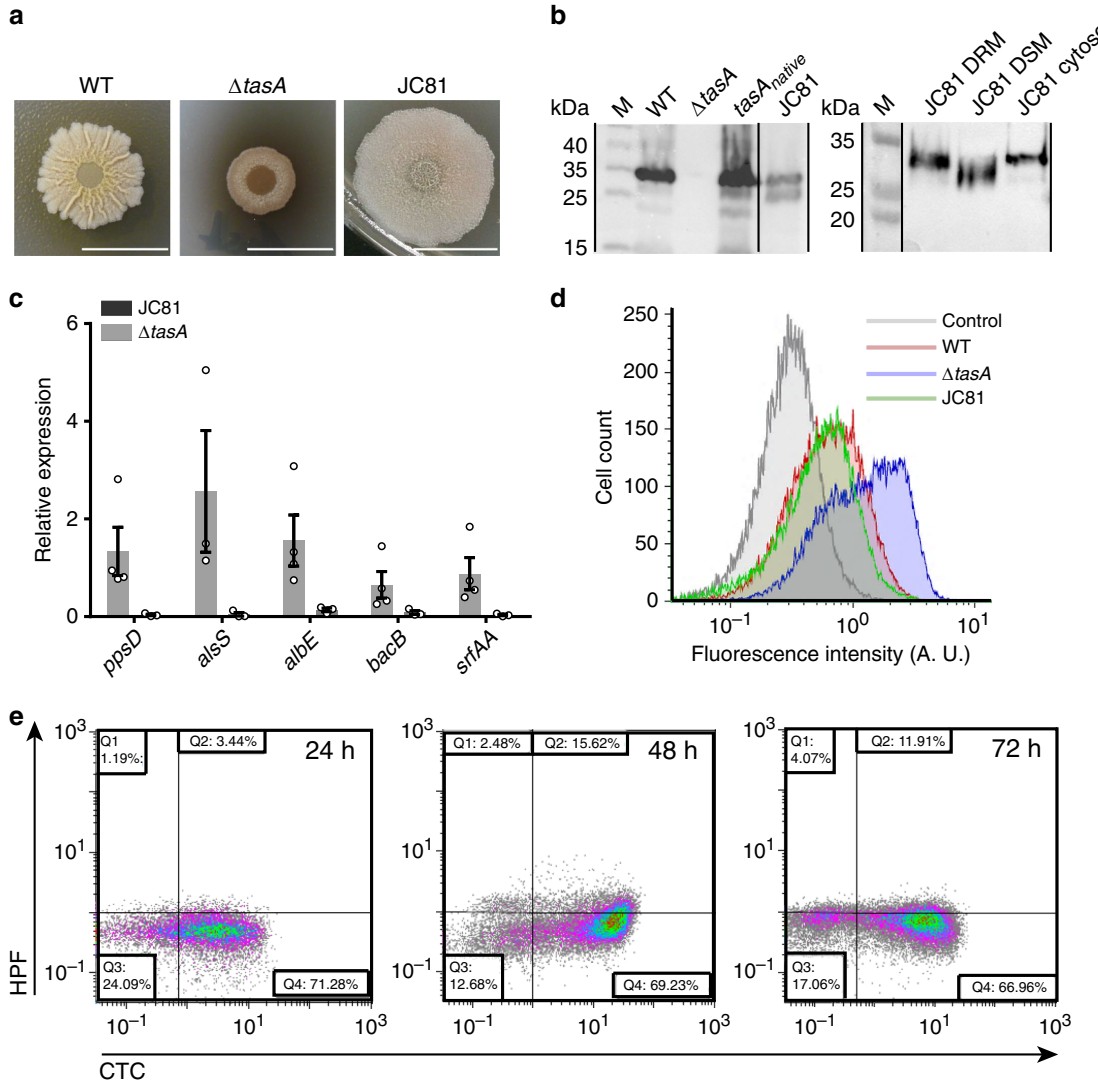

**Fig. 9 A TasA variant rescues cellular physiological status but not biofilm. a** Colony phenotypes of the three strains on MSgg agar at 72 h. Scale bars = 1 cm. **b** A western blot of the cell (left) and membrane fractions (right) at 72 h exposed to an anti-TasA antibody. Experiments have been repeated at least three times with similar results. Immunoblot images have been cropped and spliced for illustrative purposes. Black lines over the blot images delineate boundaries of immunoblot splicing. Two independents immunoblot images are shown. The two slices shown in the left image are derived from a single blot. The two slices shown in the right image are derived from a single blot. Raw images of the blots presented in this figure can be found in the source data file available with this manuscript. **c** Relative expression levels of *ppsD*, *alsS*, *albE*, *bacB*, and *srfAA* genes in JC81 compared to the WT strain. Average values of at least three biological replicates (*N* = 3 except for *ppsD*, *albE*, *bacB*, and *srfAA* in Δ*tasA*, where *N* = 4) are shown with error bars representing the SEM. **d** Flow cytometry analysis of cells expressing the promoter of the fengyncin production operon in the WT, Δ*tasA*, JC81 strains at 72 h. A non-fluorescent negative control corresponding to the unlabeled WT strain at 72 h is shown (gray). The flow cytometry data shown in Fig. 3 is from the same experiment as the data shown in this figure and are repeated here for comparative purposes with the data from strain JC81. **e** Density plots of cells double stained with the HPF (Y axis) and CTC (X axis) dyes show that JC81 behaved similarly to the WT strain.

oxidative damage and stress-induced cell death (i.e., DNA damage, membrane potential, susceptibility to lipid peroxidation, and membrane fluidity) between JC81 and WT cells (Supplementary Figs. 14 and 15 respectively), and the mutated allele complemented the sporulation defect observed in the ECM mutants (Supplementary Fig. 4). To further confirm these results, we performed a viability assay in a mixed Δ*tasA* and Δ*eps* colony co-inoculated at a 1:1 ratio, and we found that, despite the ability of the mixed colony to rescue the wrinkly phenotype typical of a WT colony (Supplementary Fig. 17A, top), the proportion of cell death is significantly higher than that observed in WT cells at 48 h (10.39 ± 1.20) and 72 h (14.04 ± 0.72) (Supplementary Fig. 17B). In addition, exogenous TasA did not revert the colony

morphology phenotype of Δ*tasA* cells on solid MSgg (Supplementary Fig. 18A) or the increased cell death rate observed in the Δ*tasA* strain (Supplementary Fig. 18B). These results show that the extracellular TasA provided by the Δ*eps* strain is sufficient to complement the ECM assembly and biofilm formation defects but not to prevent cell death, similar to the effects of exogenous TasA supplementation. Thus, TasA must be produced by the cells to reach the cell membrane and exert this function. Interestingly, a Δ*sinI* strain, which is mutant for the *sinI* anti-repressor that inhibits *sinR*, and therefore, has strong repression of ECM genes and is unable to assemble biofilms (Supplementary Fig. 17A, bottom), showed similar levels of cell death as the WT strain at all times (1.53 ± 0.25 at 48 h and 2.51 ± 0.50 at 72 h) (Supplementary

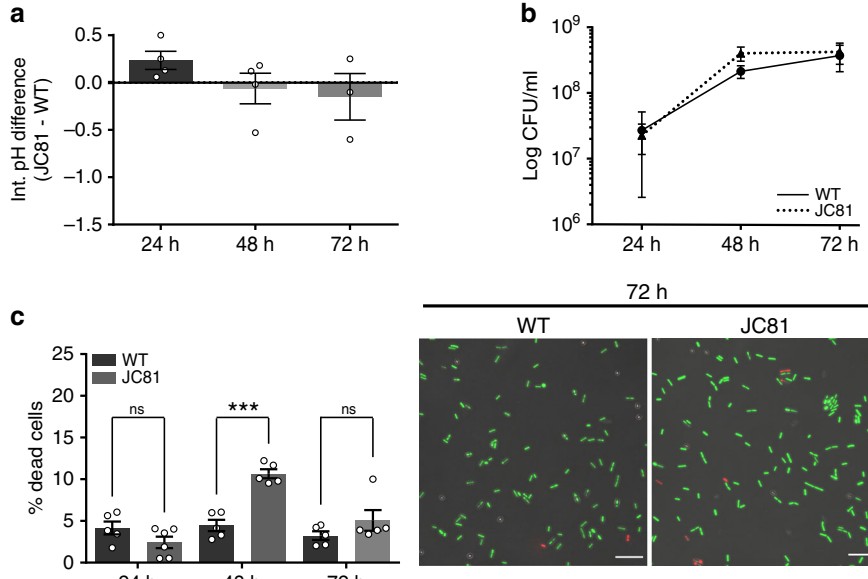

**Fig. 10 The expression of the TasA variant prevents the increase of cell death. a** Intracellular pH measurements of the WT and JC81 strains. Average values of four biological replicates are shown. Error bars represent the SEM. **b** Population dynamics (CFU counts) in WT and JC81 colonies. The WT data in Fig. 4C is from the same experiment as the data shown in this figure and is included here as a control in the comparison between the WT and JC81 strains. Average values of four biological replicates are shown. Error bars represent the SEM. **c** Left. Quantification of the proportion of dead cells in WT and JC81 colonies. $N = 5$ colonies of the corresponding strains examined over three independent experiments. Average values are shown. Error bars represent the SEM. For each experiment and sample, at least three fields-of-view were measured. Statistical significance was assessed via two-tailed independent $t$-tests at each time-point (***$p$ value $= 0.0003$). The WT data in Fig. 4 is from the same experiment as the data displayed in this figure and has been used as a control for the comparison between the WT and JC81 colonies. Right. Representative confocal microscopy images of fields corresponding to LIVE/DEAD-stained WT or JC81 cells at 72 hours. Scale bars $= 10 \mu m$. Source data are provided as a Source Data file.

Fig. 17B). This effect might reflect that even a basal amount of TasA[78] in the cell membrane is sufficient to prevent cell death but insufficient to assemble a proper ECM, confirming that indeed, cells lacking a structured ECM do not exhibit the physiological changes observed in cells lacking TasA.

Taken together, these findings assign TasA complementary functions, via its localization to the DRM fraction of the cell membrane, contributing to cell membrane dynamics and cellular physiology during normal colony development that prevent premature cell death, a role beyond the well-known structural function of amyloid proteins in biofilm ECMs.

**The TasA variant impairs B. subtilis fitness on the phylloplane.** Our analysis of the intrinsic physiological changes in ΔtasA cells showed how the absence of TasA leads to the accumulation of canonical signs of cellular damage and stress-induced cell death, a physiological condition typical of stationary phase cells. These observations help to reconcile two a priori contradictory features of *B. subtilis* ecology on plant leaves: the reduced persistence of the ΔtasA mutant on the melon phylloplane versus its ability to efficiently exert biocontrol against the fungus *P. xanthii*, which occurs via overproduction of fengycin and other antimicrobial molecules. Following this line of thought, we predicted that the JC81 strain, which expresses a version of TasA that is unable to restore biofilm formation but preserves the physiological status of the cells, would show overall signs of reduced fitness on melon leaves. The JC81 cells retained their initial ability to adhere to melon leaves (Fig. 1A); however, their persistence decreased (Fig. 1B) and their colonization showed a pattern somewhat intermediate between those of the WT and ΔtasA strains (Fig. 1C, bottom images). In agreement with our prediction, the reduced fitness of this strain resulted in a failure to manage *P. xanthii* infection (Fig. 1D). Thus, we conclude that the ECM, by means of

the amyloid protein TasA, is required for normal colonization and persistence of *B. subtilis* on the phyllosphere. These ecological features depend on at least two complementary roles of TasA: one role related to ECM assembly and a new proposed role in the preservation of the physiological status of cells via stabilization of membrane dynamics and the prevention of premature cell death.

## Discussion

The ECM provides cells with a myriad of advantages, such as robustness, colony cohesiveness, and protection against external environmental stressors[7,23,24]. Studies of *B. subtilis* biofilms have revealed that the ECM is mainly composed of polysaccharides[14] and the proteins TasA and BslA[7,15]. TasA is a confirmed functional amyloid that provides structural support to the biofilm in the form of robust fibers[16]. A recent study demonstrated that there is heterogeneity in the secondary structure of TasA; however, in biofilms, its predominant conformation is in the form of stable fibers enriched in β-sheets[17]. In this study, we demonstrate that in addition to its structural role in ECM assembly, TasA is also required for normal colony development – both of which are functions that contribute to the full fitness of *Bacillus* cells on the phylloplane.

The physiological alterations observed in ΔtasA null strain reflect a process of progressive cellular deterioration characteristic of senescence[79–81], including early activation of secondary metabolism, low energy metabolic activity, and accumulation of damaged molecular machinery that is required for vital functions. Indeed, it has been previously demonstrated that such metabolic changes can trigger cell death in other bacterial species, in which over-acidification of the cytoplasm eventually leads to the activation of cell death pathways[55]. Interestingly, cytoplasmic acidification due to the production of acetic acid has been linked to

higher ROS generation and accelerated aging in eukaryotes[82]. As mentioned throughout this study, ROS generation leads to ongoing DNA damage accumulation, phospholipid oxidation, and changes in cell membrane potential and functionality, all of which are major physiological changes that eventually lead to declines in cellular fitness and, ultimately, to cell death[83–85]. The fact that we could restore the physiological status of tasA null mutant cells by ectopically expressing a mutated TasA protein incapable of rescuing biofilm formation permitted us to separate two roles of TasA: (i) its structural function, necessary for ECM assembly; and (ii) its cytological functions involved in regulating membrane dynamics and stability and preventing premature cell death. Our data indicate that this previously unreported function does not involve TasA amyloid fibers or its role in ECM assembly, and that it is more likely related to the TasA found in the DRM of the cell membrane, where the FMMs (like the lipid rafts of eukaryotic cells) are located. It is not unprecedented that amyloid proteins interact with functional domains within the cell membrane. In eukaryotic cells, for instance, it has been reported that lipid rafts participate in the interaction between the amyloid precursor protein and the secretase required for the production of the amyloid-β peptide, which is responsible for Alzheimer's disease[86]. Indeed, our results are supported by evidence that TasA can preferentially interact with model bacterial membranes, which affects fiber assembly[87], and that TasA fibers are located and attached to the cell surface via a proposed interaction with TapA, which forms foci that seem to be present on the cell wall[76]. Interestingly, TapA has been recently characterized as a two-domain, partially disordered protein[88]. Disordered domains can be flexible enough to interact with multiple partners[89,90], suggesting a similar mechanism for TapA: the N-terminal domain might be involved in the interaction with other protein partners, whereas the C-terminal disordered domain might anchor the protein to the cell surface. All of these observations led us to propose that TasA may drive the stabilization of the FMMs in the cell membrane either directly via interactions with certain phospholipids or indirectly via interactions with other proteins. This model is further supported by the fact that the ΔtasA cells show alteration of FloT levels and loss of the typical FloT distribution pattern (Fig. 7A), which is typically present in the FMM, and induction of many genes that encode DRM components or other factors that interact with FloT alone or with FloT and FloA (Supplementary Data 1–3). Considering these novel findings, it is tempting to speculate that cells can regulate membrane dynamics by, among other processes, tuning the amount of TasA present in the membrane at a given moment, which would permit a better physiological response to different environmental cues. In particular, our results suggest that the membrane instability caused by the absence of TasA triggers a cascade of malfunctions in biological processes that eventually lead to cell death in a subset of the population. This alteration in membrane stability might cause, given the differences in the expression levels of many genes involved in the respiratory process (Supplementary Data 1–3), the impaired respiration observed in ΔtasA cells, which could lead to increased ROS generation, leading to the full range of transcriptional and cytological alterations found in the tasA mutant over the course of time.

The physiological alterations observed in the ΔtasA strain have ecological implications. The intrinsic stress affecting the mutant cells reduced their ability to survive in natural environments; however, paradoxically, their higher induction of secondary metabolism seemed to indirectly and efficiently target fungal pathogens. This could explain why ΔtasA cells, which show clear signs of stress, display efficient biocontrol properties against P. xanthii. However, the sharp time-dependent decrease in the ΔtasA population on leaves suggests that its antifungal production could be beneficial during short-term interactions, but insufficient to support long-term antagonism unless there is efficient colonization and persistence on the plant surface. In this scenario, the deletion of tasA has a strong negative effect on bacterial cells, as we have demonstrated how a ΔtasA strain is more susceptible to ROS-induced damage (Fig. 6A, center panel, 6B, center graph), especially in the phyllosphere, where microbial cells are continuously subjected to different type of stresses, including oxidative stress[91]. We previously speculated that biofilm formation and antifungal production were two complementary tools used by Bacillus cells to efficiently combat fungi. Our current study supports this concept, but also enhances our understanding of the roles of the different ECM components. More specifically, we demonstrated that the amyloid protein TasA is the most important bacterial factor during the initial attachment and further colonization of the plant host, as it is necessary for the establishment of the bacterial cells over the plant leaves and for the maintenance of the normal cellular structure. The fact that the naturally occurring overexpression of the eps genes in the ΔtasA is unable to revert the adhesion defect of this strain downplay the importance of the EPS during the early establishment of physical contact. These observations are more consistent with a role for the EPS, along with BslA, in providing biofilms with protection against external stressors[14,92]. A similar role for a functional amyloid protein in bacterial attachment to plant surfaces was found for the Escherichia coli curli protein. Transcriptomic studies showed induction of curli expression during the earlier stages of attachment after the cells came into contact with the plant surface, and a curli mutant strain was defective in this interaction[93,94]. The distinct morphological and biochemical variations typical of amyloids make them perfect candidates for modulating cellular ecology. The observation that ΔtasA cells are incapable of colonization in the rhizosphere[95] clearly indicates the need for more in-depth investigation into these two distinctive ecological niches to understand the true roles of specific bacterial components. In addition to demonstrating enhanced production of antifungal compounds, our study revealed additional features that might contribute to the potency of stressed Bacillus cells in arresting fungal growth, in particular the overproduction of acetoin via increased expression of the alsS and alsD genes. Acetoin is a volatile compound produced via fermentative and overflow metabolism, and it has been demonstrated to mediate communication between beneficial bacteria and plants by activating plant defense mechanisms either locally or over long distances in a phenomenon known as induced systemic resistance (ISR)[96,97].

In summary, we have proven that the amyloid protein TasA participates in the proper maturation of Bacillus colonies, a function that, along with its previously reported role in ECM assembly, contributes to long-term survival, efficient colonization of the phylloplane, and a competitive advantage mediated by antifungal production. The absence of TasA leads to a series of physiological changes, likely triggered by alterations in membrane stability and dynamics, and effects on the FMMs, including an arrest of cell differentiation[23] that paradoxically increases the competitiveness of the mutant cells during short-term interactions via their ability to adapt to stress and their cellular response to early maturation. However, lack of TasA reduces cell fitness during mid- to long-term interactions via increased intrinsic cellular stress and the absence of a structured ECM, both of which limit the adaptability of the cells to the stressful phylloplane.

## Methods

**Bacterial strains and culture conditions**. The bacterial strains used in this study are listed in Supplementary Table 2. Bacterial cultures were grown at 37 °C from frozen stocks on Luria-Bertani (LB: 1% tryptone (Oxoid), 0.5% yeast extract

(Oxoid) and 0.5% NaCl) plates. Isolated bacteria were inoculated in the appropriate medium. The biotrophic fungus *Podosphaera xanthii* was grown at 25 °C from a frozen stock on cucumber cotyledons and maintained on them until inoculum preparation. Biofilm assays were performed on MSgg medium: 100 mM morpholinepropane sulfonic acid (MOPS) (pH 7), 0.5% glycerol, 0.5% glutamate, 5 mM potassium phosphate (pH 7), 50 µg/ml tryptophan, 50 µg/ml phenylalanine, 50 µg/ml threonine, 2 mM $MgCl_2$, 700 µM $CaCl_2$, 50 µM $FeCl_3$, 50 µM $MnCl_2$, 2 µM thiamine, 1 µM $ZnCl_2$. For the in vitro lipopeptide detection and assays with cell-free supernatants, medium optimized for lipopeptide production (MOLP)[98] was used: 30 g/liter peptone, 20 g/liter saccharose, 7 g/liter yeast extract, 1.9 g/liter $KH_2PO_4$, 0.001 mg/ liter $CuSO_4$, 0.005 mg/liter $FeCl_3.6H_2O$, 0.004 mg/liter $Na_2MoO_4$, 0.002 mg/liter KI, 3.6 mg/liter $MnSO_4.H_2O$, 0.45 g/liter $MgSO_4$, 0.14 mg/liter $ZnSO_4.7H_2O$, 0.01 mg/liter $H_3BO_3$, and 10 mg/liter citric acid. The pH was adjusted to 7 with 5 M NaOH prior to sterilization. For cloning and plasmid replication, *Escherichia coli* DH5α was used. *Escherichia coli* BL21(DE3) was used for protein purification. *Bacillus subtilis* 168 is a domesticated strain used to transform the different constructs into *Bacillus subtilis* NCIB3610. The antibiotic final concentrations for *B. subtilis* were: MLS (1 µg/ml erythromycin, 25 µg/ml lincomycin); spectinomycin (100 µg/ml); tetracycline (10 µg/ml); chloramphenicol (5 µg/ml); and kanamycin (10 µg/ml).

**Strain construction.** All of the primers used to generate the different strains are listed in Supplementary Table 3. To build the strain YNG001, the promoter of the fengycin operon was amplified with the Ppps-ecoRI.F and Ppps-HindIII.R primer pair. The PCR product was digested with EcoRI and HindIII and cloned into the pKM003 vector cut with the same enzymes. The resulting plasmid was transformed by natural competence into *B. subtilis* 168 replacing the *amyE* neutral locus. Transformants were selected via spectinomycin resistance. The same plasmid was used to build the strain YNG002 by transforming a Δ*tasA* strain of *B. subtilis* 168.

Strain YNG003 was constructed using the primers amyEUP-Fw, amyEUP-Rv, Ppps-Fw, Ppps-Rv, Yfp-Fw, Yfp-Rv, Cat-Fw. Cat-Rv, amyEDOWN-Fw, and amyEDOWN-Rv to separately amplify the relevant fragments. The fragments were then joined using the NEB builder HiFi DNA Assembly Master Mix (New England Biolabs). The construct was made using pUC19 digested with BamHI as the vector backbone. The final plasmid was then transformed into *B. subtilis* 168 replacing *amyE*, and transformants were selected via chloramphenicol resistance.

Strain JC97 was generated using the primers bslAUP-Fw, bslADOWN-Rv, Spc-Fw, Spc-Rv, bslaUP-Fw and bslADOWN-Rv, and XbaI-digested pUC19 as the vector backbone. The fragments were assembled using NEB Builder HiFi DNA Assembly Master Mix.

Strains JC70, JC81, and JC149 were constructed via site-directed mutagenesis (QuickChange Lightning Site Directed Mutagenesis Kit – Agilent Technologies). Briefly, the *tapA* operon (*tapA-sipW-tasA*), including its promoter, was amplified using the primers TasA_1_mutb and YSRI_2, and the resulting product was digested with BamHI and SalI and cloned into the pDR183 vector[99]. Next, the corresponding primers (Supplementary Table 3) were used to introduce the alanine substitution mutations into the desired positions of the TasA amino acid sequence. The entire plasmid was amplified from the position of the primers using Pfu DNA polymerase. The native plasmid, which was methylated and lacked the mutations, was digested with DpnI enzyme. The plasmids containing the native version of TasA (JC70) or the mutated versions (JC81 and JC149) were transformed into the *B. subtilis* 168 Δ(*tapA-sipW-tasA*) strain replacing the *lacA* neutral locus. Genetic complementation was observed in strain JC70 as a control. Transformants were selected via MLS resistance.

Plasmid pDFR6 (pET22b-*tasA*), which contains the open reading frame of the *tasA* gene from *B. subtilis* NCIB3610 without the signal peptide or the stop codon, was constructed as previously described[76].

Primers used in the analysis of gene expression by qRT-PCR are listed in Supplementary Table 4.

All of the *B. subtilis* strains generated were constructed by transforming *B. subtilis* 168 via its natural competence and then using the positive clones as donors for transferring the constructs into *B. subtilis* NCIB3610 via generalized SPP1 phage transduction[100].

**Biofilm assays.** To analyze colony morphology under biofilm-inducing conditions[101], the bacterial strains were grown on LB plates overnight at 37 °C, and the resulting colonies were resuspended in sterile distilled water at an $OD_{600}$ of 1. Next, 2-µl drops of the different bacterial suspensions were spotted on MSgg or LB agar plates (depending on the assay) agar plates and incubated at 30 °C. Colonies were removed at the appropriate time points (24, 48, and 72 h) for the different analyses.

For the Δ*eps*-Δ*tasA* co-inoculation assay, colonies were resuspended in sterile distilled water and mixed at a final $OD_{600}$ of 1. Next, the bacterial suspension was inoculated onto MSgg agar plates and incubated as described above. For the external complementation assay using purified TasA, a drop containing 80 µg of protein was spotted onto MSgg agar plates and allowed to dry. Next, Δ*tasA* cells were inoculated on top of the dried drop and incubated as described above.

For the CFU counts of the colonies from the different strains, 24-, 48- and 72-h-old colonies grown on MSgg agar plates were removed, resuspended in 1 ml of sterile distilled water, and subjected to mild sonication (three rounds of 20 second pulses at 20% amplitude). The resulting suspensions were serially diluted and

plated to calculate the CFUs per colony (total CFU). To estimate the CFUs corresponding to sporulated cells (CFU endospores), the same dilutions were heated at 80 °C for 10 min and plated. The sporulation percentage was calculated as (CFU endospores/total CFU) * 100.

**Biofilm fractionation.** To analyze the presence of TasA in the different strains, biofilms were fractionated into cells and ECM[101]. Both fractions were analyzed separately. In all, 72-h-old colonies grown under biofilm-inducing conditions on MSgg-agar plates were carefully lifted from the plates and resuspended in 10 ml of MS medium (MSgg broth without glycerol and glutamate, which were replaced by water) with a 25 ⁵/₈ G needle. Next, the samples were subjected to mild sonication in a Branson 450 digital sonifier (4–5 5 s pulses at 20% amplitude) to ensure bacterial resuspension. The bacterial suspensions were centrifuged at 9000 × *g* for 20 min to separate the cells from the extracellular matrix. The cell fraction was resuspended in 10 ml of MS medium and stored at 4 °C until further processing. The ECM fraction was filtered through a 0.22-µm filter and stored at 4 °C.

For protein precipitation, 2 ml of the cell or ECM fractions were used. The cell fraction was treated with 0.1 mg/ml lysozyme for 30 min at 37 °C. Next, both fractions were treated with a 10% final concentration of trichloroacetic acid and incubated in ice for 1 h. Proteins were collected by centrifugation at 13,000 × *g* for 20 min, washed twice with ice-cold acetone, and dried in an Eppendorf Concentrator Plus 5305 (Eppendorf).

**Cell membrane fractionation.** Crude membrane extracts were purified from 50 ml MSgg liquid cultures (with shaking) of the different *B. subtilis* strains. Cultures were centrifuged at 7000 × *g* for 10 min at 4 °C and then resuspended in 10 ml of PBS. Lysozyme was added at a final concentration of 20 µg/ml and the cell suspensions were incubated at 37 °C for 30 min. After incubation, the lysates were sonicated on ice with a Branson 450 digital sonifier using a cell disruptor tip and 45 s pulses at 50% amplitude with pauses of 30 s between pulses until the lysates were clear. Next, the cell lysates were centrifuged at 10,000 × *g* for 15 min to eliminate cell debris, and the supernatants were separated and passed through a 0.45-µm filter. To isolate the cell membrane, the filtered lysate was ultracentrifuged at 100,000 × *g* for 1 h at 4 °C. The supernatant, which contained the cytosolic proteins, was separated and kept at −20 °C. The pellet, which contained the crude membrane extract, was washed three times with PBS and processed using the CelLytic MEM protein extraction kit from Sigma. Briefly, the membrane fractions were resuspended in 600 µl of lysis and separation working solution (lysis and separation buffer + protease inhibitor cocktail) until a homogeneous suspension was achieved. Next, the suspension was incubated overnight at 4 °C on a stirring wheel. After incubation, the suspension is incubated at 37 °C for 30 min and then centrifuged at 3000 × *g* for 3 min. The DSM (upper phase) was separated and kept at −20 °C, and the DRM (lower phase) was washed three times with 400 µl of wash buffer by repeating the process from the 37 °C incubation step. Three washes were performed to ensure the removal of all hydrophilic proteins. The isolated DRM was kept at −20 °C until use. The DRM, DSM, and cytosolic fractions were used directly for immunodetection.

**Protein expression and purification.** Protein was expressed and purified as previously described[102] with some changes. Briefly, freshly transformed BL21(DE3) *E. coli* colonies were picked, resuspended in 10 mL of liquid LB with 100 µg/mL of ampicillin and incubated O/N at 37 °C with shaking. The next day, the pre-inoculum was used to inoculate 500 mL of LB supplemented with ampicillin, and the culture was incubated at 37 °C until an $OD_{600}$ of 0.7–0.8 was reached. Next, the culture was induced with 1-mM isopropyl β-D-1-thiogalactopyranoside (IPTG) and incubated O/N at 30 °C with shaking to induce the formation of inclusion bodies. The next day, cells were harvested via centrifugation (5000 × *g*, 15 min, 4 °C) resuspended in buffer A (Tris 50 mM, 150 mM NaCl, pH8), and then centrifuged again. The pellets were kept at −80 °C until purification or processed after 15 min. After thawing, cells were resuspended in buffer A, sonicated on ice (3 × 45 s, 60% amplitude) and centrifuged (15,000 × *g*, 60 min, 4 °C). The supernatant was discarded, as proteins were mainly expressed in inclusion bodies. The pellet was resuspended in buffer A supplemented with 2 % Triton X-100, incubated at 37 °C with shaking for 20 min and centrifuged (15,000 × *g*, 10 min, 4 °C). The pellet was extensively washed with buffer A, centrifuged (15,000 × *g* for 10 min, 4 °C), resuspended in denaturing buffer (Tris 50 mM NaCl 500 mM, 6 M GuHCl), and incubated at 60 °C overnight until complete solubilization occured. Lysates were clarified via sonication on ice (3 × 45 s, 60% amplitude) and centrifugation (15,000 × *g*, 1 h, 16 °C) and were then passed through a 0.45-µm filter prior to affinity chromatography. Protein was purified using an AKTA Start FPLC system (GE Healthcare). Soluble inclusion bodies were loaded into a HisTrap HP 5 mL column (GE Healthcare) previously equilibrated with binding buffer (50 mM Tris, 0.5 M NaCl, 20 mM imidazole, 8 M urea, pH 8). Protein was eluted from the column with elution buffer (50 mM Tris, 0.5 M NaCl, 500 mM imidazole, 8 M urea, pH 8). After the affinity chromatography step, the purified protein was loaded into a HiPrep 26/10 desalting column (GE Healthcare), and the buffer was exchanged to Tris 20 mM, NaCl 50 mM to perform the corresponding experiments.

**SDS-PAGE and immunodetection**. Precipitated proteins were resuspended in 1x Laemmli sample buffer (BioRad) and heated at 100 °C for 5 min. Proteins were separated via SDS-PAGE in 12% acrylamide gels and then transferred onto PVDF membranes using the Trans-Blot Turbo Transfer System (BioRad) and PVDF transfer packs (BioRad). For immunodetection of TasA, the membranes were probed with anti-TasA antibody (rabbit) used at a 1:20,000 dilution in Pierce Protein-Free (TBS) blocking buffer (ThermoFisher). For immunodetection of FloT-YFP, a commercial anti-GFP primary antibody (Clontech living colors full-length polyclonal antibody) developed in rabbit were used at a 1:1000 or dilution in the buffer mentioned above. A secondary anti-rabbit IgG antibody conjugated to horseradish peroxidase (BioRad) was used at a 1:3000 dilution in the same buffer. The membranes were developed using the Pierce ECL Western Blotting Substrate (ThermoFisher).

**Mass spectrometry analysis of protein bands**. The sequence corresponding to the band of the ECM fraction of JC81 (Supplementary Fig. 13A) was identified via tandem mass spectrometry using a "nano" ion trap system (HPLC-ESI-MS/MS). Briefly, the bands obtained after electrophoresis were cut out, washed, and destained. Subsequently, the disulfide bridges were reduced with DTT, cysteines were alkylated via the use of iodoacetamide, and in-gel trypsin digestion was performed to extract the peptides corresponding to the protein samples. This entire process was carried out automatically using an automatic digester (DigestPro, Intavis Bioanalytical Instruments). The peptides were then concentrated and desalted using a capture column C18 ZORBAX 300SB-C18 (Agilent Technologies, Germany), $5 \times 0.3$ mm, with 5-μm particle diameter and 300-Å pore size, using a gradient of 98% H2O:2% acetonitrile (ACN)/0.1% formic acid (FA) with a flow rate of 20 μL/min for 6 min. The capture column was connected in line to a ZORBAX 300SB-C18 analytical column (Agilent Technologies), $150 \times 0.075$ mm, with a 3.5-μm particle diameter and 300-Å pore size, through a 6-port valve. Elution of the samples from the capture column was performed over a gradient using FA 0.1% in water as the mobile phase A and FA 0.1% in ACN 80%/water 20% as the mobile phase B. The LC system was coupled through a nanospray source (CaptiveSpray, Bruker Daltonics) to a 3D ion trap mass spectrometer (amaZon speed ETD, Bruker Daltonics) operating in positive mode with a capillary voltage set to 1500 V and a sweep range: $m/z$ 300–1500. "Data-dependent" acquisition was carried out in automatic mode, which allowed the sequential collection of an MS spectrum in "full scan" ($m/z$ 300_1400) followed by an MS spectrum in tandem via CID of the eight most abundant ions. For identification, the software ProteinScape 3 (Bruker Daltonics) coupled to the search engine Mascot 3.1 (Matrix Science) was used, matching the MS/MS data against the Swiss-Prot and NCBInr databases.

**Bioassays on melon leaves**. Bacterial strains were grown in liquid LB at 30 °C overnight. The cells in the cultures were washed twice with sterile distilled water. The bacterial cell suspensions were adjusted to the same $OD_{600}$ and sprayed onto leaves of 4- to 5-week-old melon plants. Two hours later, a suspension of *P. xanthii* conidia was sprayed onto each leaf at a concentration of $4-10 \times 10^4$ spores/ml. The plants were placed in a greenhouse or in a growth chamber at 25 °C with a 16-h photoperiod, 3800 lux, and 85% RH. The severity of the symptoms in melon leaves was evaluated by the estimation of disease severity[103]. Disease severity was calculated by quantifying the leaf area covered by powdery mildew using FiJi[104] image software analysis and pictures of infected leaves. Briefly, the channels of the image were split and the area covered by powdery mildew was measured in 8-bit images by selecting the powdery mildew damage area (white powdery stains that cover the leaf) through image thresholding, given that the stains caused by the disease have higher pixel intensity values. Total leaf area was determined by manually selecting the leaf outline using the polygon selection tool the ratio of infection was calculated using the formula (see Eq. 1):

$$\text{Ratio of infection} = \frac{\text{damaged area}}{\text{total leaf area}} \times 100 \qquad (1)$$

The persistence of bacterial strains on plant leaves was calculated via CFU counts performed over the twenty-one days following inoculation. Three different leaves from three different plants were individually placed into sterile plastic stomacher bags and homogenized in a lab blender (Colworth Stomacher-400, Seward, London, UK) for 3 min in 10 ml of sterile distilled water. The leaf extracts were serially diluted and plated to calculate the CFUs at each time point. The plates were incubated at 37 °C for 24 h before counting.

The adhesion of bacterial cells to melon leaves was estimated by comparing the number of cells released from the leaf versus the cells attached to the surface. The surfaces of individual leaves were placed in contact with 100 ml of sterile distilled water in glass beakers and, after 10 min of stirring (300 rpm), the water and leaf were plated separately. The leaves were processed as described above. Adhesion was calculated as the ratio: (water CFU/total CFU) × 100. The data from all of the different strains were normalized to the result of the WT strain (100% adhesion).

**Antifungal activity of cell-free supernatant against *P. xanthii***. *B. subtilis* strains were grown for 72 h at 30 °C in MOLP medium, and the supernatant was centrifuged and filtered (0.22 μm). One-week-old cotyledons were disinfected with 20% commercial bleach for 30 s and then submerged two times in sterile distilled water for 2 min and then air dried. 10-mm disks were excised with a sterilized cork

borer, incubated with cell-free supernatants for 2 h, and then left to dry. Finally, the disks were inoculated with *P. xanthii* conidia on their adaxial surface with a soft paintbrush[105].

**Lipopeptides production analysis**. For the in vitro lipopeptide detection, bacteria were grown in MOLP for 72 h. The cultures were centrifuged, and the supernatants were filtered (0.22 μm) prior to analysis via MALDI-TOF/TOF mass spectrometry.

For the analysis of lipopeptide production in colonies, WT or Δ*tasA* colonies were grown on MSgg plates for 72 h at 30 °C. For the cell fractions, whole colonies were resuspended as described above in 1 mL of sterile distilled water and centrifuged at $5000 \times g$ for 5 min. The pellets were then resuspended in 1 ml of methanol and sonicated in a bath for 10 min. Cells were harvested via centrifugation at $5000 \times g$ for 5 min, and the supernatant containing the solubilized lipopeptides was filtered through a 0.22-μm filter and stored at 4 °C prior to analysis. For the agar fraction, after the colonies were removed, a piece of agar of approximately the same surface was sliced out and introduced into a 2-mL Eppendorf tube containing glass beads. In all, 1 mL of methanol was added, and then the tube was vigorously vortexed until the agar was broken down. Finally, the mixture was sonicated in a bath for 10 min and centrifuged at $5000 \times g$ for 5 min. The supernatant was filtered through a 0.22-μm filter and stored at 4 °C prior to analysis by MALDI-TOF/TOF.

For in situ lipopeptide detection on inoculated leaves, leaf disks were taken 21 days post-inoculation with a sterile cork borer and then placed directly on an UltrafleXtreme MALDI plate. A matrix consisting of a combination of CHCA (α-cyano-4-hydroxycinnamic acid) and DHB (2,5-dihydroxybenzoic acid) was deposited over the disks or the supernatants (for the in vitro cultures or the colonies' analysis), and the plates were inserted into an UltrafleXtreme MALDI-TOF/TOF mass spectrometer. The mass spectra were acquired using the Bruker Daltonics FlexControl software and were processed using Bruker Daltonics FlexAnalysis.

**Electron microscopy analysis**. For the scanning electron microscopy analysis, leaf disks were taken 21 days post-inoculation as previously described and fixed in 0.1 M sodium cacodylate and 2% glutaraldehyde overnight at 4 °C. Three washes were performed with 0.1 M sodium cacodylate and 0.1 M sucrose followed by ethanol dehydration in a series of ethanol solutions from 50% to 100%. A final drying with hexamethyldisilazane was performed as indicated[106]. The dried samples were coated with a thin layer of iridium using an Emitech K575x turbo sputtering coater before viewing in a Helios Nanolab 650 Scanning Electron Microscope and Focus Ion Beam (SEM-FIB) with a Schottky-type field emission electron gun.

For the transmission electron microscopy analysis, bacterial colonies grown on MSgg agar for the appropriate times were fixed directly using a 2% paraformaldehyde-2.5% glutaraldehyde-0.2 M sucrose mix in phosphate buffer 0.1 M (PB) overnight at 4 °C. After three washes in PB, portions were excised from each colony and then post-fixed with 1% osmium tetroxide solution in PB for 90 min at room temperature, followed by PB washes, and 15 min of stepwise dehydration in an ethanol series (30%, 50%, 70%, 90%, and 100% twice). Between the 50% and 70% steps, colonies were incubated in-bloc in 2% uranyl acetate solution in 50% ethanol at 4 °C. Following dehydration, the samples were gradually embedded in low-viscosity Spurr's resin: resin:ethanol, 1:1, 4 h; resin:ethanol, 3:1, 4 h; and pure resin, overnight. The sample blocks were embedded in capsule molds containing pure resin for 72 h at 70 °C.

For the immunolabeling assays, samples from the corresponding strains were grown under biofilm-inducing conditions at 30 °C. After 48 h of incubation, carbon-coated copper grids were deposited into the wells over the pellicles formed at the interface between the medium and the air (in the case of mutants unable to form a pellicle, copper grids were deposited in the interface) and incubated with the samples at 28 °C for 2 h. After incubation, the grids were washed in PSB for 5 min, and then the samples were fixed with a solution of 2% paraformaldehyde for 10 min, washed in PBS and blocked with Pierce Protein-Free (TBS) blocking buffer (ThermoFisher) for 30 min. Anti-TasA primary antibody was used at a 1:150 dilution in blocking buffer, and grids were deposited over drops of the antibody solution and incubated for 1 h at room temperature. Samples were washed three times with TBS -T (50 mM Tris-HCl, 150 mM NaCl, pH 7.5 - Tween20 0.1%) for 5 min and then exposed to 10-nm diameter immunogold-conjugated secondary antibody (Ted Pella) for 1 h at a 1:50 dilution. The samples were then washed twice with TBS-T and once with water for 5 min each. Finally, the grids were treated with glutaraldehyde (2%) for 10 min, washed in water for 5 min, negatively stained with uranyl acetate (1%) for 20 s and, lastly, washed once with water for 30 s.

The samples were left to dry and were visualized under a FEI Tecnai G$^2$ 20 TWIN Transmission Electron Microscope at an accelerating voltage of 80 KV. The images were taken using a side-mounted CCD Olympus Veleta with 2k x 2k Mp.

**Whole-transcriptome analysis and qRT-PCR**. Biofilms were grown on MSgg agar as described above. 24-, 48-, and 72-h colonies of the corresponding strains (WT or Δ*tasA*) were recovered and stored at −80 °C. All of the assays were performed in duplicate. The collected cells were resuspended and homogenized via passage through a $25^{5/8}$ G needle in BirnBoim A[107] buffer (20% sucrose, 10 mM Tris-HCl pH 8, 10 mM EDTA and 50 mM NaCl). Lysozyme (10 mg/ml) was added, and the

mixture was incubated for 30 min at 37 °C. After disruption, the suspensions were centrifuged, and the pellets were resuspended in Trizol reagent (Invitrogen). Total RNA extraction was performed as instructed by the manufacturer. DNA removal was carried out via in-column treatment with the rDNAse included in the Nucleo-Spin RNA Plant Kit (Macherey-Nagel) following the instructions of the manufacturer. The integrity and quality of the total RNA was assessed with an Agilent 2100 Bioanalyzer (Agilent Technologies) and by gel electrophoresis.

To perform the RNA sequencing analysis, rRNA removal was performed using the RiboZero rRNA removal (bacteria) Kit from Illumina, and 100-bp single-end read libraries were prepared using the TruSeq Stranded Total RNA Kit (Illumina). The libraries were sequenced using a NextSeq550 instrument (Illumina). The raw reads were pre-processed with SeqTrimNext[108] using the specific NGS technology configuration parameters. This pre-processing removes low quality, ambiguous and low complexity stretches, linkers, adapters, vector fragments, and contaminated sequences while keeping the longest informative parts of the reads. SeqTrimNext also discarded sequences below 25 bp. Subsequently, clean reads were aligned and annotated using the *B. subtilis subsp. subtilis str. 168* genome (NC_000964.3) as the reference with Bowtie2[109] in BAM files, which were then sorted and indexed using SAMtools v1.484[110]. Uniquely localized reads were used to calculate the read number value for each gene via Sam2counts (https://github.com/vsbuffalo/sam2counts). Differentially expressed genes (DEGs) between WT and ΔtasA were analyzed via DEgenes Hunter[111], which provides a combined *p* value calculated (based on Fisher's method[112]) using the nominal *p* values provided by from edgeR[113] and DEseq2[114]. This combined *p* value was adjusted using the Benjamini-Hochberg (BH) procedure (false discovery rate approach)[115] and used to rank all the obtained differentially expressed genes. For each gene, combined *p* value < 0.05 and log2-fold change >1 or <−1 were considered as the significance threshold. Heatmap and DEGs clusterization was performed using ComplexHeatmap[116] in Rstudio. STEM[117] was used to model temporal expression profiles independent of the data. Only profiles with a *p* value < 0.05 were considered in this study. The DEGs annotated with the *B. subtilis subsp. subtilis str. 168* genome were used to identify the Gene Ontology functional categories using sma3s[118] and TopGo Software[119]. Gephi software (https://gephi.org) was used to generate the DEG networks, and the regulon list was downloaded from subtiwiki (http://subtiwiki.uni-goettingen.de). The data were deposited in the GEO database (GEO accession GSE124307).

Quantitative real-time (qRT)-PCR was performed using the iCycler-iQ system and the iQ SYBR Green Supermix Kit from Bio-Rad. The primer pairs used to amplify the target genes were designed using the Primer3 software (http://bioinfo.ut.ee/primer3/) and Beacon designer (http://www.premierbiosoft.com/qOligo/Oligo.jsp?PID=1), maintaining the parameters described elsewhere[120]. For the qRT-PCR assays, the RNA concentration was adjusted to 100 ng/μl. Next, 1 μg of DNA-free total RNA was retro-transcribed into cDNA using the SuperScript III reverse transcriptase (Invitrogen) and random hexamers in a final reaction volume of 20 μl according to the instructions provided by the manufacturer. The qRT-PCR cycle was: 95 °C for 3 min, followed by PCR amplification using a 40-cycle amplification program (95 °C for 20 s, 56 °C for 30 s, and 72 °C for 30 s), followed by a third step of 95 °C for 30 s. To evaluate the melting curve, 40 additional cycles of 15 s each starting at 75 °C with stepwise temperature increases of 0.5 °C per cycle were performed. To normalize the data, the *rpsJ* gene, encoding the 30S ribosomal protein S10, was used as a reference gene[121]. The target genes *fenD*, encoding fengycin synthetase D, *alsS*, encoding acetolactate synthase, *albE*, encoding bacteriocin subtilosin biosynthesis protein AlbE, *bacB*, encoding the bacilysin biosynthesis protein BacB, and *srfAA* encoding surfactin synthetase A, were amplified using the primer pairs given in Supplementary Table 4, resulting in the generation of fragments of 147 bp, 82 bp, 185 bp, 160 bp, and 94 bp, respectively. The primer efficiency tests and confirmation of the specificity of the amplification reactions were performed as previously described[122]. The relative transcript abundance was estimated using the ΔΔ cyclethreshold (Ct) method[123]. Transcriptional data of the target genes was normalized to the *rpsJ* gene and shown as the fold-changes in the expression levels of the target genes in each *B. subtilis* mutant strain compared to those in the WT strain. The relative expression ratios were calculated as the difference between the qPCR threshold cycles (Ct) of the target gene and the Ct of the *rpsJ* gene ($\Delta Ct = Ctr_{\text{gene of interest}} - Ct_{rpsJ}$). Fold-change values were calculated as $2^{-\Delta\Delta Ct}$, assuming that one PCR cycle represents a two-fold difference in template abundance[124,125]. The qRT-PCR analyses were performed three times (technical replicates) using three independent RNA isolations (biological replicates).

**Flow cytometry assays**. Cells were grown on MSgg agar at 30 °C. At different time points, colonies were recovered in 500 μL of PBS and resuspended with a 25^5/8 G needle. For the promoter expression assays, colonies were gently sonicated as described above to ensure complete resuspension, and the cells were fixed in 4% paraformaldehyde in PBS and washed three times in PBS. To evaluate the physiological status of the different *B. subtilis* strains, cells were stained without fixation for 30 min with 5 mM 5-cyano-2,3-ditolyltetrazolium chloride (CTC) and 15 μM 3-(p-hydroxyphenyl) fluorescein (HPF).

The flow cytometry runs were performed with 200 μl of cell suspensions in 800 μL of GTE buffer (50 mM glucose, 10 mM EDTA, 20 mM Tris-HCl; pH 8), and the cells were measured on a Beckman Coulter Gallios™ flow cytometer using 488 nm

excitation. YFP and HPF fluorescence were detected with 550 SP or 525/40 BP filters. CTC fluorescence was detected with 730 SP and 695/30BP filters. The data were collected using Gallios™ Software v1.2 and further analyzed using Kaluza Analysis v1.3 and Flowing Software v2.5.1. Negative controls corresponding to unstained bacterial cells (or unlabeled cells corresponding to each strain for the promoter expression analysis) were used to discriminate the populations of stained bacteria in the relevant experiments and for each dye (Supplementary Fig. 19).

**Intracellular pH analysis**. Intracellular pH was measured as previously described[55]. Colonies of the different strains grown on MSgg agar at 30 °C were taken at different time points and recovered in potassium phosphate buffer (PPB) pH 7 and gently sonicated as described above. Next, the cells were incubated in 10 μl of 1 mM 5-(6)carboxyfluorescein diacetate succinimidyl (CFDA) for 15 min at 30 °C. PPB supplemented with glucose (10 mM) was added to the cells for 15 min at 30 °C to remove the excess dye. After two washes with the same buffer, the cells were resuspended in 50 mM PPB (pH 4.5).

Fluorescence was measured in a FLUOstar Omega (BMG labtech) microplate spectrofluorometer using 490 nm/525 nm as the excitation and emission wavelengths, respectively. Conversion from the fluorescence arbitrary units into pH units was performed using a standard calibration curve.

**Confocal laser scanning microscopy**. Cell death in the bacterial colonies was evaluated using the LIVE/DEAD BacLight Bacterial Viability Kit (Invitrogen). Equal volumes of both components included in the kit were mixed, and 2 μl of this solution was used to stain 1 ml of the corresponding bacterial suspension. Sequential acquisitions were configured to visualize the live or dead bacteria in the samples. Acquisitions with excitation at 488 nm and emission recorded from 499 to 554 nm were used to capture the images from live bacteria, followed by a second acquisition with excitation at 561 nm and emission recorded from 592 to 688 nm for dead bacteria.

For the microscopic analysis and quantification of lipid peroxidation in live bacterial samples, we used the image-iT Lipid Peroxidation Kit (Invitrogen) following the manufacturer's instructions with some slight modifications. Briefly, colonies of the different strains were grown on MSgg plates at 30 °C, isolated at different time points, and resuspended in 1 ml of liquid MSgg medium as described in the previous sections. In all, 5 mM cumene hydroperoxide (CuHpx)-treated cell suspensions of the different strains at the corresponding times were used as controls. The cell suspensions were then incubated at 30 °C for 2 h and then stained with a 10-μM solution of the imageIT lipid peroxidation sensor for 30 min. Finally, the cells were washed three times with PBS, mounted, and visualized immediately. Images of the stained bacteria were acquired sequentially to obtain images from the oxidized to the reduced states of the dye. The first image (oxidized channel) was acquired by exciting the sensor at 488 nm and recording the emissions from 509 to 561 nm, followed by a second acquisition (reduced channel) with excitement at 561 nm and recording of the emissions from 590 to 613 nm.

Membrane potential was evaluated using the image-iT TMRM (tetramethylrhodamine, methyl ester) reagent (Invitrogen) following the manufacturer's instructions. Colonies grown at 30 °C on MSgg solid medium were isolated at different time points and resuspended as described above. Samples treated prior to staining with 20 μM carbonyl cyanide m-chlorophenyl hydrazine (CCCP), a known protonophore and uncoupler of bacterial oxidative phosphorylation, were used as controls (Supplementary Fig. 8). The TMRM reagent was added to the bacterial suspensions to a final concentration of 100 nM, and the mixtures were incubated at 37 °C for 30 min. After incubation, the cells were immediately visualized by confocal laser scanning microscopy (CLSM) with excitation at 561 nm and emission detection between 576 and 683 nm.

The amounts of DNA damage in the *B. subtilis* strains at the different time points were evaluated via terminal deoxynucleotidyl transferase (TdT) dUTP Nick-End Labeling (TUNEL) using the *In-Situ* Cell Death Detection Kit with fluorescein (Roche) according to the manufacturer's instructions. *B. subtilis* colonies were resuspended in PBS and processed as described above. The cells were centrifuged and resuspended in 1% paraformaldehyde in PBS and fixed at room temperature for 1 h on a rolling shaker. The cells were then washed twice in PBS and permeabilized in 0.1% Triton X-100 and 0.1% sodium citrate for 30 min at room temperature with shaking. After permeabilization, the cells were washed twice with PBS and the pellets were resuspended in 50 μl of the TUNEL reaction mixture (45 μl label solution + 5 μl enzyme solution), and the reactions were incubated for one hour at 37 °C in the dark with shaking. Finally, the cells were washed twice in PBS, counterstained with DAPI (final concentration 500 nM), mounted, and visualized by CLSM with excitation at 488 nm and emission detection between 497 and 584 nm.

Membrane fluidity was evaluated via Laurdan generalized polarization (GP)[126]. Colonies of the different *B. subtilis* strains were grown and processed as described above. The colonies were resuspended in 50 mM Tris pH 7.4 with 0.5% NaCl. Laurdan reagent (6-dodecanoyl-N,N-dimethyl-2-naphthylamine) was purchased from Sigma-Aldrich (Merck) and dissolved in N,N-dimethylformamide (DMF). Samples treated prior to staining with 2% benzyl alcohol, a substance known to increase lipid fluidity[127,128], were used as positive controls (Supplementary Fig. 10). Laurdan was added to the bacterial suspensions to a final concentration of 100 μM. The cells were incubated at room temperature for 10 min, mounted, and then

visualized immediately using two-photon excitation with a Spectraphysics MaiTai Pulsed Laser tuned to 720 nm (roughly equivalent to 360 nm single photon excitation), attached to a Leica SP5 microscope. Emissions between 432 and 482 nm (gel phase) and between 509 and 547 nm (liquid phase) were recorded using the internal PMT detectors.

The localization of FloT in *B. subtilis* cells was evaluated using a FloT-YFP translational fusion in a WT genetic background (see Supplementary Table 2 for full genotype of the strains). Colonies grown at 30 °C on MSgg solid medium were isolated at different time points and resuspended as described above. Samples were mounted and visualized immediately with excitation at 514 nm and emission recorded from 518 to 596 nm.

All images were obtained by visualizing the samples using an inverted Leica SP5 system with a 63x NA 1.4 HCX PL APO oil-immersion objective. For each experiment, the laser settings, scan speed, PMT or HyD detector gain, and pinhole aperture were kept constant for all of the acquired images.

**Image analysis**. Image processing was performed using Leica LAS AF (LCS Lite, Leica Microsystems) and FIJI/ImageJ[104] software.

Images of live and dead bacteria from viability experiments were processed automatically, counting the number of live (green) or dead (red) bacteria in their corresponding channels. The percentage of dead cells was calculated dividing the number of dead cells by the total number of bacteria found on a field.

For processing the lipid peroxidation images, images corresponding to the reduced and oxidized channels were smoothed and a value of 3 was then subtracted from the two channels to eliminate the background. The ratio image was calculated by dividing the processed reduced channel by the oxidized channel using the FiJi image calculator tool. The ratio images were pseudo-colored using a color intensity look-up table (LUT), and intensity values of min 0 and max 50 were selected. All of the images were batch processed with a custom imageJ macro, in which the same processing options were applied to all of the acquired images. Quantification of the lipid peroxidation was performed in Imaris v7.4 (Bitplane) by quantifying the pixel intensity of the ratio images with the Imaris "spots" tool.

The Laurdan GP acquisitions were processed similarly. Images corresponding to the gel phase channel and the liquid phase channel were smoothed and a value of 10 was subtracted to eliminate the background. The Laurdan GP image was then calculated by applying the following formula (see equation 2):

$$\text{Laurdan GP} = \frac{(\text{gel phase channel} - \text{liquid phase channel})}{(\text{gel phase channel} + \text{liquid phase channel})} \quad (2)$$

The calculation was performed step by step using the FiJi image calculator tool. Pixels with high Laurdan GP values, typically caused by residual background noise, were eliminated with the "Remove outliers" option using a radius of 4 and a threshold of 5. Finally, the Laurdan GP images were pseudo-colored using a color intensity LUT, and intensity values of min 0 and max 1.5 were selected. This processing was applied to all of the acquisitions for this experiment. To quantify the Laurdan GP, bright field images were used for thresholding and counting to create counts masks that were applied to the Laurdan GP images to measure the mean Laurdan GP value for each bacterium.

TUNEL images were analyzed by subtracting a value of 10 in the TUNEL channel to eliminate the background. The DAPI channel was then used for thresholding and counting as described above to quantify the TUNEL signal. The same parameters were used to batch process and quantify all of the images.

To quantify the membrane potential, the TMRM assay images were analyzed as described above using the bright field channel of each image for thresholding and counting to calculate the mean fluorescence intensity in each bacterium. Endospores, which exhibited a bright fluorescent signal upon TMRM staining, were excluded from the analysis. This processing was applied to all of the acquisitions for this experiment.

To quantify the fluorescence of the bacteria expressing the *floT-yfp* construct, images were analyzed as described above using the bright field channel of each image for thresholding and counting to calculate the mean fluorescence intensity in each bacterium.

**Statistical analysis**. All of the data are representative of at least three independent experiments with at least three technical replicates. The results are expressed as the mean ± standard error of the mean (SEM). Statistical significance was assessed by performing the appropriate tests (see the figure legends). All analyses were performed using GraphPad Prism version 6. *p* values < 0.05 were considered significant. Asterisks indicate the level of statistical significance: *$p < 0.05$, **$p < 0.01$, ***$p < 0.001$, and ****$p < 0.0001$.

**Reporting summary**. Further information on research design is available in the Nature Research Reporting Summary linked to this article.

## Data availability

The RNA-seq data that support the findings of this study have been deposited in GEO database with the accession code GSE124307 [https://www.ncbi.nlm.nih.gov/geo/query/acc.cgi?acc=GSE124307]. The source data underlying Figs. 1A, B, D, 3A, C, 4A–D, 5B, 6B, 7B, 8C, D, 9B, C, D, E, and 10A–C; and Supplementary Figs. S1A, C, D, S4, S7B, C, D, S8B, S10B, S11, S12B, S13A, S14B, S15B, S16, S17B, S18B, and S19A, B are provided as a Source Data file.

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

## Acknowledgements

We thank Josefa Gómez Maldonado from the Ultrasequencing Unit of the SCBI-UMA for RNA sequencing, Juan Félix López Téllez at Bionand for his technical support in the transmission electron microscopy analysis, and the flow cytometry service at Bionand. We wish to thank Daniel López and Julia García-Fernández from the National Center for Biotechnology of the Spanish National Research Council (CNB-CSIC) for critical discussion and for kindly providing some of the bacterial strains used in this study. We would like to thank Ákos T. Kovács for critical reading of the manuscript and experimental suggestions. C.M.S is funded by the program Juan de la Cierva Formación (FJCI-2015-23810). This work was supported by grants from ERC Starting Grant (BacBio 637971) and Plan Nacional de I+D+I of Ministerio de Economía y Competitividad (AGL2016-78662-R).

## Author contributions

D.R. conceived the study. D.R., and J.C.A. and Y.N. designed the experiments. J.C.A. and Y.N. performed the main experimental work. M.C.P.B. gave support to some physiological experiments and did q-RT-PCR experiments. C.M.S. and L.D.M. analyzed and processed the whole transcriptomes. J.C.A. and J.P. performed and designed the confocal microscopy work and data analysis. D.R., J.C.A., C.M.S., A.V, A.P.G., and L.D.M. contributed critically to writing the final version of the manuscript.

## Competing interests

The authors declare no competing interests
