## [Peer Review File · Nature Communications]

Reviewers' comments:

Reviewer #1 (Remarks to the Author):

In their very interesting paper, the group of Romero discovered that in biofilm forming cells TasA, a highly secreted protein which forms extracellular protein fibrils essential for the formation and stabilization of the extracellular biofilm matrix, appear to have a previously unknown distinct role in the protection and supporting survival of the biofilm forming cells. Their results, which especially involved the in depth analysis of a Δ tasA mutant strain, suggest that the presence of TasA during biofilm formation might influence primary and secondary metabolism, pH and cell survival, thereby possibly preventing a kind of programmed cell death pathway (involving reactive oxygen species, altered membrane potential and fluidity). Their experiments suggest that TasA can also interact specifically with the membrane and might stabilize functional membrane microdomains (FMM), based on the observation that FMM associated Flotilins become delocalized in the absence of TasA. Most interestingly they could identify a TasA variant (JC81) which did not support the biofilm stabilizing extracellular fibril formation, but still complemented the newly observed phenotypes of a Δ tasA mutant strain, thereby separating both phenotypes.

I have some questions and comments

-During the whole study there were always different timepoints, 24, 48 and 72h, during biofilm developments investigated or assessed. But the changes in transcription were Δ tasA and wt were only assessed at 72h and not at 24 or 48h. I am quite sure that a lot of important information on regulatory processes during biofilm formation influenced by TasA could be detected especially in these earlier and medium developmental stages.

-Another important question would be whether the observed Δ tasA phenotypes are only detectable specifically during biofilm formation. In other words, is TasA important for e.g. cellular survival only during the specific circumstances and cellular environment of biofilm forming cells?

Do you observe these phenotypes (such as the lower survival rate) also in non-biofilm forming cells growing for example in full media? There one could check also different growthphases (exponential and stationary growth phase cells). One could also put an inducible tasA (operon) in trans and test its effect during normal growth.

Or one could examine single mutations in sinI or other regulators necessary for biofilm formation and combine them with the tasA deletion under biofilm growth conditions. Are the Δ tasA cells in a mixed biofilm culture of Δ tasA and Δ eps strains still more sensitive during survival tests (or any other of the observed relevant phenotype?).

-p 10 the change in fengycin production was discussed, whatabout surfactin?

-p 14 l395 please don't use this DRM acronym in the subheading, which was explained only later in the text (l419)

-16 l 491 What does TasAp.[Lys68_Asp69delinsAlaAla] mean? TasA (Lys68Ala, Asp69Ala)? Maybe it would be good just to list or provide the sequence of the mentioned TasA variants (JCXXX) in the supplement.

-p19 l544 and p20 613. I don't think that the term aggressor is appropriate to use here. Maybe other terms are better suited.

Reviewer #2 (Remarks to the Author):

In this manuscript the authors examine the role of the secreted protein TasA in the physiology and fitness of *Bacillus subtilis* when it is in the phyllophane (and in the laboratory). The context is that TasA forms fibers in the biofilm matrix but there are more global impacts that occur to the cell physiology of the *tasA* deletion strain when the protein is not made. The *tasA* mutant cells have a lower survival level in the biofilm community. Building from this experiment the authors define the mode of cell death in more detail and also go on to conclude that TasA restricted to membrane fractions is responsible for the survival.

It is this reviewer's opinion that some alternative ways of interpreting some of the data could be explored. That is not to refute the broader claim made in the paper that TasA has roles beyond acting as a scaffolding protein.

Alternative interpretations of data and experiments:

1) The authors examine the difference between the wild type strain and the *tasA* strain using gene expression analysis and many other techniques. One could make the argument that in this experimental setup the wild-type strain is perhaps not the most valuable control. This is because the environment the *tasA* cells are in will be so different from the environment provided by the wild type strain – e.g. in terms of oxygen gradient for starters. I am not advocating not having the wild type but suggesting that some of the comments and interpretations may have been better compared with the *eps* deletion strain. This strain would provide more of a habitat similar to that of the *tasA* mutant (flat and unstructured) and would therefore perhaps reduce the number of gene expression changes seen, thus highlighting those that are linked more directly with the absence of TasA than those linked with not being in a structured community. In short why would you not expect there to be major differences in gene expression? Key experiments could be included to support the differences being linked specifically to *tasA* by using the *eps* mutant as a “negative” control.

2) The authors rely on the use of a variant from TasA that they indicate cannot be secreted due to the mutation they have introduced. They provide evidence that they interpret as meaning that the strain JC81 cannot recover biofilm formation – however by comparison to the deletion strain the recovery is substantial and if the image had been presented as a wild-type it would not have been questioned by many people. To me there is an alternative way of viewing these data. It could be that the protein is simply not secreted as well (this is supported by the data in the immunoblots that are presented) and that the level of TasA in the ECM that is needed for full structuring is not reached when the TasA variant form is made. The role for TasA in stabilizing the membrane and helping cell survival could simply rely on a lower level of TasA than is needed for fully structuring of the biofilm. To help distinguish this proposed interpretation from that presented by the authors the following experiments would help:

a) TEM analysis of the JC81 strain to examine for TasA fibers using immunogold analysis – if they are present this would indicate that the cells have a way of circumventing the mutation to form fibers with the variant TasA.

b) Mass spec analysis of TasA from the extracellular environment to see if the peptides recovered correspond to the mature secreted form of the protein.

3) Some of the authors arguments are built on stating that the *tapA* deletion strain cannot form fibers of TasA (line 484)– this is in conflict with the authors previously published work (<https://www.ncbi.nlm.nih.gov/pmc/articles/PMC3103627/figure/F6/>), the *tapA* deletion strain does have extracellular TasA and some fibers of TasA are detectable. This should be considered and the reasoning in the results/discussion adjusted. Again the data presented would fit with a lower level of TasA being needed to recover the cell physiology defect than those required for facilitating complete rugose biofilm assembly. The argument is also built on the argument that making mutations in the TasA signal sequence blocks function – but there is very little protein here and that could easily be the cause. Line 458 – be specific about the mutations you have introduced and the anticipated defect – do not hide them in the table – it was hard to find the details. Keep in mind you have not shown that the protein cannot be secreted – you have shown that it is likely to be unstable when the variant amino acids are used.

4) Evidence that the TasA-mCherry fusion functions as the native TasA. This should be shown using biofilm complementation assays and also immunoblot assays using antibodies against both TasA and mCherry to confirm the validity of using the tool.

Figure comments:

1. There is an overuse of bar charts that present average values. The data should be presented as individual points – especially as the apparent deviation appears to be substantial.

2. For figure S2B- the images are presented such that the photographs of the strains are from a consecutive time course. This does not appear to be the case from the position of the bubbles in the images. Please make this clear.

3. Check ALL of the figures and figure legends for accuracy. There are errors in several where parts are mislabeled – are presented in the wrong order or are indicated as being there and are not (eg Fig S3 – A-E are mentioned in the legend but the image is A-C; Fig 1 B and C are in the wrong place either in the legend or text; . Make sure that all data presented in the figure is discussed in the text. Most of this is editorial work.

4. Full gel blots should be provided for Fig 7 C in the supplement to show that the fusion protein is not degraded to smaller forms.

Editorial comments:

The way in which the strains are referred to and called short hand terms in the paper is not easy to follow. Please use the genotypes to allow the reader to understand. For example in figure 10D you refer to *tasA* – this is not the case when you read the text. It is the JC81 strain. The “SiPmutant” is equally confusing and should simply be the genotype. In some places it sounds as if you are talking about a gene deletion, but it seems you are talking about a form of the *tasA* coding region with specific mutations in it.

The section on the cell health is very long, as is the data that is presented in the main figures. Consider ways of shortening this to get the key message more succinctly.

There are several places in the manuscript where the language needs tightening so that the correct meaning comes across to the reader. For example you mutate a gene not the protein – the protein is variant. When you say the fibers are resistant – be specific – to what? Under what circumstances? If you are talking about a mutation - detail it - don't make the reader hunt for the information.

Remove the phrase “validate our hypothesis”- test, explore, probe – validate is biased and suggests you do not have an open mind regarding the results.

Reviewer #3 (Remarks to the Author):

This manuscript by Camara-Almiron et al, well describes new functions of TasA, a protein that had been known as the main ECM component in *B. subtilis* biofilm. The authors show that knockout of TasA leads to physiological changes in the cells. While this finding is novel in terms of what was known about TasA, the multifaceted-role of major membrane proteins in maintaining membrane stability is not too surprising (for example, OmpA). Some major membrane proteins are proposed to maintain the membrane stability by interacting with other proteins and/or by tethering the membrane to other structural components such as the cell wall, and/or by playing a more direct role as a major component of the membrane integrity.

Given the ECM function of TasA, the authors findings have great potential in further understanding how membrane proteins/ECM may integrate the extracellular information to the membrane stability and alter the cell physiology, but the current study lacks in-depth on these topics limiting the novelty of the study. It is not clear at this point if the authors are observing a truly significant phenomenon or a side effect of deleting a major component of the cell envelope.

Concerning the biological significance of the findings, the authors attempt to propose TasA's role in PCD but this is hard to agree as TasA is not inducing cell death but preventing it, implying that the cell death is not programmed, and is rather a mere consequence of membrane instability caused by deleting this protein.

Taken together, the manuscript is well written and the reviewer thinks that the basic findings are very interesting that can potentially lead to more novel findings or develop a new concept which the current study is lacking.

Below are more comments.

1. Given the SEM image in Fig. 1C it is obvious that the *tasA* mutant has impaired cell structure that presumably resulted in the fixation of cells, already implying the membrane instability, but the authors seems to overlook this point.

2. Looking at the SEM images of the leaves (for example Fig. 1C) it looks as if more mutant cells are present on the leaves than the WT. The mutants are spreading but the WT are localized in a limited space. This is somewhat contradictory to the CFU counting (for example Fig. 1B). Does this mean most of the mutant cells are dead?

3. The authors explain the *tasA* mutant paradoxically release more fengycin than the WT and further discuss the ecological role, but this phenomenon does not seem to be a paradox to the reviewer and may simply be explained by the membrane instability. By other words, fengycin could be “leaking” out of the cells in *tasA* mutant, that is already implied by the data from the authors explaining that fengycin level in the *tasA* mutant supernatant was equivalent to that of the fengycin level found in the WT cell fraction (line 262). Still, key data are missing, and should be further examined by comparing fengycin in the cell fraction, cell-free supernatant, and total fraction between the WT and the mutant.

4. While the authors examined the membrane integrity using different methods, this study do not give

much information on the mechanism of how TasA stabilize the membrane. Given the interaction with FloT, one can test if TasA is influencing the membrane stability via FloT or not, by making a series of mutants. In addition, expressing FloT-YFP and TasA-mcherry in the same cell and comparing their localization is also something interesting in terms of their interactions.

5. Related to the comment above, the conclusion that TasA alter FloT localization is not convincing. As shown in Fig. 7. *tasA* mutant has very low level of FloT that is hardly detected even by WB. If FloT is mislocalized one would expect to detect FloT in a different fraction in *tasA* mutant compared to the WT but was not shown whereas FloT is also detected in the cytosolic fraction of the WT. Taken together, Fig. 7A, B and C is just pointing to the fact that TasA alter FloT expression level and do not provide information on the influence on the localization. The authors should compare the localization of FloT in strains where the total expression level in WB is similar among the strains. Expressing FloT-YFP under an inducible promoter in both strains may help.

6. It seems that the mcherry signal is leaking into the FM4-64 image, and should be confirmed or better use other membrane dyes.

7. While it is nicely shown that TasA is involved in the survival of cells, it is not convincing to call the observed phenomenon as PCD because no data was shown that this cell death is triggered nor any purpose of cell death was shown. It would have been interesting, for example, if the authors studied how the different timing and population of cell death controlled by TasA leads to different biofilm formations that may give biological significance of the cell death.

8. Given the self-assembly of TasA on model membranes (ref82), an interesting experiment would be to add exogenous TasA to *tasA* mutant cells, and even to other species, and see if they would restore or alter membrane stability. This would be something unique as opposed to the findings in conventional major membrane proteins.

9. It is tempting to speculate that the membrane instability led to impaired respiration that would generate ROS and therefore DNA damage, but is not fully discussed.

10. The reviewer could not find the explanation of the pps mutant used in Fig. 1D. Please explain in the manuscript or figure legend.

11. In fig.6A there are some black cells that presumably are spores, which abundance are largely different between the WT and the *tasA* mutant but has not been examined or well discussed. The data could be implying that TasA is regulating sporulation, in other word, cell development. Given the analogy of the role of ECM to eukaryotes, as also introduced by the authors in the Introduction, it would be interesting to examine how TasA is regulating sporulation. Using JC81 may be a good start to separate the role of TasA on biofilm formation and the cell physiology for sporulation.

12. Also, the *tasA* cells in fig.6A looks like they are elongated compared to the WT cells.

Minor comments

- Please check the order of figure Fig. 1B and C legends (B should be C, and C should be B).
- Why is the number of inoculated cells different between the WT and *tasA* mutant in Fig1B.
- The reviewer assume that he labeling on the Y-axis of *tasA* mutant should be " 10^4 " and not 10.
- Fig. 4D. From the provided microscopic image, it does not look like 20% of cells are dead in the *tasA* mutant.

• Fig. 10B is presented in a different magnification compared to Fig. 1C and difficult to compare, especially for the biofilm formation on the leaf.

Reviewer's responses

Reviewer 1

In their very interesting paper, the group of Romero discovered that in biofilm forming cells TasA, a highly secreted protein which forms extracellular protein fibrils essential for the formation and stabilization of the extracellular biofilm matrix, appear to have a previously unknown distinct role in the protection and supporting survival of the biofilm forming cells. Their results, which especially involved the in depth analysis of a Δ tasA mutant strain, suggest that the presence of TasA during biofilm formation might influence primary and secondary metabolism, pH and cell survival, thereby possibly preventing a kind of programmed cell death pathway (involving reactive oxygen species, altered membrane potential and fluidity). Their experiments suggest that TasA can also interact specifically with the membrane and might stabilize functional membrane microdomains (FMM), based on the observation that FMM associated Flotilins become delocalized in the absence of TasA. Most interestingly they could identify a TasA variant (JC81) which did not support the biofilm stabilizing extracellular fibril formation, but still complemented the newly observed phenotypes of a Δ tasA mutant strain, thereby separating both phenotypes.

I have some questions and comments

R- First of all, we are grateful to the reviewer for the time and effort spent in reviewing the manuscript and appreciate the constructive criticism. We have followed the suggestions proposed by the reviewer, and we believe that the manuscript has been improved considerably. Below are the responses to the specific comments of the referee.

Q- During the whole study there were always different timepoints, 24, 48 and 72h, during biofilm developments investigated or assessed. But the changes in transcription were Δ tasA and WT were only assessed at 72h and not at 24 or 48h. I am quite sure that a lot of important information on regulatory processes during biofilm formation influenced by TasA could be detected especially in these earlier and medium developmental stages.

R- We agree with the reviewer, and to address this point, we additionally extracted total RNA from WT and Δ *tasA* colonies at 24 h and 48 h, performed RNA-seq analysis, and reanalyzed the data obtained at 72 h. The new information supports and improves the conclusions derived from the RNA-seq analysis at 72 h and has given us more confidence in the initial hypothesis and clarified some points that were missing from our previous analysis. Specifically, we have seen that the transcriptomic changes in Δ *tasA* colonies occur from the very beginning at 24 h (1356 differentially expressed genes compared to the WT) and reach their maximum at 48 h (1765 differentially expressed genes compared to the WT). In fact, what we see at 72 h compared to the other time points is, in general, a stabilization of the gene expression changes and a switch-off of many of the differentially expressed genes (833 differentially expressed genes). This observation is consistent with all of the experimental data, in which most of the physiological changes, such as, stabilization of the CFU counts (fig. 4C), increase in the percentage of cell death (fig. 4D), alteration of membrane potential and changes in membrane fluidity (figs. 6A and 6B) are present from 48 h of growth. Furthermore, at 24 h we see strong activation of the general stress response and the SOS regulon, consistent with the higher generation of ROS that we see at this time-point (fig. 4A), suggesting a relationship between all of the observed changes at later times with those events originated at 24 h. The results section (lines 152-254) has been rewritten to incorporate all of the new data. We have also edited figure 2 of the manuscript and modified supplementary figure 3.

Q- Another important question would be whether the observed Δ *tasA* phenotypes are only detectable specifically during biofilm formation. In other words, is TasA important for e.g. cellular survival only during the specific circumstances and cellular environment of biofilm forming cells? Do you observe these phenotypes (such as the lower survival rate) also in non-biofilm forming cells growing for example in full media? There one could check also different growth phases (exponential and stationary growth phase cells). One could also put an inducible *tasA* (operon) in trans and test its effect during normal growth. Or one could examine single mutations in *sinI* or other regulators necessary for biofilm formation and combine them with the *tasA* deletion under biofilm growth conditions. Are the Δ *tasA* cells in a mixed biofilm culture of Δ *tasA* and Δ *eps* strains still more sensitive during survival tests (or any other of the observed relevant phenotype?).

R- We agree with the reviewer's comments and believe that they raise important points that must be properly addressed. To see if the studied phenotypes are dependent on

biofilm formation, we initially performed a live/dead assay via confocal microscopy (as described in the Materials and Methods section of the manuscript) of colonies grown for 24 h or 48 h on solid LB (a complex medium). Although LB does not induce biofilm formation as strongly as MSgg, *Bacillus subtilis* can still form wrinkly colonies as it does on MSgg, and $\Delta tasA$ forms morphologically undifferentiated colonies (fig. S7A). Accordingly, we saw that at 48 h, the $\Delta tasA$ colonies contain a significantly higher proportion of dead cells (17.86 ± 0.92) compared with the WT colonies (3.88 ± 0.33) (fig. S7B). These values mirror those obtained on solid medium at the same time-point for both strains (16.80 ± 1.17 for $\Delta tasA$ and 4.45 ± 0.67 for WT) (fig. 4D). To continue exploring this hypothesis, we followed the reviewer's suggestion and measured the proportion of cell death in both strains in liquid MSgg cultures grown with shaking at 30 °C (conditions prone to planktonic growth). We generated growth curves for both strains in this medium to discard growth defects associated to the absence of TasA and took samples during exponential and stationary phase. Our results show that the proportion of cell death for both strains remained very low, and no significant differences were detected between $\Delta tasA$ and WT under planktonic growth conditions. Therefore, we conclude that all the phenotypes associated with the absence of TasA occur when the two strains are growing under biofilm-inducing conditions and that they are not directly related to the medium composition. This new data have been included in the Results section of the manuscript (lines 333-344) and a new supplementary figure has been added (fig. S7)

To further demonstrate that the alternative role proposed for TasA is independent from its function in ECM assembly, we did two complementary experiments, as suggested by the reviewer. First, we constructed a $\Delta sinI$ strain (an anti-repressor of SinR, which is a transcriptional repressor of the ECM genes) (fig. S18A) and performed a viability assay. We observed that this strain shows percentages of cell death similar to those of the WT strain at all times assayed (1.53 ± 0.25 at 48 h and 2.51 ± 0.50 at 72 h) (fig. S18B), suggesting that even when the matrix genes are strongly repressed¹, the basal expression levels of TasA are sufficient to prevent the increased cell death of the $\Delta tasA$ strain. Second, we analyzed the percentage of cell death in a mixed biofilm of $\Delta tasA$ and Δeps strains co-inoculated at a 1:1 proportion. We observed a complete recovery of the wrinkly phenotype in MSgg medium (which does not happen when the two strains are grown separately) (fig. S18A and fig. S2A); however, the proportion of cell death was higher than that observed for the WT at 48 h (10.39 ± 1.20) and 72 h (14.04 ± 0.72) (fig. S18B). These findings indicate that the extracellular TasA provided by the Δeps cells is sufficient to restore ECM assembly (wrinkly colonies), but not to prevent

cell death. These results agree with the data related to the JC81 strain and further confirm that: i) The role of TasA in preventing cell death does not rely on its structural role in ECM assembly, and ii) TasA must be produced by the cells in order to reach the cell membrane and prevent cell death. We have included these new data in the Results section of the manuscript (lines 568-586), and a new supplementary figure has been added (Fig. S18).

Q- p 10, the change in fengycin production was discussed, what about surfactin?

R- We focused that section of the manuscript on the lipopeptide fengycin given its relevance in the antagonistic interaction between *B. subtilis* and fungal pathogens. In fact, *in situ* mass spectrometry analyses of plants inoculated with WT or Δ *tasA* revealed higher relative levels of both lipopeptides, surfactin and fengycin, in plants inoculated with Δ *tasA* cells (fig. 1E). Consistently, MALDI-TOF/TOF analysis of cell-free supernatants (fig. 3B) also showed higher relative levels of fengycin in the Δ *tasA* strain compared to the level in the WT strain, supportive of the antifungal activity exhibited by Δ *tasA* cells (figure 1D). As suggested by reviewer 3, apart from the analysis of cell-free supernatants (figure 3B), we have performed mass spectrometry analysis of colonies grown for 72 h on solid MSgg. We analyzed the relative levels of fengycin and surfactin in the cells (cells fraction) and in the solid medium underneath and surrounding the WT or Δ *tasA* colonies (agar fractions) (fig. S6). The relative levels of both lipopeptides were higher in both fractions of the Δ *tasA* colony compared to those of the WT colony (fig. 3B top and fig. S6). We have reported these new data also mentioning surfactin in the manuscript (lines 271-280). Fig. 3B has been updated to include the spectrum of the agar fractions, and a new supplementary figure (fig. S6) has been added.

Q- p 14 I395 please don't use this DRM acronym in the subheading, which was explained only later in the text (I419)

This has been corrected.

Q- 16 | 491 What does TasAp.[Lys68_Asp69delinsAlaAla] mean? TasA (Lys68Ala, Asp69Ala)? Maybe it would be good just to list or provide the sequence of the mentioned TasA variants (JCXXX) in the supplement.

The reviewer is right. We have clarified this point and replaced TasAp.[Lys68_Asp69delinsAlaAla] by TasA (Lys68Ala, Asp69Ala) as suggested by the reviewer.

Q- p19 l544 and p20 613. I don't think that the term aggressor is appropriate to use here. Maybe other terms are better suited.

This change has been introduced in the manuscript, and the term "aggressor" has been replaced by "environmental stressors" in p19 line 619 and "stressor" in p21 line 702.

Reviewer 2

In this manuscript the authors examine the role of the secreted protein TasA in the physiology and fitness of *Bacillus subtilis* when it is in the phyllophane (and in the laboratory). The context is that TasA forms fibers in the biofilm matrix but there are more global impacts that occur to the cell physiology of the *tasA* deletion strain when the protein is not made. The *tasA* mutant cells have a lower survival level in the biofilm community. Building from this experiment the authors define the mode of cell death in more detail and also go on to conclude that TasA restricted to membrane fractions is responsible for the survival. It is this reviewer's opinion that some alternative ways of interpreting some of the data could be explored. That is not to refute the broader claim made in the paper that TasA has roles beyond acting as a scaffolding protein.

R- We would like to express our gratitude for the reviewer's critical evaluation of our manuscript, and we appreciate the constructive criticism and the time invested in reading and interpreting the data. We have considered most of the reviewer's comments, suggestions, and experiments, and we believe that the manuscript has been improved as a result. The responses to the specific points raised by the reviewer can be found below.

Alternative interpretations of data and experiments:

Q-1) The authors examine the difference between the wild type strain and the *tasA* strain using gene expression analysis and many other techniques. One could make the argument that in this experimental setup the wild-type strain is perhaps not the most valuable control. This is because the environment the *tasA* cells are in will be so different from the environment provided by the wild type strain – e.g. in terms of oxygen gradient for starters. I am not advocating not having the wild type but suggesting that some of the comments and interpretations may have been better compared with the *eps* deletion strain. This strain would provide more of a habitat similar to that of the *tasA* mutant (flat and unstructured) and would therefore perhaps reduce the number of gene expression changes seen, thus highlighting those that are linked more directly with the absence of TasA than those linked with not being in a structured community. In short why would you not expect there to be major differences in gene expression? Key experiments could be included to support the differences being linked specifically to *tasA* by using the *eps* mutant as a “negative” control.

R- We agree with the reviewer that the Δeps strain is structurally more similar to the $\Delta tasA$ strain than the WT strain; therefore, its use as a control might be useful to highlight the transcriptional changes that occur only in the $\Delta tasA$ strain. However, we have performed several new key experiments (detailed below), including transcriptomics and microscopy, and considering the new amount of data generated and the focus of the current work on the physiological effects of the absence of TasA, we believe these results are better suited to be communicated as part of a new manuscript.

[Redacted]

Q- 2) The authors rely on the use of a variant from TasA that they indicate cannot be secreted due to the mutation they have introduced. They provide evidence that they interpret as meaning that the strain JC81 cannot recover biofilm formation – however by comparison to the deletion strain the recovery is substantial and if the image had been presented as a wild-type it would not have been questioned by many people. To me there is an alternative way of viewing these data. It could be that the protein is simply not secreted as well (this is supported by the data in the immunoblots that are presented) and that the level of TasA in the ECM that is needed for full structuring is not reached when the TasA variant form is made. The role for TasA in stabilizing the membrane and helping cell survival could simply rely on a lower level of TasA than is needed for fully structuring of the biofilm. To help distinguish this proposed interpretation from that presented by the authors the following experiments would help:

R- We appreciate the observations made by the reviewer; however, we do not fully agree with some of the statements. The strain JC81 bears a point mutation within the *tasA* sequence (TasA Lys68Ala, Asp69Ala) that affects biofilm formation, while no effects are evident in the physiological phenotypes caused by the *tasA* deletion, indicating that the phenotypes observed in the absence of TasA are not related to the structural role of the protein in biofilm formation as the main protein component of the ECM.

The WT allele encoding TasA (*tasA_{native}*) (fig. S12A) rescues the typical wrinkle phenotype of *B. subtilis* WT NCIB 3610 in the Δ *tasA* background, and the resulting strain is morphologically different from JC81 (the strain expressing the *tasA_{variant}* allele TasA Lys68Ala, Asp69Ala)(fig. S12A), a finding supporting that indeed, the TasA allele expressed in JC81 is unable to support WT biofilm formation. Nevertheless, according to the data presented, we agree with the reviewer that there is less TasA Lys68Ala, Asp69Ala in the ECM of the JC81 strain; however, there is no evidence suggesting that this version of TasA is not secreted as well as the WT version of the protein. This possibility is supported by the following observations: i) we do not see any larger bands corresponding to the unprocessed form of the protein in any fraction analyzed via western blot, as occurs, for instance, in mutants of the signal peptidase *sipW*^{3,4}; nor ii) there are no mutations in the signal peptide of the *tasA_{variant}* allele TasA Lys68Ala, Asp69Ala expressed by strain JC81 that could explain why the protein is not properly processed or secreted. Our working hypothesis is that TasA Lys68Ala, Asp69Ala, given its inability to function properly as part of the ECM, as evidenced by the altered phenotype of JC81 compared to the WT, is less stable than the WT version of the

protein; therefore, it is less resistant to the physicochemical conditions of the cell, i.e. protease susceptibility, and it can be easily and faster degraded than is the WT version of TasA. In fact, the study of this protein variant is part of ongoing work that aims to answer these questions. Nevertheless, the experiments suggested by the reviewer have been conducted, as we believe they can help clarify much of the discussion.

Q- a) TEM analysis of the JC81 strain to examine for TasA fibers using immunogold analysis – if they are present this would indicate that the cells have a way of circumventing the mutation to form fibers with the variant TasA.

The results from this experiment can be found in the new fig. S13B. We analyzed the presence of TasA in WT, JC81 and Δ *tasA* cells via immunolabeling with secondary antibodies conjugated to 10-nm diameter gold particles and transmission electron microscopy (see the updated Materials and Methods section lines 1044-1059). The WT samples show a dense network of fibers and extracellular material that exhibit high reactivity with the secondary antibody and, therefore, a large proportion of gold particles can be observed (fig. S13B, left panel). The JC81 samples, in contrast, show dense masses of extracellular material but no well-defined fibers can be observed. This extracellular material shows low reactivity with the secondary antibody compared to the WT and thus, fewer gold particles are detected (fig. S13B center panel). Lastly, no fibers were observed in Δ *tasA* cells and almost no gold particles were detected (fig. S13B right panel).

b) Mass spec analysis of TasA from the extracellular environment to see if the peptides recovered correspond to the mature secreted form of the protein.

The results from this experiment can be found in the new fig. S13A. For this experiment, we performed a biofilm fractionation assay with WT, JC81 and Δ *tasA* (as a negative control) colonies as described in the Materials and Methods section of the manuscript. The precipitated proteins from each sample were resolved via SDS-PAGE followed by Coomassie staining or western blot analysis using an anti-TasA antibody (fig. S13A left). The western blot allowed us to identify the highest running anti-TasA-reacting bands in the ECM fraction corresponding to the JC81 sample (fig. S13A left, lane 6) and to cut out the corresponding Coomassie-stained gel at the exact height. This band was sequenced via tandem mass spectrometry (see the updated Materials and Methods section lines 921-946). The analysis revealed that all of the peptides detected in the main anti-TasA-reacting band of the ECM fraction of the JC81 strain correspond to the mature form of TasA.

The conclusions from both experiments are: i) The TasA variant protein present in the ECM of JC81 (TasA Lys68Ala, Asp69Ala) cannot form well defined fibers compared with those observed in the WT ECM, and it is less reactive against an anti-TasA antibody; and ii) the TasA variant Lys68Ala, Asp69Ala present in the ECM of strain JC81 corresponds to the mature form of the protein. These results support that, indeed, the mutated version of the protein is unable to form fibers similar to those formed by the WT protein; therefore, this protein is less stable in the ECM and is easily degraded after a certain period of time.

The information from both experiments has been added to the results section of the manuscript (lines 542-550). As mentioned above, a new supplementary figure has been incorporated (supplementary figure S14) and the Materials and Methods section has been updated.

Q- 3) Some of the authors arguments are built on stating that the *tapA* deletion strain cannot form fibers of TasA (line 484)– this is in conflict with the authors previously published work (<https://www.ncbi.nlm.nih.gov/pmc/articles/PMC3103627/figure/F6/>), the *tapA* deletion strain does have extracellular TasA and some fibers of TasA are detectable. This should be considered and the reasoning in the results/discussion adjusted. Again, the data presented would fit with a lower level of TasA being needed to recover the cell physiology defect than those required for facilitating complete rugose biofilm assembly. The argument is also built on the argument that making mutations in the TasA signal sequence blocks function – but there is very little protein here and that could easily be the cause. Line 458 – be specific about the mutations you have introduced and the anticipated defect – do not hide them in the table – it was hard to find the details. Keep in mind you have not shown that the protein cannot be secreted – you have shown that it is likely to be unstable when the variant amino acids are used.

R- We wish to thank reviewer 2 very much for pointing out this issue. Indeed, the *tapA* deletion strain does form TasA “fibrils”, although these are produced in much fewer quantity and they appear separated from the cells⁵; therefore, following the reviewer’s suggestion, the results section of the manuscript has been modified accordingly (line 530).

We agree with the reviewer, and we believe that the information regarding the mutations introduced for the signal peptide mutant of TasA (SiPmutant) are not easy to find in the text. The signal peptide of TasA was mutated via amino acid substitution of

the three continuous lysines located at the N-terminal end of the TasA sequence, which have been reported to be important for signal peptidase-dependent secretion in *B. subtilis*^{3,6}. The introduced mutations were: Lys4Ala, Lys5Ala, Lys6Ala. We have modified the results section of the manuscript to clarify this point (lines 508-510).

Again, we agree with the reviewer in the fact that less protein is found in both the cellular and ECM fractions of the SiPmutant strain (fig. 8C) and that one possible explanation is the decreased in protein stability due to the introduced mutations. We repeated the western blot with the same samples at earlier time-points (24 and 48 h) and the obtained results similar to those shown in fig. 8C (data not shown). However, we believe that there is also a valid alternative interpretation. The phenotype observed for the TasA SiPmutant strain is completely different from that observed for a *tasA* deletion strain, and the fact that TasA is a secreted protein with a signal peptide suggests that mutations affecting the secretion of the protein cause accumulation of the unprocessed product that is, most likely, unstable and more easily degraded by housekeeping proteases. One could speculate that the mRNA of *tasA* has a regulatory role in the physiological status of cell; thus, what is clear from this experiment is that having the mRNA of TasA is insufficient to prevent cellular damage. Considering what has been discussed above, both explanations would be correct or complementary, and no changes have been introduced in the manuscript or the figures regarding this section. Nevertheless, we remain open to the reviewer's reasoning and opinion, and if this data must be modified or deleted from the text, we leave it to the reviewer's or the editor criteria.

Q- Evidence that the TasA-mCherry fusion functions as the native TasA. This should be shown using biofilm complementation assays and also immunoblot assays using antibodies against both TasA and mCherry to confirm the validity of using the tool.

R- The results from this experiment can be found in supplementary figure S20. The genotype of this strain is: *tasA::spc, amyE::(tapA-sipW-tasA-mCherry) (mIs)*, which has been corrected from supplementary table 5 due to an error. This very same strain or strains with the same genotype have been used and published elsewhere⁷⁻⁹. The expression of *tasA-mCherry* can restore the characteristic wrinkles typical of the WT colony morphology in a Δ *tasA* background (fig. S20A). We performed a biofilm fractionation assay followed by western blotting using anti-mCherry or anti-TasA antibodies to track the sizes of the translational fusion in the complemented strain. The western blot with the anti-mCherry antibody showed a band between 48 and 63 Kda,

corresponding to the size of the TasA-mCherry translational fusion (approximately 55 Kda) in the cellular and ECM fraction of the strain (fig. S20B left, lanes 1 and 2). The western blot with the anti-TasA antibody revealed a clear band between 48 and 63 Kda, corresponding to the expected size of TasA-mCherry in the cellular and ECM fractions of the strain (fig. S20B right, lanes 3 and 4); however, we observed smaller size bands with much higher intensity corresponding to different sizes of TasA in both fractions. As part of ongoing work, we have observed that TasA is further processed apart from the SipW-signal peptide-dependent cleavage, and this process is clear based on the smaller size bands that appear in any western blot performed with an anti-TasA antibody, even in cellular or ECM protein fractions from a WT strain (for instance, see fig. 9B, lane 1). Nonetheless, we can detect the size corresponding to the full translational fusion TasA-mCherry, and given the data obtained via confocal and fluorescence microscopy, this result supports the use of this strain as a means to study the cellular localization of TasA.

Figure comments:

Q- 1. There is an overuse of bar charts that present average values. The data should be presented as individual points – especially as the apparent deviation appears to be substantial.

R- Done. Dots representing the individual values have been added to the bar charts.

Q- 2. For figure S2B- the images are presented such that the photographs of the strains are from a consecutive time course. This does not appear to be the case from the position of the bubbles in the images. Please make this clear.

R- We usually run experiments with several colonies from the same strain (with the same genotype) grown on the same plate. Sometimes, at the different time points, it was aesthetically more convenient to photograph a different colony. Nonetheless, we have accepted this comment from the reviewer and modified figure S2B with images from the same colonies at all of the studied time points.

Q- 3. Check ALL of the figures and figure legends for accuracy. There are errors in several where parts are mislabeled – are presented in the wrong order or are indicated as being there and are not (eg Fig S3 – A-E are mentioned in the legend but the image is A-C; Fig 1 B and C are in the wrong place either in the legend or text; . Make sure that all data presented in the figure is discussed in the text. Most of this is editorial work.

R- We really thank the reviewer for pointing out this issue, and we truly apologize for all of the errors related to the figures. All of the figures, including those derived from the reviewers' comments, along with the manuscript have been carefully revised to avoid errors in labelling or order.

Q- 4. Full gel blots should be provided for Fig 7 C in the supplement to show that the fusion protein is not degraded to smaller forms.

R- Full gel blots corresponding to fig. 7C have been added as a new supplementary figure (fig. S21).

Editorial comments:

Q- The way in which the strains are referred to and called shorthand terms in the paper is not easy to follow. Please use the genotypes to allow the reader to understand. For example, in figure 10D you refer to *tasA* – this is not the case when you read the text. It is the JC81 strain. The “SiPmutant” is equally confusing and should simply be the genotype. In some places it sounds as if you are talking about a gene deletion, but it seems you are talking about a form of the *tasA* coding region with specific mutations in it.

R- As also suggested by reviewer 1, we have added a detailed description of the introduced mutations in the corresponding sections of the manuscript, and we believe that this should be clearer now. We would like to thank reviewer 2 for pointing the error regarding figure 10D. This has already been corrected in the revised submission.

Q- The section on the cell health is very long, as is the data that is presented in the main figures. Consider ways of shortening this to get the key message more succinctly.

R- Considering the contributions of reviewer 1 and 3 to this part of the results section of the article, we do not see how we can make this section shorter at the moment. We did our best to accommodate the new data without increasing the length; however, we are willing to accept any suggestions from the reviewer or the editor to shorten this part of the article if it is still considered necessary.

Q- There are several places in the manuscript where the language needs tightening so that the correct meaning comes across to the reader. For example, you mutate a gene not the protein – the protein is variant. When you say the fibers are resistant – be specific – to what? Under what circumstances? If you

are talking about a mutation - detail it - don't make the reader hunt for the information.

R- We have tried to improve the language in certain parts of the manuscript and have corrected the specific examples pointed by the reviewer (lines 78-79 of the introduction). As suggested by the reviewer, we have detailed the introduced mutations in the manuscript (lines 508-510 and lines 538-539).

Q- Remove the phrase “validate our hypothesis”- test, explore, probe – validate is biased and suggests you do not have an open mind regarding the results.

R- This has been corrected.

Reviewer 3

This manuscript by Camara-Almiron et al, well describes new functions of TasA, a protein that had been known as the main ECM component in *B. subtilis* biofilm. The authors show that knockout of TasA leads to physiological changes in the cells. While this finding is novel in terms of what was known about TasA, the multifaceted-role of major membrane proteins in maintaining membrane stability is not too surprising (for example, OmpA). Some major membrane proteins are proposed to maintain the membrane stability by interacting with other proteins and/or by tethering the membrane to other structural components such as the cell wall, and/or by playing a more direct role as a major component of the membrane integrity.

Given the ECM function of TasA, the authors findings have great potential in further understanding how membrane proteins/ECM may integrate the extracellular information to the membrane stability and alter the cell physiology, but the current study lacks in-depth on these topics limiting the novelty of the study. It is not clear at this point if the authors are observing a truly significant phenomenon or a side effect of deleting a major component of the cell envelope. Concerning the biological significance of the findings, the authors attempt to propose TasA's role in PCD but this is hard to agree as TasA is not inducing cell death but preventing it, implying that the cell death is not programmed, and is rather a mere consequence of membrane instability caused by deleting this protein.

Taken together, the manuscript is well written, and the reviewer thinks that the basic findings are very interesting that can potentially lead to more novel findings or develop a new concept which the current study is lacking.

R- We would like to thank Reviewer 3 for the critical evaluation of our manuscript, the constructive criticism and the helpful comments. We have accepted most of the reviewer's proposed experiments, and we believe that the revised submission has been greatly improved as a result of the review process. However, we respectfully disagree with some of the reviewer's statements presented above.

TasA, as correctly pointed by the Reviewer, is a bacterial functional amyloid implicated in ECM formation. However, up to now this protein has not been described or characterized as a membrane protein and was not regarded as a major membrane protein. We did not see the same cytological damage and physiological changes triggered by the absence of TasA, when other membrane proteins, such as the flotillin-like proteins FloT or FloA or other proteins of the cell envelop such as TapA, were deleted. Therefore, the role of TasA seems to go far beyond that of all the other membrane proteins tested. Concerning the new proposed biological role of TasA in maintaining normal cellular physiology, it is true that the presence of TasA in the membrane prevents cell death and indeed, this is precisely what it was stated in our initial submission (line 590 and line 614).

We believed the term programmed cell death (PCD) to be appropriate for the Δ tasA phenotype, based on previous literature¹⁰⁻¹⁴ and to differentiate it from accidental cell death (ACD), defined as the "virtually instantaneous and uncontrollable form of cell death corresponding to the physical disassembly of the plasma membrane caused by extreme physical, chemical, or mechanical cues"¹⁵. However, we understand and accept the reviewer's concerns over use of the term and we have moderated statements where PCD was used, replacing it by simply "cell death" or similar phrases to refer to the stress-induced cellular damage and death that occurs in Δ tasA cells.

However, we would argue that this does not affect the importance to the results presented, showing a range of cytological and physiological alterations in the absence of TasA, and crucially, demonstrating for the first time that a bacterial functional amyloid is directly associated with the plasma membrane. Therefore, we strongly believe that our contribution to the field is sufficiently novel and significant to be considered for publication.

In any case, we appreciate the time invested by the reviewer and believe that the points raised and experimental suggestions given have helped to clarify important aspects of the manuscript and have considerably improved it. The responses to the reviewer's comments can be found below.

Q- 1. Given the SEM image in Fig. 1C it is obvious that the *tasA* mutant has impaired cell structure that presumably resulted in the fixation of cells, already implying the membrane instability, but the authors seems to overlook this point.

R- The reviewer is correct, and differences in cellular structure can be observed between WT and Δ *tasA* cells in the SEM images in fig. 1C. We opted to be conservative in the interpretation of these electron microscopy images. We initially considered highlighting different patterns of colonization on the phylloplane, especially when more detailed and specific data on membrane instability and functionality were further obtained in our study. Nonetheless, we have mentioned what it has been pointed by the reviewer in the results and discussion sections of the article (results section lines 124-125).

Q- 2. Looking at the SEM images of the leaves (for example Fig. 1C) it looks as if more mutant cells are present on the leaves than the WT. The mutants are spreading but the WT are localized in a limited space. This is somewhat contradictory to the CFU counting (for example Fig. 1B). Does this mean most of the mutant cells are dead?

R- Certainly, the reviewer is right, and the SEM images from fig. 1C show information that might appear to contradict to the CFU counts from inoculated leaves (fig. 1B). The SEM images were selected as representative of the different spatial distribution pattern of cells lacking TasA compared to those of the WT. Some of the Δ *tasA* cells are, indeed, dead, as mentioned by the reviewer and shown later in the text (fig. 4D). However, part of this apparent contradiction is also due to the intrinsic error of the experimental method, as it is physically not possible to recover all of the cells from the inoculated leaves. This method, with variations, has been extensively used in many other publications¹⁶⁻¹⁸, and it is established that with an appropriate number of biological replicates and significant statistics, this method is sufficiently valid to evaluate bacterial persistence on plant tissues.

Q- 3. The authors explain the *tasA* mutant paradoxically release more fengycin than the WT and further discuss the ecological role, but this phenomenon does not seem to be a paradox to the reviewer and may simply be explained by the membrane instability. By other words, fengycin could be “leaking” out of the cells in *tasA* mutant, that is already implied by the data from the authors explaining that fengycin level in the *tasA* mutant supernatant was equivalent to that of the fengycin level found in the WT cell fraction (line 262). Still, key data

are missing, and should be further examined by comparing fengycin in the cell fraction, cell-free supernatant, and total fraction between the WT and the mutant.

R- We agree with the reviewer. In retrospect, after analyzing the physiological defects associated to the absence of TasA and discovering the membrane instability, the leaking of lipophilic small-size molecules is not that strange. We have changed the results section of the manuscript to indicate this (lines 410-413). However, we would like to call the reviewer's attention to our flow cytometry analysis (fig. 3A) that clearly demonstrated that there is a larger population of Δ *tasA* cells that express the promoter of the fengycin operon compared with the number of WT cells at any analyzed time point, providing additional support for robust production of this molecule.

In the initially submitted manuscript, we analyzed the levels of fengycin in cell-free supernatant fractions of WT or Δ *tasA* cultures in MOLP medium (which is optimized for lipopeptide production) via mass spectrometry (lines 271-273, fig. 3B bottom spectrum). According to the flow cytometry and the RNAseq analysis, larger relative amounts of fengycin were found in cell free supernatants from Δ *tasA* cultures compared to the amount found in WT samples. As suggested by the reviewer (and additionally reviewer 1), we have performed a new mass spectrometry analysis of WT and Δ *tasA* cells growing as colonies on solid MSgg medium. The results from this analysis revealed the presence of one order of magnitude higher relative amounts of fengycin in the cells and agar fractions of Δ *tasA* colonies compared to the same fractions of WT colonies (fig. 3B top spectrum and fig. S6). We have added the mass spectrometry data corresponding to the agar fraction in figure 3B, and we have included this new information in the results section (lines 271-280) and updated the Materials and Methods section (lines 997-1009).

Q- 4. While the authors examined the membrane integrity using different methods, this study do not give much information on the mechanism of how TasA stabilize the membrane. Given the interaction with FloT, one can test if TasA is influencing the membrane stability via FloT or not, by making a series of mutants. In addition, expressing FloT-YFP and TasA-mcherry in the same cell and comparing their localization is also something interesting in terms of their interactions.

R- In the manuscript, we do not mention the interaction of FloT and TasA; in fact, the initial version of the manuscript lacked any experimental evidence to support this statement. What we found is that TasA is associated with the DRM as is FloT, a protein used as a reference for known proteins of the DRM. However, considering that both

proteins are associated with the DRM of the cell membrane, it was reasonable to suggest a putative interaction between the two proteins.

[Redacted]

Furthermore, as also suggested by the reviewer, we analyzed the cell death phenotypes of a $\Delta floT$ strain and a $\Delta floT floA$ double mutant strain to test whether the loss of normal FloT distribution due to the variations in gene expression caused by the absence of TasA is responsible for the increased cell death rate observed in $\Delta tasA$ colonies. The results from this experiment are shown in supplementary figure 11C, and they demonstrate that the cell death levels in the $\Delta floT$ and $\Delta floT floA$ strains are similar to that found in the WT strain, indicating that the cell death observed in the $\Delta tasA$ mutant is not directly related to the loss of FloT. These findings also suggest that the defects in FloT expression and the cellular distribution are consequences of the altered membrane stability caused by the absence of TasA. Taken together, these results show that i) deletion of other major membrane proteins, such as FloT and FloA, which contrary to TasA, are present in almost all the cells, does not trigger the cell death observed in a *tasA* mutant and ii) show that the alterations related to FloT are a consequence (and not the cause) of the membrane instability.

[Redacted]

These findings that support our statements on the role of TasA at the membrane level are part of an ongoing project in our laboratory and thus are not included in this manuscript.

[Redacted]

Q- 5. Related to the comment above, the conclusion that TasA alter FloT localization is not convincing. As shown in Fig. 7. *tasA* mutant has very low level of FloT that is hardly detected even by WB. If FloT is mislocalized one would expect to detect FloT in a different fraction in *tasA* mutant compared to the WT but was not shown whereas FloT is also detected in the cytosolic fraction of the WT. Taken together, Fig. 7A, B and C is just pointing to the fact that TasA alter FloT expression level and do not provide information on the influence on the localization. The authors should compare the localization of FloT in strains where the total expression level in WB is similar among the strains. Expressing FloT-YFP under an inducible promoter in both strains may help.

R- As correctly pointed by the reviewer, it is true that the *tasA* mutant shows lower levels of FloT compared to that found in the WT strain based on the western blot shown in fig. 7C. We have followed the reviewer's recommendation, and we have generated strains expressing a FloT-YFP translational fusion under the control of an IPTG-inducible promoter. We have performed confocal microscopy experiments with 72 h colonies grown in MSgg medium supplemented with different IPTG concentrations to see if FloT localization was recovered in the Δ *tasA* mutant (fig. R3). We found that even at the lowest IPTG concentration used in our experiment, FloT-YFP did not form foci (the typical pattern observed for this protein expressed from its native promoter)

and instead, the fluorescent signal decorated nearly the complete cell membrane in WT or $\Delta tasA$ cells. These findings point to an artifactual pattern of the protein, due to the saturation levels of FloT, which consequently alters the normal distribution pattern of the protein in the cell membrane both in WT and $\Delta tasA$ cells.

Furthermore, we have analyzed the newly obtained RNA-seq data comparing the $\Delta tasA$ and WT strains at 24 h, 48 h and 72 h to examine the *floT* expression levels. These data show fluctuations in the expression levels of this protein over time, i.e., induction of *floT* at 24 h ($\log_2FC = 1.25$), repression at 48 h ($\log_2FC = -1.66$), and no significant differences at 72 h compared to the WT levels. Indeed, fig. 7B and C could reflect this point; however, fig 7C also points out that TasA is located in the DRM of the cell membrane, which has not been previously described. We used the term “mislocalization” because we observed that the *tasA* mutant shows, in the cells that exhibit any signal, bright points corresponding to FloT-YFP that are concentrated in certain parts of the cell instead of homogeneously distributed foci across the cell membrane, as occurs in the WT strain (fig. 7A). It is important to recall that we originally used FloT as complementary evidence of the membrane instability caused by the absence of TasA. The fact that TasA is located in the DRM and that its absence leads to alterations in the levels of FloT with a complete loss of the normal distribution pattern, still seems relevant to explain this observation.

Based on these newly obtained results, we have rewritten the results section of the manuscript to state that the loss of the normal FloT distribution pattern in $\Delta tasA$ cells might reflect changes in the stability of the cell membrane or fluctuations in the expression levels of FloT in the *tasA* mutant (lines 434-463). However, if it is reviewer’s opinion that more changes are needed in this section of the manuscript, we are willing to accept any suggestions.

Figure R4. Induction of *P_{hyperspank}-floT-yfp* expression via IPTG in WT and Δ *tasA* samples. The images correspond to cells of the different strains from 72 h colonies grown on solid MSgg medium supplemented with the indicated IPTG concentrations.

Q. 6. It seems that the mcherry signal is leaking into the FM4-64 image and should be confirmed or better use other membrane dyes.

R- We would like to thank reviewer 3 for pointing out this issue. The reviewer is absolutely right, and there is some overlap between the excitation/emission spectra of both fluorophores. We repeated this experiment via confocal microscopy analysis using the membrane stain CellBrite Fix 488 (Biotium), which is excited at 480 nm and emits at 513 nm. Both maxima are well separated from those of mCherry (Ex. 587/ Em. 610), ensuring that no leaking in the signal is observed when using both fluorophores. We have modified the microscopy images presented in figure 7D, and we have added a supplementary figure showing 48 h cells labelled with TasA-mCherry without CellBrite staining as a negative control for membrane stain and 48 h WT cells with CellBrite staining as negative control for the mCherry fluorescence (fig. S20C).

Q- 7. While it is nicely shown that TasA is involved in the survival of cells, it is not convincing to call the observed phenomenon as PCD because no data was shown that this cell death is triggered, nor any purpose of cell death was shown. It would have been interesting, for example, if the authors studied how the different timing and population of cell death controlled by TasA leads to different biofilm formations that may give biological significance of the cell death.

R- We have discussed above why the term PCD was originally used and we have accepted the reviewer's concerns regarding the term. We have replaced "PCD" throughout the manuscript using alternative phrasing to refer to the stress-induced cell death that occurs in Δ *tasA* cells.

Concerning the second part of the reviewer's comment, we believe that there is already work in the literature that describes part of what the reviewer is suggesting², and this is not the focus of the current work. However, we remain open to the reviewer's suggestion if specific experiments are necessary and required to complement this part of the work.

Q- 8. Given the self-assembly of TasA on model membranes (ref82), an interesting experiment would be to add exogenous TasA to *tasA* mutant cells, and even to other species, and see if they would restore or alter membrane stability. This would be something unique as opposed to the findings in conventional major membrane proteins.

R- We believe that the reviewer raised an interesting point, and as suggested, we purified TasA and performed an external complementation experiment with a Δ *tasA*

strain on solid medium by adding different amounts of TasA, which has already been shown to be biologically active^{20,21} (see the Materials and Methods lines 975-903). The results of this experiment have been included in supplementary figure S19, and they indicate that exogenous TasA has no detectable effect on the phenotype (fig. S19A) under these conditions. Next, we used a LIVE/DEAD viability stain to examine whether the cell death observed in the *tasA* mutant was somehow recovered to WT levels, which would indicate a restoration of membrane stability. However, we observed that the Δ *tasA* strain complemented with exogenous TasA had as similar proportion of cell death as did the Δ *tasA* strain (fig. S19B) (results section lines 573-575). This finding is consistent to the results obtained via the co-inoculation of a mixture of the Δ *tasA* and Δ *eps* strains (as suggested by reviewer 1) in which we observed that the TasA produced and secreted by the Δ *eps* cells is sufficient to restore biofilm and colony morphology (fig. S18A) but not the cell death rate, which resembled that of the Δ *tasA* strain alone (fig. S18B). This observation indicates, once again, that the phenotypes described in this work for a *tasA* mutant are not related to the absence of a structured ECM, but rather to the absence of TasA itself in the cell membrane. Therefore, considering that exogenous TasA has no biological effect on the producing strain lacking TasA, we see no point in doing the same experiments with other species.

Q- 9. It is tempting to speculate that the membrane instability led to impaired respiration that would generate ROS and therefore DNA damage but is not fully discussed.

R- We agree with the reviewer. A paragraph considering this possibility has been added to the results (lines 346-348) and discussion (lines 673-677) sections of the manuscript.

Q- 10. The reviewer could not find the explanation of the *pps* mutant used in Fig. 1D. Please explain in the manuscript or figure legend.

R- *pps* is the acronym for plipastatin, which is equivalent to fengycin, the main antifungal compound produced by *B. subtilis*. The Δ *pps* strain is a mutant carrying a deletion in the *ppsB* gene, which is required for fengycin synthesis. This strain was used as a negative control in the biocontrol experiments in plants against the fungal pathogen *Podosphaera xanthii*.

Q- 11. In fig.6A there are some black cells that presumably are spores, which abundance are largely different between the WT and the *tasA* mutant but has not been examined or well discussed. The data could be implying that TasA is

regulating sporulation, in other word, cell development. Given the analogy of the role of ECM to eukaryotes, as also introduced by the authors in the Introduction, it would be interesting to examine how TasA is regulating sporulation. Using JC81 may be a good start to separate the role of TasA on biofilm formation and the cell physiology for sporulation.

R- As correctly pointed by the reviewer, the endospore abundance in the Δ *tasA* strain is much lower than that of the WT strain. This is due, as mentioned in the manuscript, to the sporulation defect observed in ECM mutants, which has been previously studied in other works^{19,22}. We analyzed the proportion of spores present in the JC81 strain and found normal level of spores compared with that of the WT strain at all time points (fig. S4), demonstrating that the mutated allele could complement the sporulation defect associated with the Δ *tasA* strain. We agree with the reviewer that it is interesting to speculate that TasA might somehow regulate the sporulation process, especially considering that TasA was initially discovered as part of the spore coat⁴. We greatly appreciate the reviewer's suggestion, and we will save it for future work; however, we believe that this question must be thoroughly investigated and is beyond the scope of this manuscript.

Q- 12. Also, the *tasA* cells in fig.6A looks like they are elongated compared to the WT cells.

R- Again, the reviewer is correct. We have found consistent alterations in the shape and size of Δ *tasA* cells during the course of this study. We speculated that these defects are associated with the altered expression levels of genes involved in cell shape, such as *mreB*, *mreC* or *mreBH*, or in cell division and septum placement, such as *minD* or *minJ* (supplementary tables 1 to 3) in the Δ *tasA* strain. However, these observations are the subject of a new and ongoing project.

Minor comments

Q- · Please check the order of figure Fig. 1B and C legends (B should be C, and C should be B).

R- Done.

Q- · Why is the number of inoculated cells different between the WT and *tasA* mutant in Fig1B.

R- The first point in the graph does not really correspond to the number of inoculated cells, but rather to the number of cells harvested from the leaf after 4 hours post inoculation. This point has been clarified in the figure legend.

Q- · The reviewer assumes that the labeling on the Y-axis of *fasA* mutant should be “*10⁴” and not 10.

R- We want to thank reviewer 3 for pointing out this issue. This has been corrected.

Q- · Fig. 4D. From the provided microscopic image, it does not look like 20% of cells are dead in the *fasA* mutant.

R- We appreciate the reviewer’s observation. The chosen confocal microscopy images for that figure corresponded to fields that, precisely, had as similar percentages of cell death as the quantified mean value. The two images from the WT and $\Delta fasA$ strains have 133 and 142 cells, respectively, and the proportions of cell death in the images are 4.51% and 21.83%, respectively, which is nearly 20% of the cells in the field of the $\Delta fasA$ strain.

Q- · Fig. 10B is presented in a different magnification compared to Fig. 1C and difficult to compare, especially for the biofilm formation on the leaf.

R- This issue has been corrected. The images in figure 10B have been modified, and the scale bars have been adjusted to make the comparison with figure 1C easier. The detail of biofilm formation on the leaf is now clearly observable.

Supplementary references

- 1 Chai, Y., Chu, F., Kolter, R. & Losick, R. Bistability and biofilm formation in *Bacillus subtilis*. *Mol Microbiol* **67**, 254-263, doi:10.1111/j.1365-2958.2007.06040.x (2008).
- 2 Asally, M. *et al.* Localized cell death focuses mechanical forces during 3D patterning in a biofilm. *Proc Natl Acad Sci U S A* **109**, 18891-18896, doi:10.1073/pnas.1212429109 (2012).
- 3 Tjalsma, H. *et al.* Conserved serine and histidine residues are critical for activity of the ER-type signal peptidase SipW of *Bacillus subtilis*. *J Biol Chem* **275**, 25102-25108, doi:10.1074/jbc.M002676200 (2000).
- 4 Stover, A. G. & Driks, A. Secretion, localization, and antibacterial activity of TasA, a *Bacillus subtilis* spore-associated protein. *J Bacteriol* **181**, 1664-1672 (1999).
- 5 Romero, D., Vlamakis, H., Losick, R. & Kolter, R. An accessory protein required for anchoring and assembly of amyloid fibres in *B. subtilis* biofilms. *Mol Microbiol* **80**, 1155-1168, doi:10.1111/j.1365-2958.2011.07653.x (2011).
- 6 Tjalsma, H., Bolhuis, A., Jongbloed, J. D., Bron, S. & van Dijl, J. M. Signal peptide-dependent protein transport in *Bacillus subtilis*: a genome-based survey of the secretome. *Microbiol Mol Biol Rev* **64**, 515-547 (2000).
- 7 Kolodkin-Gal, I. *et al.* D-amino acids trigger biofilm disassembly. *Science* **328**, 627-629, doi:10.1126/science.1188628 (2010).
- 8 van Gestel, J., Vlamakis, H. & Kolter, R. From cell differentiation to cell collectives: *Bacillus subtilis* uses division of labor to migrate. *PLoS Biol* **13**, e1002141, doi:10.1371/journal.pbio.1002141 (2015).
- 9 Vogt, C. M., Schraner, E. M., Aguilar, C. & Eichwald, C. Heterologous expression of antigenic peptides in *Bacillus subtilis* biofilms. *Microb Cell Fact* **15**, 137, doi:10.1186/s12934-016-0532-5 (2016).
- 10 Bayles, K. W. Bacterial programmed cell death: making sense of a paradox. *Nat Rev Microbiol* **12**, 63-69, doi:10.1038/nrmicro3136 (2014).
- 11 Peeters, S. H. & de Jonge, M. I. For the greater good: Programmed cell death in bacterial communities. *Microbiol Res* **207**, 161-169, doi:10.1016/j.micres.2017.11.016 (2018).
- 12 Engelberg-Kulka, H., Amitai, S., Kolodkin-Gal, I. & Hazan, R. Bacterial programmed cell death and multicellular behavior in bacteria. *PLoS Genet* **2**, e135, doi:10.1371/journal.pgen.0020135 (2006).

- 13 Dwyer, D. J., Camacho, D. M., Kohanski, M. A., Callura, J. M. & Collins, J. J. Antibiotic-induced bacterial cell death exhibits physiological and biochemical hallmarks of apoptosis. *Mol Cell* **46**, 561-572, doi:10.1016/j.molcel.2012.04.027 (2012).
- 14 Thomas, V. C. *et al.* A central role for carbon-overflow pathways in the modulation of bacterial cell death. *PLoS Pathog* **10**, e1004205, doi:10.1371/journal.ppat.1004205 (2014).
- 15 Galluzzi, L. *et al.* Molecular mechanisms of cell death: recommendations of the Nomenclature Committee on Cell Death 2018. *Cell Death Differ* **25**, 486-541, doi:10.1038/s41418-017-0012-4 (2018).
- 16 Zeriouh, H., de Vicente, A., Perez-Garcia, A. & Romero, D. Surfactin triggers biofilm formation of *Bacillus subtilis* in melon phylloplane and contributes to the biocontrol activity. *Environ Microbiol* **16**, 2196-2211, doi:10.1111/1462-2920.12271 (2014).
- 17 Kim, J. G. & Mudgett, M. B. Tomato bHLH132 Transcription factor controls growth and defense and is activated by *Xanthomonas euvesicatoria* effector XopD during pathogenesis. *Mol Plant Microbe Interact*, MPMI05190122R, doi:10.1094/MPMI-05-19-0122-R (2019).
- 18 Molina-Santiago, C. *et al.* The extracellular matrix protects *Bacillus subtilis* colonies from *Pseudomonas* invasion and modulates plant co-colonization. *Nat Commun* **10**, 1919, doi:10.1038/s41467-019-09944-x (2019).
- 19 Vlamakis, H., Aguilar, C., Losick, R. & Kolter, R. Control of cell fate by the formation of an architecturally complex bacterial community. *Genes Dev* **22**, 945-953, doi:10.1101/gad.1645008 (2008).
- 20 Romero, D., Aguilar, C., Losick, R. & Kolter, R. Amyloid fibers provide structural integrity to *Bacillus subtilis* biofilms. *Proc Natl Acad Sci U S A* **107**, 2230-2234, doi:10.1073/pnas.0910560107 (2010).
- 21 El Mammeri, N. *et al.* Molecular architecture of bacterial amyloids in *Bacillus biofilms*. *FASEB J* **33**, 12146-12163, doi:10.1096/fj.201900831R (2019).
- 22 Aguilar, C., Vlamakis, H., Guzman, A., Losick, R. & Kolter, R. KinD is a checkpoint protein linking spore formation to extracellular-matrix production in *Bacillus subtilis* biofilms. *MBio* **1**, doi:10.1128/mBio.00035-10 (2010).

Reviewers' comments:

Reviewer #1 (Remarks to the Author):

All my questions and comments were addressed and I was very happy to have read the revised version of this much improved and very interesting manuscript.

P.S. Just a little thing: In Fig Suppl 12 B the Fig legends do not match the Fig. You might want to consider to replace the numbers in the the figure by the description of the strains

Reviewer #2 (Remarks to the Author):

In this revised manuscript the authors have undertaken a number of experiments that have increased the quality of the work and should be commended for this. However there still seem to be a number of issues with the figures.

For instance, there appear to be a large number of cases of where data has been repeatedly presented in different parts of the manuscript and where the repetition has not been disclosed. This is not to infer that the data has been misrepresented as another strain etc – more that these issues have not been disclosed and it would suggest that there could be ways of condensing the data into more compressed formats. It would also be more transparent.

There are also other presentation issues that warrant attention in both the figures and the legends. Finally, check that time points are mentioned in the legend and that ALL of the data in the images is represented in the legend.

Examples are given below but please note that this is NOT an exhaustive list:

Figure 1- Need to add in E to the legend – currently there are two part D's.

Figure 3 A – the grey histogram looks very similar between the top and bottom graphs and also in the graph show in Fig 9D. This information is not detailed in the legends and nor is a time point for the analysis of what is presumably a non-fluorescent control detailed.

Figure 3A bottom and Fig 9D – the blue and red look very similar to each other.

Figure 8, Figure 9 and Fig S2– the Wt and *tasA* mutant colonies look unexpectedly similar. If the same image has been used in the figures this should be noted in the legend.

Figure 9 and Fig 2S the JC81 isolate shown looks unexpectedly similar. If the same image has been used in the figures this should be noted in the legend.

Figure S12 – the immunoblot in part B looks to show evidence of undisclosed splicing. Show full data collected. Also the legend has the wrong labels. No lane 4 instead WT and lanes 1-3. Ensure correct.

Figure 9A JC81 looks like Fig S2A JC81

Fig S20B – what is M – not detailed in the legend.

Fig 8C- is this a spliced image or are the lines shown for clarity?

Fig 4D WT 72 hrs same in Fig 9H and Fig 8A and Fig 8D

Fig 9C- show as dot plots like all other data- has very large error

Fig 8 A – graph WT and *tasA* data appear to be reproduced in Fig 8D and in other places.

Fig 9 G some data looks very similar in profile to data in Fig 4C

Fig 10 A repeat (presentation) of WT data in Fig 1A

Fig 10C – repeat (presentation) of data from Fig 1B

Fig 10 D repeat (presentation) of data from Fig 1D for WT and tasA.

Fig S14 repeat of WT data from Fig 5

Fig S16 repeat of WT data from Fig 6

Fig S17 repeat of WT data from Fig 6

Fig S18 is the image in 48 hours and 72 hours from the same experiment?

Fig S20B- is this spliced or are the lines for clarity. The way the bubbles look suggest that there are splicing events that have not been clearly disclosed.

In addition this reviewer feels that the data with the TasA-mCherry fusion are not convincing, and adding a reference to the previous Science paper is not an adequate response. The Science paper (<https://science.sciencemag.org/content/328/5978/627.long>) also does not contain adequate evidence showing the majority of mCherry signal is linked with TasA (and of course vice versa). There is an immunoblot in the Science supplemental data but the molecular mass of the bands, and controls to illustrate the specificity of reactivity of the antibody, are lacking. Moreover the corresponding authors have in effect refuted many of the findings in the work published initially in Science through further publications. This raises concerns about the reporter fusion - which is albeit not mentioned specifically in the subsequent papers.

As many of the experiments in the paper under review depend on this reporter, time to show conclusively that the majority of the mCherry signal associates with the reporter fusion seems warranted. For example, the inclusion of a negative control to show specificity of the mCherry antibody to the fusion (eg include a wild type sample and tasA mutant that are probed alongside the reporter fusion with the TasA and mCherry antibodies) would seem wise given the weak nature of the bands associated with the mCherry antibody in the image shown in the revised paper. It could be that the functionality of the reporter fusion in the biofilm is due to cleavage of mCherry and release of TasA that is free to act as "normal".

Papers that refute the general findings of the Science paper (but not specifically the integrity of the reporter fusions) that was quoted in the comments to reviewers are:

<https://www.ncbi.nlm.nih.gov/pubmed/24097941>

<https://www.ncbi.nlm.nih.gov/pubmed/25733611>

Reviewer #3 (Remarks to the Author):

I think the authors did a great job in improving their manuscript and cleared the concerns of the initial manuscript. Some questions still remain unanswered and would be looking forward of the future studies.

Reviewer's responses

Reviewer 2

Q-In this revised manuscript the authors have undertaken a number of experiments that have increased the quality of the work and should be commended for this. However there still seem to be a number of issues with the figures.

For instance, there appear to be a large number of cases of where data has been repeatedly presented in different parts of the manuscript and where the repetition has not been disclosed. This is not to infer that the data has been misrepresented as another strain etc – more that these issues have not been disclosed and it would suggest that there could be ways of condensing the data into more compressed formats. It would also be more transparent.

There are also other presentation issues that warrant attention in both the figures and the legends. Finally, check that time points are mentioned in the legend and that ALL of the data in the images is represented in the legend.

R- We would like to thank reviewer 2 for taking the time to review the revised version of the manuscript. We appreciate the reviewer's comments and the time invested in carefully reading the manuscript and thoroughly checking all the figures. We have addressed all the concerns raised by the reviewer and believe that the revised version of the manuscript has improved as a result.

Indeed, as correctly pointed out by the reviewer, the data from some experiments is repeated in some of the figures. This is done, most of the time, with results from the same experiments performed with different strains. For the sake of comparison, to illustrate differences between strains that are mentioned later on the text or the use of data from one of the strains as a control to compare with other strains (for example, the case of the WT or $\Delta tasA$ strains, repeated multiple times through several figures), we included data from the same experiment that was previously shown in an earlier figure. We believe that this improves the readability of the manuscript and makes all the points easier to follow. However, we have incorporated the reviewer's suggestions and we have merged some of the data that is repeated within the same figure and we have disclosed those cases in which data from other figures is repeated for explicative purposes.

We are grateful to the reviewer for spotting these issues. We would like to clarify that we are fully committed to transparency and all the cases in which certain information has not been disclosed in the figure legends (e.g. image cropping in blots, data repetition between figures, etc) have been honest mistakes that we have corrected in this version of the manuscript.

Below are the responses to the specific points raised by the reviewer.

Examples are given below but please note that this is NOT an exhaustive list:

Q- Figure 1- Need to add in E to the legend – currently there are two part D's.

R- Done.

Q- Figure 3 A – the grey histogram looks very similar between the top and bottom graphs and also in the graph show in Fig 9D. This information is not detailed in the legends and nor is a time point for the analysis of what is presumably a non-fluorescent control detailed.

Figure 3A bottom and Fig 9D – the blue and red look very similar to each other.

R- Indeed, as mentioned by the reviewer, the grey histogram is a non-fluorescent negative control corresponding to the unlabeled WT strain at 72 h. This has been clarified in the figure legends. The data of the negative control (grey), WT (red) or Δ tasA (blue) histograms in Figure 3A and repeated in Figure 9D, is repeated for the sake of the comparison between the different strains. The repetition of this data has been disclosed in the figure legend of Figure 9.

Q- Figure 8, Figure 9 and Fig S2– the Wt and tasA mutant colonies look unexpectedly similar. If the same image has been used in the figures this should be noted in the legend.

R- The WT and Δ tasA colony images in Figures 8 and 9 have been replaced to avoid data repetition.

Q- Figure 9 and Fig S2 the JC81 isolate shown looks unexpectedly similar. If the same image has been used in the figures this should be noted in the legend.

R- The JC81 image in Figure 9 has been replaced to avoid data repetition.

Q- Figure S12 – the immunoblot in part B looks to show evidence of undisclosed splicing. Show full data collected. Also, the legend has the wrong labels. No lane 4 instead WT and lanes 1-3. Ensure correct.

R- This has been corrected. Uncropped raw images of all the immunoblots presented in the figures are available in the source data file provided with the manuscript files. We have disclosed in the figure legends any splicing or cropping (when necessary) of all the blots presented in the figures. We have corrected the specific error in the legend pointed out by the reviewer.

Q- Figure 9A JC81 looks like Fig S2A JC81

R- The colony image in Figure 9A has been replaced.

Q- Fig S20B – what is M – not detailed in the legend.

R- Figure S20 containing the TasA-mCherry fusion colony phenotypes, the western blot and the fluorescent controls is now Figure S19. The western blot images have been replaced by new ones and will be detailed below.

Q- Fig 8C- is this a spliced image or are the lines shown for clarity?

R- It is a spliced image and the lines were shown for clarity as well. The lines have been removed from this and all the immunoblot images. The splicing of the immunoblot image has been disclosed in the figure legend.

Q- Fig 4D WT 72 hrs same in Fig 9H and Fig 8A and Fig 8D

R-In this case, the WT data is used as control in the comparison of the proportion of cell death between the different strains. Since all the experiments were performed simultaneously with all the strains, the WT data on the graph is repeated in some figures (this has been disclosed in the corresponding figure legends), however, we have replaced the cell death images by different microscopy fields of the WT sample to avoid repetition of data. In addition, Figure 8 has been condensed to avoid data repetition within the same figure and the Δ *tasA* data from the quantification of the proportion of cell death has been eliminated.

Q- Fig 9C- show as dot plots like all other data- has very large error

R- Figure 9C is now shown as dot plots. One outlier data-point in the value of the *alsS* gene expression in the Δ *tasA* strain has been excluded from the analysis.

Q- Fig 8 A – graph WT and *tasA* data appear to be reproduced in Fig 8D and in other places.

R- This has been clarified in the above comment and is mentioned in the corresponding figure legends.

Q- Fig 9 G some data looks very similar in profile to data in Fig 4C

R- Indeed, it is the same WT data that has been repeated as a control for comparing with strain JC81. This has been disclosed in the figure legend.

Q- Fig 10 A repeat (presentation) of WT data in Fig 1A, Fig 10C – repeat (presentation) of data from Fig 1B, Fig 10 D repeat (presentation) of data from Fig 1D for WT and *tasA*.

R- Figure 10 has been deleted and all the data corresponding to the JC81 strain has now been included as part of Figure 1 to avoid data repetition. Figure S1 also contains repetition of the WT data from Figure 1 (disclosed in the figure legend), as a control. However, the Δ *tasA* data has been removed to avoid data repetition and this figure only shows the data for the WT strain and the single ECM mutants that were not previously shown.

Q- Fig S14 repeat of WT data from Fig 5, Fig S16 repeat of WT data from Fig 6, Fig S17 repeat of WT data from Fig 6.

R- We have removed the WT images from Figure S14A, leaving only the JC81 images. Again, the WT data in the graph of Figure S14B is from the same experiment as the one from figure 5, that was performed with all the strains, therefore, we have left the WT data in Figure S14B as a control (this has been disclosed in the figure legend). Figures S15, S16 and S17 have been fused into one figure (new Figure S15) that shows only the images corresponding to the WT data. The graphs show the WT data from the same experiments as the ones displayed in Figure 6 and is used in this figure as a control (this has been disclosed in the figure legend). The data from the 24 h time-point in this figure has been eliminated. One of the graphs that did not fit into the figure is now the new Figure S16.

Q-Fig S18 is the image in 48 hours and 72 hours from the same experiment?

R- The colony images in figure S18 (now Figure S17) have been replaced for a new time-course experiment of the same Δ *sinI* and Δ *tasA*- Δ *eps* mixture colonies

Q- Fig S20B- is this spliced or are the lines for clarity. The way the bubbles look suggests that there are splicing events that have not been clearly disclosed.

R-The western blot images from Figure S20B (now S19B) have been replaced. As mentioned in the above comments, any lines over the images were placed for clarity, however, due to editorial recommendations, they have been removed from all the blot images in the manuscript. The cropping and splicing of the blot image shown in Figure S19B is disclosed in the figure legend and the corresponding raw images can be found in the source data file available with the manuscript files.

Q- In addition, this reviewer feels that the data with the TasA-mCherry fusion are not convincing and adding a reference to the previous Science paper is not an adequate response. The Science paper (<https://science.sciencemag.org/content/328/5978/627.long>) also does not contain adequate evidence showing the majority of mCherry signal is linked with TasA (and of course vice versa). There is an immunoblot in the Science supplemental data but the molecular mass of the bands, and controls to illustrate the specificity of reactivity of the antibody, are lacking. Moreover, the corresponding authors have in effect refuted many of the findings in the work published initially in Science through further publications. This raises concerns about the reporter fusion - which is albeit not mentioned specifically in the subsequent papers.

As many of the experiments in the paper under review depend on this reporter, time to show conclusively that the majority of the mCherry signal associates with the reporter fusion seems warranted. For example, the inclusion of a negative control to show specificity of the mCherry antibody to the fusion (eg include a wild type sample and tasA mutant that are probed alongside the reporter fusion with the TasA and mCherry antibodies) would seem wise given the weak nature of the bands associated with the mCherry antibody in the image shown in the revised paper. It could be that the functionality of the reporter fusion in the biofilm is due to cleavage of mCherry and release of TasA that is free to act as "normal".

R- As requested by the reviewer, we have performed a new fractionation assay from 48 h colonies (where the TasA expression is at its maximum) encoding the TasA-mCherry fusion. Given the weak bands observed for the anti-mCherry antibody shown in the previous Figure S20B, we have used a new anti-mCherry polyclonal antibody for this experiment (Invitrogen mCherry Polyclonal Antibody Catalog # PA5-34974, this

information has been updated from the Materials and Methods section and from the reporting summary file). The new blots are presented in the new Figure S19B. As suggested by the reviewer, lanes 1 to 4 contain the fractions corresponding to the unlabeled WT or Δ *tasA* samples (1 = WT cell fraction, 2 = WT ECM fraction, 3 = Δ *tasA* cell fraction, 4 = Δ *tasA* ECM fraction). Lanes 5 and 6 contain the cells or the ECM fraction respectively of the strain carrying the TasA-mCherry fusion construct (5 = Δ *tasA*, *amyE*::(*tapA-sipW-tasA-mCherry*) cell fraction, 6 = Δ *tasA*, *amyE*::(*tapA-sipW-tasA-mCherry*) ECM fraction). The western blot with the new anti-mCherry antibody shows a clear band in lanes 5 and 6 between the 50 and 70 Kda size range that corresponds to the expected size of the TasA-mCherry fusion (approximately 55 Kda). No clear bands are observable in this size-range in the corresponding WT or Δ *tasA* unlabeled samples (lanes 1 to 4). The same bands in the same size-range are observed when the same samples are probed against the anti-TasA antibody in lanes 5 and 6. No clear bands of the same size, as in the case of the anti-mCherry antibody, are observable in the unlabeled WT or Δ *tasA* controls (lanes 1 to 4). In conclusion, in this new western blot we have been able to unambiguously detect the size of the fusion protein only in the strains carrying the TasA-mCherry construct and not in the unlabeled controls.

Reviewer's comments:

Reviewer #2 (Remarks to the Author):

Regarding figure construction.

The authors have made improvements to the figures throughout.

Many of the improvements are adequate and the compression of the data in the figures is good, however not all of the figures are entirely satisfactory. While the legends state splicing of blots, the images don't make it easy to understand how many blots the final data presented is derived from.

The presentation of the spliced immunoblots are not in line with Nature Publishing guidelines. The abutting of the images through vertical splicing is not permitted. See guidance.

<https://www.nature.com/nature-research/editorial-policies/image-integrity#electrophoretic-gels-and-blots>

Regarding experimental data new to this manuscript:

Regarding the immunoblot looking at the integrity of the TasA-mCherry fusion protein. The data in the manuscript is of much better quality and now contains the control strains needed. However this immunoblot data clearly shows that the majority of the TasA in the samples is NOT in the form of the TasA-mCherry fusion. This means that the biological activity of the construct that has been "shown" to complement the TasA mutant strain cannot be conclusively linked with the TasA-mCherry form of the protein. The release TasA, which is in the majority, could be active.

These new data therefore open up the question of if the data associated with the reporter fusion are informative. It also raises the issue of if the data linked with this fusion protein are critical to the message or main conclusions of the paper.

That TasA has a range of impacts on the cell physiology is not in question. This has been shown clearly by the authors.

However, in light of the new immunoblots the data regarding the mechanism of how TasA promotes cell survival becomes weaker. It is specifically the evidence discussed in the text between lines 462-482 that is impacted. These are experiments linking localisation of TasA to FloT. The evidence that remains to show TasA can be membrane located is the detergent extraction of the membranes, which is less direct, but shown in a different manner.

Extracted from the abstract.

"The presence of TasA in cellular membranes, which would place it in proximity to functional membrane microdomains and the alteration of the normal distribution pattern of the flotillin-like protein FloT in Δ tasA cells led us to propose a role for TasA in the stabilization of membrane dynamics as cells enter stationary phase. Taken together, our results allow the separation of two complementary roles of this functional amyloid protein: i) structural functions during ECM assembly and interactions with plants, and ii) a physiological function in which TasA, via its localization to the cell membrane, stabilizes membrane dynamics and supports more effective cellular adaptation to

environmental cues.”

In conclusion, the strength of the conclusions made in the abstract is weakened, not ablated, if you question the validity of the use of the TasA-mCherry fusion to report on TasA localisation and function.

Minor points:

The authors talk about surfactin production being bimodal using this reference:

<https://www.ncbi.nlm.nih.gov/pubmed/19605685>

They have missed a later publication by the same corresponding author indicating that more recent studies show that transcription is unimodal in the population:

<https://www.ncbi.nlm.nih.gov/pubmed/25448819> Therefore this phrase deleted “reminiscent of the expression pattern reported for surfactin41.”

This sentence is an overstatement-

“The strain JC81, which expresses the TasA (Lys68Ala, Asp69Ala) variant protein, failed to rescue the biofilm formation phenotype in the WT strain (Fig. 9A, fig. S2B and fig. S12A). ”

There is a partial recovery of biofilm morphology as compared with the TasA deletion strain. Also the immunoblots do not support the conclusion that the variant form of the protein is stable (which is contrary to the variants of TasA the authors state they look to identify), there is obvious differential degradation compared with wild type TasA.

Reviewer's responses

Reviewer 2

Regarding figure construction.

The authors have made improvements to the figures throughout.

Many of the improvements are adequate and the compression of the data in the figures is good, however not all of the figures are entirely satisfactory. While the legends state splicing of blots, the images don't make it easy to understand how many blots the final data presented is derived from.

The presentation of the spliced immunoblots are not in line with Nature Publishing guidelines. The abutting of the images through vertical splicing is not permitted. See guidance.

<https://www.nature.com/nature-research/editorial-policies/image-integrity#electrophoretic-gels-and-blots>

R- We thank reviewer 2 for the time invested in carefully revising the manuscript. As suggested by the reviewer, the editorial team and guidelines of the journal, we have added black lines to delineate the boundaries of the splicing in those figures that show spliced immunoblot images. We have also added a text line to the corresponding figure legends to state what the black lines mean.

Regarding experimental data new to this manuscript:

Regarding the immunoblot looking at the integrity of the TasA-mCherry fusion protein. The data in the manuscript is of much better quality and now contains the control strains needed. However, this immunoblot data clearly shows that the majority of the TasA in the samples is NOT in the form of the TasA-mCherry fusion. This means that the biological activity of the construct that has been "shown" to complement the TasA mutant strain cannot be conclusively linked with the TasA-mCherry form of the protein. The release TasA, which is in the majority, could be active.

These new data therefore open up the question of if the data associated with the reporter fusion are informative. It also raises the issue of if the data linked with this fusion protein are critical to the message or main conclusions of the paper.

That TasA has a range of impacts on the cell physiology is not in question. This has been shown clearly by the authors.

However, in light of the new immunoblots the data regarding the mechanism of how TasA promotes cell survival becomes weaker. It is specifically the evidence discussed in the text between lines 462-482 that is impacted. These are experiments linking localization of TasA to FloT. The evidence that remains to show TasA can be membrane located is the detergent extraction of the membranes, which is less direct, but shown in a different manner.

Extracted from the abstract.

“The presence of TasA in cellular membranes, which would place it in proximity to functional membrane microdomains and the alteration of the normal distribution pattern of the flotillin-like protein FloT in Δ tasA cells led us to propose a role for TasA in the stabilization of membrane dynamics as cells enter stationary phase. Taken together, our results allow the separation of two complementary roles of this functional amyloid protein: i) structural functions during ECM assembly and interactions with plants, and ii) a physiological function in which TasA, via its localization to the cell membrane, stabilizes membrane dynamics and supports more effective cellular adaptation to environmental cues.”

In conclusion, the strength of the conclusions made in the abstract is weakened, not ablated, if you question the validity of the use of the TasA-mCherry fusion to report on TasA localization and function.

R- Based on the reviewer's comments regarding the data of the TasA-mCherry fusion and editorial recommendations, we have removed all the figures that include data of the fusion protein, and we have tone down those part of the manuscript, related to the presence of TasA in the cell membrane, by mentioning only the association of TasA to the DRM fraction of the membrane. It is our believe, as also noticed by the reviewer, that the data from the TasA-mCherry fusion is not critical to the message and the main conclusions are not affected. We have shown that: i) TasA is located in the membrane (through the western blot showing the association of TasA to the DRM in fig. 7C) and ii) the contribution of TasA to the cell membrane dynamics (as its absence leads to alterations in membrane fluidity, fig. 6A and B, right). As suggested by the reviewer, we

have also modified the abstract to tone down the conclusions regarding the presence of TasA in the cell membrane (lines 28-35).

Minor points:

The authors talk about surfactin production being bimodal using this reference:

<https://www.ncbi.nlm.nih.gov/pubmed/19605685>

They have missed a later publication by the same corresponding author indicating that more recent studies show that transcription is unimodal in the population: <https://www.ncbi.nlm.nih.gov/pubmed/25448819> Therefore this phrase deleted “reminiscent of the expression pattern reported for surfactin41.”

R- We appreciate this comment of the reviewer. It is totally true, thus the phrase “reminiscent of the expression pattern reported for surfactin”, as well as the corresponding reference, have been deleted from the manuscript.

This sentence is an overstatement:

“The strain JC81, which expresses the TasA (Lys68Ala, Asp69Ala) variant protein, failed to rescue the biofilm formation phenotype in the WT strain (Fig. 9A, fig. S2B and fig. S12A)”. There is a partial recovery of biofilm morphology as compared with the TasA deletion strain. Also, the immunoblots do not support the conclusion that the variant form of the protein is stable (which is contrary to the variants of TasA the authors state they look to identify), there is obvious differential degradation compared with wild type TasA.

R- We have modified the sentence to include the reviewer’s observation. It now reads: “The strain JC81, which expresses the TasA (Lys68Ala, Asp69Ala) variant protein, **failed to fully restore the WT biofilm formation phenotype** (Fig. 9A, fig. S2B and fig. S12A)” (lines 518-520).